# Universal Skeleton Understanding via Differentiable Rendering and MLLMs

**Ziyi Wang** [1] [*] **Peiming Li** [1] [*] **Xinshun Wang** [1] [*] **Yang Tang** [2] **Kai-Kuang Ma** [3] **Mengyuan Liu** [1] [†]

## Abstract

Multimodal large language models (MLLMs) exhibit strong visual-language reasoning, yet cannot process structured, non-visual data such as human skeletons. Existing methods either compress skeleton dynamics into lossy feature vectors for text alignment, or quantize motion into discrete tokens that generalize poorly across heterogeneous skeleton formats. We present SkeletonLLM, which achieves universal skeleton understanding by translating arbitrary skeleton sequences into the MLLM's native visual modality. At its core is DrAction, a differentiable, format-agnostic renderer that converts skeletal kinematics into compact image sequences. Because the pipeline is end-to-end differentiable, MLLM gradients can directly guide the rendering to produce task-informative visual tokens. To further enhance reasoning capabilities, we introduce a cooperative training strategy: Causal Reasoning Distillation transfers structured, step-by-step reasoning from a teacher model, while Discriminative Finetuning sharpens decision boundaries between confusable actions. SkeletonLLM demonstrates strong generalization in open-vocabulary action recognition, while its learned reasoning capabilities naturally extend to motion captioning and question answering across heterogeneous skeleton formats—suggesting a viable path for applying MLLMs to non-native modalities. Code: https://github.com/wangzy01/SkeletonLLM.

## 1. Introduction

Multimodal Large Language Models (MLLMs), such as GPT-5.2 (OpenAI, 2025) and Qwen-VL (Bai et al., 2025),

*Equal contribution †Corresponding author [1]State Key Laboratory of General Artificial Intelligence, Peking University Shenzhen Graduate School, Shenzhen, Guangdong, China [2]Tencent, Shenzhen, Guangdong, China [3]Nanjing University of Aeronautics and Astronautics, Nanjing, Jiangsu, China. Correspondence to: Mengyuan Liu <liumengyuan@pku.edu.cn>.

*Proceedings of the 43rd International Conference on Machine Learning*, Seoul, South Korea. PMLR 306, 2026. Copyright 2026 by the author(s).

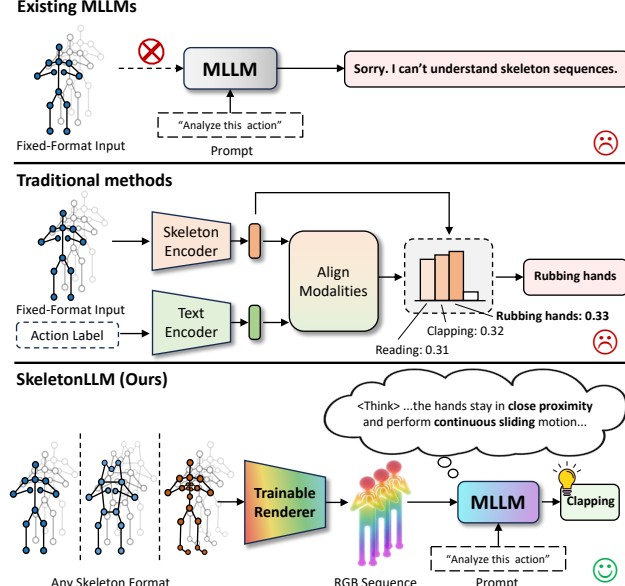

*Figure 1.* **Breaking Format Silos and the Modality Gap. (Top)** MLLMs possess strong reasoning capabilities but cannot natively process structured skeleton data. **(Middle)** Traditional alignment methods are tied to specific skeleton topologies, compressing motion into a single vector for matching against text embeddings, which creates representation bottlenecks and brittle semantics. **(Bottom)** Our SkeletonLLM uses DrAction, a differentiable renderer that translates a single skeleton sequence of any format into the MLLM's native visual language, enabling end-to-end optimization and unlocking powerful visual reasoning for diverse tasks.

have become powerful general-purpose reasoning engines. They show remarkable zero-shot generalization on vision and language tasks. Their success comes from learning rich, transferable representations from massive image-text corpora, enabling them to reason about novel visual concepts using broad world knowledge. A natural question arises: *Can we extend this reasoning capability to structured, non-visual data modalities that MLLMs cannot natively process, thereby achieving universal understanding of such data?*

Human skeleton sequences are a compelling case study. Skeletons offer a compact, privacy-preserving, and appearance-invariant representation of motion (Sun et al., 2022). This makes them attractive for applications where raw video is impractical due to bandwidth, storage, or privacy constraints, such as healthcare and human-robot interaction (Wang et al., 2025b). However, skeleton data faces a critical format silo problem. Skeleton topologies

vary widely across acquisition systems: Kinect v2 provides 25 joints (Zhang, 2012), MoCap systems use 22 SMPL joints (Loper et al., 2015), and 2D pose estimators output 17 COCO keypoints (Sun et al., 2019). This heterogeneity means a model trained on one format cannot be directly applied to another without costly retraining or error-prone joint remapping. Beyond recognition, skeleton-based methods also lack open-ended semantic understanding—they output discrete labels but cannot explain *why* an action occurs or answer natural language questions about fine-grained body dynamics. These limitations—format silos and shallow semantics—have long hindered the goal of universal skeleton understanding: a single model that comprehends any skeleton format across diverse tasks.

Existing approaches to bridge skeletons and foundation models fall into two paradigms, both with fundamental limitations. Feature-text alignment methods (Zhu et al., 2024; Chen et al., 2024b;a) train skeleton encoders (e.g., GCNs (Yan et al., 2018; Shi et al., 2018)) to align with text embeddings (e.g., CLIP (Radford et al., 2021)) in a shared latent space. However, these encoders are topology-specific and compress rich spatiotemporal dynamics into a single vector, creating a representation bottleneck that discards details crucial for distinguishing subtle actions. Tokenization-based LLM methods (Jiang et al., 2023; Chen et al., 2025) discretize motion into a learned codebook. But this quantization is inherently lossy, and the codebook inherits format dependency—changing the input skeleton structure requires retraining. Critically, both paradigms fail to leverage the MLLM's profound, pre-existing visual understanding. They force the model to learn a "pose language" from scratch rather than exploiting its native perceptual capabilities.

We propose **SkeletonLLM** (Figure 1), a framework that achieves universal skeleton understanding by "translating" skeleton sequences into the visual modality that MLLMs natively understand. At its core is **DrAction** (**D**ifferentiable **R**endering of **Action**s), a learnable renderer that converts skeletal kinematics into image sequences. DrAction is built on 3D Gaussian Splatting (Kerbl et al., 2023) and Linear Blend Skinning (Loper et al., 2015). It models each pose as deformable Gaussian primitives bound to the kinematic chain. Because the entire pipeline is differentiable, gradients from the MLLM flow back to guide the renderer, enabling it to learn visual representations that are maximally informative for downstream tasks. Crucially, DrAction is format-agnostic: it renders any skeleton—regardless of joint count, connectivity, or coordinate system—into a visually consistent representation, creating a universal visual interface that bypasses the format silo problem entirely.

To strengthen this visual translation, we introduce a cooperative training strategy. **Causal Reasoning Distillation (CR-Distill)** uses a teacher model to generate step-by-step causal descriptions of body-part dynamics (Hinton et al., 2015), guiding the student to learn structured reasoning rather than superficial label mapping. **Discriminative Fine-tuning (Disc-FT)** sharpens decision boundaries by training on binary judgments over confusing action pairs (e.g., "clapping" vs. "rubbing hands"). Together, these equip the model with causal understanding and fine-grained discrimination.

We validate SkeletonLLM on diverse tasks: open-vocabulary action recognition on NTU-60/120 (Shahroudy et al., 2016; Liu et al., 2019) and PKU-MMD (Liu et al., 2017), cross-format transfer (Kinect ↔ MoCap ↔ 2D poses), motion captioning on HumanML3D (Guo et al., 2022), and motion question answering. SkeletonLLM substantially outperforms prior methods, with gains amplifying under extreme data scarcity (e.g., +11.96% on the NTU-60 30/30 split). Notably, models trained on one skeleton format transfer directly to unseen formats without finetuning, demonstrating the format-agnostic nature of our visual translation approach.

**Our contributions are as follows:**

- We propose a new paradigm that translates skeletons into the MLLM's native visual language. For the first time, a single framework unifies heterogeneous skeleton formats (e.g., Kinect, MoCap, 2D poses) and supports multi-task understanding (recognition, captioning, reasoning), moving toward universal skeleton understanding.

- We introduce DrAction, the first differentiable, format-agnostic skeleton renderer for MLLMs, enabling task-optimized visual representations via end-to-end optimization.

- We design a cooperative training strategy (CR-Distill + Disc-FT) that instills causal reasoning and discriminative capability, yielding strong generalization to novel actions and unseen formats, and substantially outperforming prior methods.

## 2. Related Work

### 2.1. Skeleton-based Action Recognition

Skeleton sequences provide a compact, privacy-preserving, and appearance-invariant representation of human motion, making them attractive for action recognition. Related structured 3D inputs such as point-cloud videos also highlight the importance of robust spatiotemporal modeling (Li et al., 2025). Graph Convolutional Networks (GCNs) have become the dominant paradigm, with methods such as ST-GCN (Yan et al., 2018) and CTR-GCN (Chen et al., 2021) achieving strong supervised performance by modeling the spatial and temporal dependencies of joints. However, these architectures are intrinsically tied to a specific skeleton topology. A model trained on Kinect v2 data (25 joints)

cannot be directly applied to MoCap data (22 SMPL joints) or 2D pose estimations (17 COCO keypoints) without retraining or complex joint remapping. This format silo problem severely limits the deployment of skeleton-based models across heterogeneous data sources. Furthermore, these methods can only predict discrete class labels, lacking the capacity for open-ended semantic understanding, such as describing *why* an action is performed or answering questions about fine-grained body-part dynamics.

To generalize to novel action categories, research has extended to Open-Vocabulary Action Recognition. The dominant paradigm here is feature-text alignment (Zhu et al., 2024; Chen et al., 2024b; Do & Kim, 2025; Li et al., 2024; Chen et al., 2024a), where a skeleton encoder is aligned with text embeddings (e.g., CLIP (Radford et al., 2021)) in a shared latent space. While effective, this alignment paradigm suffers from a representation bottleneck: compressing rich spatiotemporal dynamics into a single global feature vector often discards the fine-grained motion details needed to distinguish subtle actions. It also suffers from brittle semantic alignment, where simple text labels produce nearly indistinguishable embeddings for similar actions.

## 2.2. LLM/MLLM for Motion Understanding

The rise of Large Language Models (LLMs) and Multimodal LLMs (MLLMs) has opened new avenues for motion understanding. A prominent approach is to tokenize continuous motion into a discrete "pose vocabulary" via VQ-VAE, as in MotionGPT (Jiang et al., 2023) and MotionLLM (Chen et al., 2025). Recent unified motion-text-vision frameworks such as UniMotion (Wang et al., 2026) further broaden this line toward both understanding and generation. This tokenization allows LLMs to process motion as a sequence of tokens. However, the tokenization paradigm introduces critical limitations: (1) the quantization process is inherently lossy, discarding subtle kinematic details; (2) the learned codebook generalizes poorly to novel skeleton formats; and (3) it imposes an artificial semantic gap, forcing the model to learn an abstract "pose language" instead of leveraging its profound, pre-existing visual understanding. SUGAR (Ye et al., 2026) instead adopts a more efficient coordinate-to-language pipeline via visual-motion supervision and Temporal Query Projection. An alternative strategy is to encode skeletons into the embedding space of MLLMs, as explored by SKI models (Sinha et al., 2025). However, this coordinate-based projection struggles to fully activate the MLLM's powerful pretrained perception of visual patterns and lacks a unified mechanism to natively handle diverse skeleton formats. In contrast, our work introduces a differentiable renderer that translates skeletons into the visual modality that MLLMs natively understand. This allows gradients from the MLLM to directly refine the rendering process, learning a visual representation that is maximally informative for the task at hand.

## 2.3. Universal Skeleton Representation

Achieving a universal skeleton representation that transcends format heterogeneity is a long-standing challenge. Prior attempts include joint remapping heuristics (Duan et al., 2022), zero-padding to a maximum joint count (Wang et al., 2024), or learning format-specific adapters (Guo et al., 2023; Wang et al., 2025a). These solutions are either lossy, introduce noise, or require retraining for each new format. Our approach sidesteps these issues entirely. By rendering any skeleton—regardless of its joint count, connectivity, or coordinate system—into a visually consistent image sequence, DrAction creates a format-agnostic visual lingua franca. This allows a single model to seamlessly process data from Kinect, MoCap, 2D pose estimators, or any other source, enabling true cross-format generalization without architectural modifications or retraining.

# 3. Method

## 3.1. Problem Formulation and Overview

We address the problem of enabling MLLMs to understand structured, non-visual data outside their native modalities. Specifically, we focus on human skeleton sequences—a modality that is compact, privacy-preserving, and appearance-invariant, yet fundamentally incompatible with the image-text paradigm on which MLLMs are trained.

Let $\mathbf{S} = \{\mathbf{p}_t\}_{t=1}^{T}$ denote a skeleton sequence, where $\mathbf{p}_t \in \mathbb{R}^{P \times J \times 3}$ represents the 3D coordinates of $J$ joints for $P$ persons at frame $t$. Our goal is to construct a mapping $\mathcal{R} : \mathbf{S} \mapsto \mathbf{V}$ that translates $\mathbf{S}$ into a visual representation $\mathbf{V} = \{\mathbf{I}_t\}_{t=1}^{T'}$ that the MLLM can natively process, while satisfying three desiderata:

1. **Differentiability**: $\mathcal{R}$ must be end-to-end differentiable, allowing gradients from the MLLM's task loss to optimize the translation process.
2. **Format Agnosticism**: $\mathcal{R}$ must handle heterogeneous skeleton topologies (varying $J$, different joint definitions, 2D/3D coordinates) without architectural modifications.
3. **Information Preservation**: $\mathcal{R}$ must preserve fine-grained spatiotemporal dynamics necessary for distinguishing subtle actions.

We realize this mapping through **DrAction** (**D**ifferentiable **R**endering of **Action**s), a learnable renderer built on 3D Gaussian Splatting and Linear Blend Skinning. As illustrated in Figure 2, the complete SkeletonLLM framework operates as a Render-Reason-Respond pipeline: DrAction translates skeleton sequences into image sequences, which are then processed by the MLLM's vision encoder to produce visual tokens for the language model. A cooperative training strategy further enhances reasoning and discriminative capabilities.

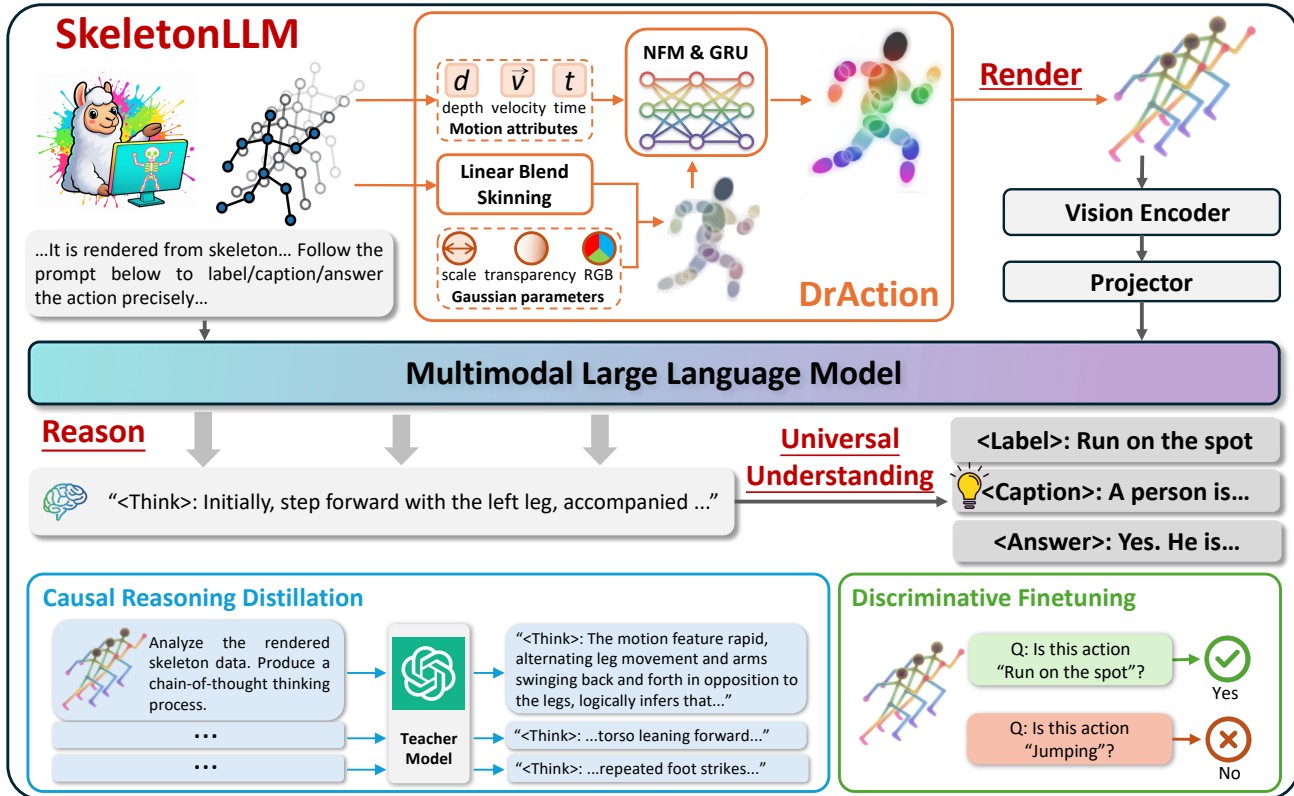

*Figure 2.* Overview of SkeletonLLM. The pipeline follows a Render-Reason-Respond process for universal understanding. Given a skeleton sequence, DrAction lifts joint trajectories into deformable 3D Gaussian primitives and renders motion-aware images. Joint transforms are computed via Linear Blend Skinning, and kinematic cues (depth, velocity) are fused through a Neural Feature Modulator. All parameters are optimized end-to-end by gradients from the MLLM. The rendered frames are processed by the MLLM's vision encoder and a projector to yield visual tokens. During training, CR-Distill supervises with teacher-generated causal chains describing body-part dynamics, while Disc-FT sharpens decision boundaries via binary queries over confusing action pairs.

## 3.2. DrAction: Differentiable Skeleton Renderer

Existing skeleton visualization methods (e.g., stick figures, heatmaps) are fixed transformations that cannot adapt to downstream tasks. We design DrAction as a learnable visual interface between skeletal kinematics and MLLM perception. The key insight is to model each pose as deformable 3D Gaussian primitives bound to the kinematic chain, enabling both differentiability and expressiveness.

**Gaussian Primitive Representation** We model the human body as $K$ deformable 3D Gaussian primitives rather than meshes. To ensure topology invariance, we define these primitives in a canonical, pose-independent space.

For a skeleton with $J$ joints and varying bone connectivity, we instantiate $K = J + K_{\text{bone}}$ Gaussians, where $K_{\text{bone}} = |\mathcal{E}| \times N_{\text{samples}}$ ($|\mathcal{E}|$: number of bone edges from the skeleton's adjacency list; $N_{\text{samples}}$=10: intermediate points per bone). The first $J$ Gaussians are anchored at joints, while $K_{\text{bone}}$ Gaussians are uniformly sampled along bone edges via linear interpolation. Because both $J$ and $|\mathcal{E}|$ are read from the input skeleton, the primitive count adapts automatically to any topology (see Appendix C.2). This density ensures continuous visual connectivity even for sparse skeletons. Each Gaussian $k$ in canonical space is parameterized

as:

$$\mathcal{G}_k^c = \{\boldsymbol{\mu}_k^c, \mathbf{s}_k^c, \mathbf{q}_k^c, \alpha_k^c, \mathbf{f}_k\} \quad (1)$$

where $\boldsymbol{\mu}_k^c \in \mathbb{R}^3$ is the center, $\mathbf{s}_k^c \in \mathbb{R}^3$ is the scale, $\mathbf{q}_k^c \in \mathbb{R}^4$ is the rotation quaternion, $\alpha_k^c \in [0, 1]$ is the base opacity, and $\mathbf{f}_k \in \mathbb{R}^d$ is a learnable appearance feature. Unlike standard 3DGS which optimizes per-scene parameters, here $\boldsymbol{\mu}_k^c$ is initialized from the first frame of each sequence, while $\mathbf{f}_k$, scales $\mathbf{s}_k^c$, and orientations $\mathbf{q}_k^c$ are learned globally across the dataset.

**Kinematic Deformation via Linear Blend Skinning** To animate canonical Gaussians according to input skeletal motion, we employ Linear Blend Skinning (LBS) (Loper et al., 2015)—a standard technique in computer graphics that provides a principled, differentiable mapping from joint configurations to surface deformations.

**Per-Joint Rigid Transforms.** For each joint $i$ at frame $t$, we compute a rigid transformation $\mathbf{T}_i \in \mathrm{SE}(3)$ that maps from the canonical pose to the current pose. While Section 3.1 defines a skeleton purely by joint *positions*, some capture systems (e.g., Kinect v2) also record per-joint orientation quaternions that describe the local rotation of the bone segment attached to each joint; these rotations are applied to the Gaussian primitives bound to that joint, not to the joint point

itself. Given the current joint position $\mathbf{j}_i^t$ and orientation quaternion $\mathbf{q}_i^t$ (when available), the transformation is:

$$\mathbf{R}_i = \mathrm{quat2mat}(\mathbf{q}_i^t), \quad \mathbf{t}_i = \mathbf{j}_i^t - \mathbf{j}_i^c, \quad \mathbf{T}_i = \begin{bmatrix} \mathbf{R}_i & \mathbf{t}_i \\ \mathbf{0}^\top & 1 \end{bmatrix} \quad (2)$$

where $\mathbf{j}_i^c$ denotes the canonical joint position—the position of joint $i$ in the first frame of the input sequence, serving as the rest-pose reference for computing per-frame displacements. When orientation data is unavailable (e.g., for 2D pose estimations or position-only MoCap), we set $\mathbf{R}_i = \mathbf{I}_3$, reducing LBS to translation-only skinning—a key mechanism enabling format agnosticism.

**Blending and SO(3) Projection.** Each Gaussian $k$ is associated with the skeleton via blend weights $\mathbf{w}_k \in \Delta^{J-1}$ (on the probability simplex). These weights are pre-computed and fixed based on the skeleton's topology: a joint Gaussian at joint $j$ receives a one-hot weight ($w_{k,j}{=}1$); a bone Gaussian interpolated between joints $a$ and $b$ at factor $\alpha$ has $w_{k,a}{=}1{-}\alpha$, $w_{k,b}{=}\alpha$ (zero elsewhere). Thus, the summation in Equation (3) reduces to at most two non-zero terms for any Gaussian. The blended translation and rotation are computed as:

$$\mathbf{t}_k = \sum_{i=1}^{J} w_{k,i}\mathbf{t}_i, \qquad \tilde{\mathbf{R}}_k = \sum_{i=1}^{J} w_{k,i}\mathbf{R}_i \qquad (3)$$

Since the linear combination $\tilde{\mathbf{R}}_k$ does not generally lie in SO(3), we project it onto the rotation manifold using polar decomposition via SVD:

$$\tilde{\mathbf{R}}_k = \mathbf{U}\boldsymbol{\Sigma}\mathbf{V}^\top \Rightarrow \mathbf{R}_k = \mathbf{U}\,\mathrm{diag}(1, 1, \det(\mathbf{U}\mathbf{V}^\top))\,\mathbf{V}^\top \quad (4)$$

This projection is differentiable and numerically stable, ensuring valid rotations even for extreme poses.

**Gaussian Transformation.** The transformed Gaussian parameters are:

$$\boldsymbol{\mu}_k = \mathbf{R}_k\boldsymbol{\mu}_k^c + \mathbf{t}_k, \qquad \mathbf{R}_k^{\mathrm{tot}} = \mathbf{R}_k\mathbf{R}_k^c \qquad (5)$$

$$\boldsymbol{\Sigma}_k = \mathbf{R}_k^{\mathrm{tot}} \cdot \mathrm{diag}\big((\mathbf{s}_k^c)^2\big) \cdot (\mathbf{R}_k^{\mathrm{tot}})^\top \qquad (6)$$

where $\mathbf{R}_k^c = \mathrm{quat2mat}(\mathbf{q}_k^c)$ is the canonical orientation.

**Neural Feature Modulator for Pose-Conditioned Appearance** Static appearance cannot capture the dynamic nature of actions—the same joint configuration may correspond to different motion phases (e.g., arm rising vs. falling). We introduce a **Neural Feature Modulator (NFM)** that adaptively adjusts each Gaussian's color and opacity based on local kinematics.

For each Gaussian $k$, we aggregate position $\mathbf{p}_k^t = \sum_i w_{k,i}\mathbf{j}_i^t$ and velocity $\mathbf{v}_k^t = \sum_i w_{k,i}\dot{\mathbf{j}}_i^t$ (finite-difference: $\dot{\mathbf{j}}_i^t = \mathbf{j}_i^t - \mathbf{j}_i^{t-1}$) from associated joints, concatenate with base appearance from a lightweight MLP, and process through a single-layer GRU for temporal modeling (it captures motion-phase cues such as acceleration vs. deceleration;

see Appendix A.3 for ablations on temporal modeling choices). The NFM then predicts RGB and opacity residuals ($\Delta\mathrm{RGB}_k, \Delta\alpha_k$) plus a saliency gate $g_k$. The gate modulates the final opacity as $\alpha_k{=}\sigma(\alpha_k^{\mathrm{base}}{+}\Delta\alpha_k){\cdot}\sigma(g_k)$, suppressing visually uninformative (e.g., stationary) primitives while amplifying motion-salient ones. The final color blends the modulated appearance with a depth-based colormap: $\mathbf{C}_k = (1{-}\lambda)\sigma(\mathrm{RGB}_k^{\mathrm{base}}{+}\Delta\mathrm{RGB}_k) + \lambda\mathbf{C}_k^{\mathrm{depth}}$, where $\sigma(\cdot)$ is the sigmoid function and $\lambda{=}\sigma(\theta_{\mathrm{mix}})$ is a learnable mixing weight.

**Differentiable Rasterization** We render the transformed Gaussians using the differentiable rasterizer from 3DGS (Kerbl et al., 2023). Each Gaussian is projected onto the image plane via perspective projection:

$$\boldsymbol{\mu}_{k,2D} = \pi(\mathbf{K}, \mathbf{W}, \boldsymbol{\mu}_k), \qquad \boldsymbol{\Sigma}_{k,2D} = \mathbf{J}\boldsymbol{\Sigma}_k\mathbf{J}^\top \quad (7)$$

where $\mathbf{K}$ is the camera intrinsic matrix, $\mathbf{W}$ is the world-to-camera transform, and $\mathbf{J}$ is the projection Jacobian.

The pixel color is computed via front-to-back alpha compositing over depth-sorted Gaussians:

$$\mathbf{I}(x,y) = \sum_{k\in\mathcal{N}(x,y)} \mathbf{C}_k\alpha_k' \prod_{j<k}(1 - \alpha_j') \qquad (8)$$

where $\alpha_k' = 1 - \exp(-\alpha_k \cdot w_k(x,y))$ and $w_k(x,y)$ is the Gaussian influence at pixel $(x,y)$.

### 3.3. Vision-Language Backbone Integration

We integrate DrAction as a differentiable front-end to a pre-trained MLLM, forming a unified skeleton-to-language pipeline. The rendered frames $\mathbf{V}$ are processed by the MLLM's vision encoder (ViT) to produce visual tokens, which are projected into the language model's embedding space via a learnable MLP. These tokens replace `<image>` placeholders in the text prompt, enabling the language model to reason about the depicted motion.

**Generative classification via MQA.** Since the MLLM is an auto-regressive generative model, we do *not* add a separate classification head. Instead, action recognition is cast as a *multiple-choice question answering* (MQA) task: the text prompt lists candidate action labels and instructs the model to generate a short answer ending with `Label: <action>`. The predicted label is extracted via pattern matching at inference. All training losses below are therefore standard auto-regressive cross-entropy on the target token sequence, with prompt tokens masked (see Appendix E.3.1 for prompt templates).

The entire architecture—from skeleton input through rendering, visual encoding, and language generation—is differentiable. This allows the MLLM's task-specific gradients to propagate back through the rendering pipeline, enabling DrAction to learn visual representations that are maximally informative for downstream objectives.

## 3.4. Cooperative Training Strategy

Jointly optimizing a randomly initialized renderer with a pre-trained MLLM presents a "chicken-and-egg" dilemma: the MLLM requires recognizable visuals for meaningful supervision, while the renderer needs gradients to learn visualization. We address this via a four-stage cooperative training strategy that progressively instills visual intelligibility, discriminative precision, and causal reasoning.

**Stage 1: Alignment Warm-up**  We first establish a baseline visual protocol. Keeping the MLLM frozen, we optimize only the DrAction renderer ($\Theta_{\text{render}}$). The objective is a multiple-choice question answering (MQA) task where the model selects the correct action from candidates. This forces the renderer to discover a visual mapping intelligible to the pre-trained vision encoder, aligning skeleton semantics with the MLLM's existing visual priors.

**Stage 2: Discriminative Finetuning (Disc-FT)**  To address "brittle semantics" between similar actions (e.g., "rubbing hands" vs. "clapping"), we introduce a contrastive-like binary classification task. We construct hard negative pairs and ask the model: "*Does this video show [Action A]?*" (when it actually shows similar Action B). We update the renderer ($\Theta_{\text{render}}$) and projector ($\Theta_{\text{proj}}$); the model generates a single answer token ("Yes"/"No") and we minimize the auto-regressive negative log-likelihood of this token—equivalently, binary cross-entropy on the positive-token probability—denoted $\mathcal{L}_{\text{Disc}}$ (see Appendix E.3.2). This directs the renderer to attend to subtle, discriminative motion details.

**Stage 3: Causal Reasoning Distillation (CR-Distill)**  Moving beyond recognition to deep understanding, we distill structured reasoning from a stronger teacher model (Hurst et al., 2024). We prompt the teacher to generate a causal chain of thought: step-by-step analysis of body-part movements, temporal evolution, and potential intent, followed by a conclusion. Our model is trained to generate this complete rationale given the rendered video. We optimize the renderer ($\Theta_{\text{render}}$), projector ($\Theta_{\text{proj}}$), and LLM (via LoRA, $\Theta_{\text{LoRA}}$) using auto-regressive loss $\mathcal{L}_{\text{CR}}$ over the full teacher token sequence (prompt tokens masked; see Appendix E.3.3). This instills structured, causal action understanding, enabling the model to explain *why* an action is classified as such.

**Stage 4: Recognition Refinement**  Finally, we freeze the mature renderer and update only the projector ($\Theta_{\text{proj}}$) and LLM ($\Theta_{\text{LoRA}}$) to refine the mapping for open-vocabulary recognition, minimizing auto-regressive cross-entropy loss $\mathcal{L}_{\text{MQA}}$ on the target label tokens (same as Stage 1).

This strategy effectively decouples the complexity of learning to render from learning to reason, ensuring stable convergence and strong generalization.

## 4. Experiments

We design experiments to answer the following questions:

1. **Generalization to novel actions**: Can SkeletonLLM recognize actions unseen during training, especially under extreme data scarcity?
2. **Cross-format transfer**: Can SkeletonLLM generalize across heterogeneous skeleton formats (e.g., Kinect v2 → MoCap) without retraining?
3. **Semantic understanding**: Beyond classification, can SkeletonLLM perform open-ended motion understanding tasks such as captioning and question answering?
4. **Component contributions**: What are the individual contributions of DrAction and each training stage?

### 4.1. Experimental Setup

**Datasets**  We evaluate SkeletonLLM on diverse benchmarks spanning different skeleton formats, acquisition systems, and task types:

- **NTU-60 & NTU-120** (Shahroudy et al., 2016; Liu et al., 2019): Two large-scale benchmarks captured with Kinect v2 (25 joints). NTU-60 contains 60 action classes, and NTU-120 extends it to 120 classes. For NTU-60, we adopt the seen/unseen splits from Gupta et al. (2021) (55/5, 48/12) and the more challenging splits from Zhu et al. (2024) (40/20, 30/30). For NTU-120, we use 110/10, 96/24, 80/40, and 60/60 splits.
- **NTU-60 (2D)** (Duan et al., 2022): 2D pose estimations (17 joints) obtained from NTU-60 RGB videos using HRNet (Sun et al., 2019), used for cross-format evaluation from MoCap to 2D poses.
- **NW-UCLA** (Wang et al., 2014): A dataset captured with Kinect v1 (20 joints), used for cross-format evaluation.
- **HumanML3D** (Guo et al., 2022): A motion-language dataset based on AMASS MoCap data (22 SMPL joints), used for motion captioning and cross-format evaluation.

**Implementation Details**  SkeletonLLM is built upon InternVL3-8B (Zhu et al., 2025b) and trained on 2 NVIDIA H20 GPUs. For skeleton rendering, each joint and bone segment is modeled as a set of 3D Gaussians, with bone segments constructed by uniformly sampling 10 intermediate points along each connection. We sample 12 frames per sequence using a uniform segment-based strategy and render at $448 \times 448$ resolution. The four progressive training stages are trained for 1, 1, 1, and 3 epochs, respectively. We use the AdamW optimizer with a learning rate of $2 \times 10^{-5}$ and LoRA adapters (rank=32, $\alpha$=64) for efficient finetuning.

**Baselines**  We compare three categories of methods:

- **Traditional alignment methods**: Methods such as PURLS (Zhu et al., 2024) and TDSM (Do & Kim, 2025) that use skeleton encoders aligned with text embeddings.
- **LLM-based motion understanding**: Methods that to-

*Table 1.* Top-1 accuracy (%) of various methods on NTU-60 and NTU-120. Each split is denoted as X/Y, where X is the number of seen classes and Y is the number of unseen classes. The best results are in **red**, and the second-best are blue. †Results for Qwen2.5-VL-7B and InternVL3-8B were obtained by rendering skeletons with the non-learnable 3D+Velocity renderer (same as in Table 7; $448\times448$) and finetuning on MQA for 6 epochs.

| Methods | Type | Venue | NTU-60 (Acc, %) | | | | NTU-120 (Acc, %) | | | |
|---|---|---|---|---|---|---|---|---|---|---|
| | | | 55/5 | 48/12 | 40/20 | 30/30 | 110/10 | 96/24 | 80/40 | 60/60 |
| ReViSE (Hubert Tsai et al., 2017) | Align | ICCV'17 | 53.91 | 17.49 | 24.26 | 14.81 | 55.04 | 32.38 | 19.47 | 8.27 |
| JPoSE (Wray et al., 2019) | Align | ICCV'19 | 64.82 | 28.75 | 20.05 | 12.39 | 51.93 | 32.44 | 13.71 | 7.65 |
| CADA-VAE (Schonfeld et al., 2019) | Align | CVPRW'19 | 76.84 | 28.96 | 16.21 | 11.51 | 59.53 | 35.77 | 10.55 | 5.67 |
| SynSE (Gupta et al., 2021) | Align | ICIP'21 | 75.81 | 33.30 | 19.85 | 12.00 | 62.69 | 38.70 | 13.64 | 7.73 |
| PURLS (Zhu et al., 2024) | Align | CVPR'24 | 79.23 | 40.99 | 31.05 | 23.52 | 71.95 | 52.01 | 28.38 | 19.63 |
| SCoPLe (Zhu et al., 2025a) | Align | CVPR'25 | 84.10 | 52.96 | - | - | 74.53 | 52.17 | - | - |
| TDSM (Do & Kim, 2025) | Align | ICCV'25 | 86.49 | 56.03 | 36.09 | 25.88 | 74.15 | 65.06 | 36.95 | 27.21 |
| MotionGPT (Jiang et al., 2023) | LLM | NeurIPS'23 | 29.88 | 15.91 | 12.14 | 8.57 | 31.10 | 20.39 | 12.96 | 5.15 |
| MotionLLM (Chen et al., 2025) | LLM | TPAMI'25 | 50.24 | 26.98 | 21.58 | 16.46 | 49.80 | 33.62 | 16.91 | 11.45 |
| Qwen2.5-VL-7B† (Bai et al., 2025) | MLLM | - | 76.08 | 53.70 | 31.76 | 26.95 | 63.25 | 56.86 | 35.17 | 25.12 |
| InternVL3-8B† (Zhu et al., 2025b) | MLLM | - | 79.66 | 56.28 | 32.04 | 28.15 | 67.03 | 58.33 | 36.06 | 26.48 |
| **SkeletonLLM (Ours)** | MLLM | - | **87.37** | **64.72** | **46.15** | **37.84** | **76.05** | **67.20** | **44.37** | **34.94** |

kenize motion via VQ-VAE for LLM processing (Jiang et al., 2023; Chen et al., 2025) or encode skeletons into VLMs (Sinha et al., 2025).

- **Finetuned MLLMs**: Qwen2.5-VL and InternVL3 rendered with the non-learnable 3D+Velocity renderer (same as in Table 7; 12 frames, $448\times448$, identical front-view camera), finetuned on MQA recognition for 6 epochs.

### 4.2. Open-Vocabulary Action Recognition

We first evaluate SkeletonLLM on standard action recognition benchmarks, focusing on its ability to generalize to action classes unseen during training.

**Main Results on NTU-60 & NTU-120**   Table 1 compares SkeletonLLM against state-of-the-art methods on NTU-60 and NTU-120. To rigorously assess generalization, we evaluate not only conventional splits (Gupta et al., 2021) but also extreme splits (Zhu et al., 2024) that sharply reduce the number of seen classes.

**Performance gap widens under data scarcity.**   On the standard 55/5 split, SkeletonLLM outperforms the best traditional method (TDSM) by 0.88%. However, on the extreme 30/30 split, this gap expands to 11.96%. This trend is consistent on NTU-120 (7.73% improvement on the 60/60 split). We attribute this to two factors: (1) DrAction learns visual representations optimized for the MLLM rather than fixed encodings, enabling better generalization; (2) the MLLM's pretrained visual knowledge provides a strong inductive bias for understanding novel motion patterns.

**LLM-based methods underperform finetuned MLLMs.** Despite using large language models, tokenization-based methods (MotionGPT, MotionLLM) consistently lag behind traditional alignment methods and our approach. This suggests that quantizing continuous motion into discrete tokens creates an information bottleneck, discarding the fine-grained kinematic details needed to distinguish similar

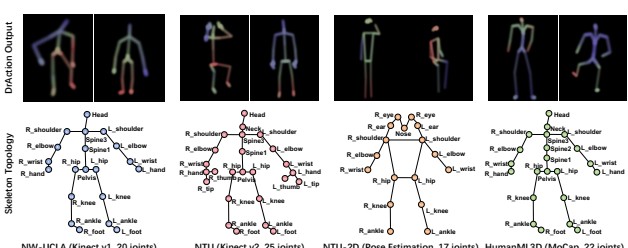

*Figure 3.* Cross-format rendering by DrAction. Top row: DrAction renders skeletons from four different formats into visually consistent image sequences. Bottom row: the underlying skeleton topologies vary significantly in joint count and connectivity—NW-UCLA (Kinect v1, 20 joints), NTU (Kinect v2, 25 joints), NTU-2D (pose estimation, 17 joints), and HumanML3D (MoCap, 22 joints). Despite these differences, DrAction produces a unified visual language, enabling seamless cross-format transfer.

actions. In contrast, our rendering-based approach preserves the full spatiotemporal richness of motion.

**Differentiable rendering is key.**   Comparing SkeletonLLM with InternVL3 using fixed rendering reveals the importance of end-to-end optimization. Despite using the same backbone, SkeletonLLM achieves 8.44% and 9.69% higher accuracy on the 48/12 and 30/30 splits of NTU-60, respectively. This confirms that gradients from the MLLM guide DrAction to produce visual representations that are maximally informative for the recognition task.

### 4.3. Cross-Format Generalization

A limitation of existing methods is their dependence on specific skeleton topologies. Models trained on Kinect data (25 joints) cannot directly process MoCap data (22 joints) without joint remapping or retraining. We investigate whether SkeletonLLM can overcome this "format silo" problem.

**Experimental Protocol**   We design cross-format transfer scenarios (NTU-60→NW-UCLA, NTU-60→HumanML3D, HumanML3D→NW-UCLA, and HumanML3D→NTU-60

*Table 2.* Cross-format transfer accuracy (%) for action recognition. Models are trained on the source dataset and evaluated directly on the target without finetuning.

| Training | Testing | Skeleton Format | Method | Acc |
|---|---|---|---|---|
| NTU-60 | NW-UCLA | Kinect v2 → Kinect v1 | TDSM | 43.19 |
| NTU-60 | NW-UCLA | Kinect v2 → Kinect v1 | MotionGPT | 10.35 |
| NTU-60 | NW-UCLA | Kinect v2 → Kinect v1 | SKI-LVLM | 31.87 |
| NTU-60 | NW-UCLA | Kinect v2 → Kinect v1 | **SkeletonLLM** | **60.38** |
| HumanML3D | NW-UCLA | MoCap → Kinect v1 | MotionGPT | 9.11 |
| HumanML3D | NW-UCLA | MoCap → Kinect v1 | SKI-LVLM | 28.74 |
| HumanML3D | NW-UCLA | MoCap → Kinect v1 | **SkeletonLLM** | **56.73** |
| HumanML3D | NTU-60 (2D) | MoCap → 2D Pose | MotionGPT | 5.69 |
| HumanML3D | NTU-60 (2D) | MoCap → 2D Pose | SKI-LVLM | 17.13 |
| HumanML3D | NTU-60 (2D) | MoCap → 2D Pose | **SkeletonLLM** | **40.36** |

*Table 3.* Cross-format motion captioning on HumanML3D. Models are trained on NTU-60 (Kinect v2, 25 joints) with recognition supervision and evaluated on HumanML3D (SMPL, 22 joints) captioning without finetuning.

| Method | R@1↑ | R@3↑ | MM-Dist↓ | BLEU@4↑ | ROUGE-L↑ | CIDEr↑ | BertScore↑ |
|---|---|---|---|---|---|---|---|
| MotionGPT | 2.83 | 9.86 | 12.27 | 3.50 | 13.83 | 4.16 | 14.03 |
| MotionLLM | 4.90 | 14.34 | 9.67 | 5.57 | 19.72 | 5.08 | 23.44 |
| InternVL3-8B | 6.25 | 20.08 | 6.36 | 7.15 | 27.01 | 9.49 | 29.45 |
| **Ours** | **11.60** | **28.56** | **5.94** | **9.63** | **39.84** | **18.25** | **37.28** |

(2D)) with no finetuning on the target domain. For baselines: MotionGPT uses its expected 263-dim representation after format conversion; SKI-LVLM/TDSM use zero-padding following Wang et al. (2024) and train from scratch.

**Cross-Format Action Recognition** Table 2 presents results for three cross-format transfer scenarios. When skeleton topology changes, both traditional methods and tokenization-based LLM methods degrade significantly. Zero-padding partially alleviates this but introduces noise that corrupts learned spatial relationships. In contrast, DrAction renders skeletons of different formats into the same visual language—image sequences depicting human motion. As shown in Figure 3, despite significant differences in skeleton topology (joint count and connectivity), DrAction produces visually consistent renderings. This "visual lingua franca" enables motion reasoning regardless of joint configuration. SkeletonLLM achieves 17.19% and 27.99% improvements over the best baselines for NTU-60→NW-UCLA and HumanML3D→NW-UCLA transfers, respectively. For HumanML3D→NTU-60 (2D), transferring from 3D MoCap to 2D poses, SkeletonLLM achieves 40.36% accuracy with a 23.23% improvement, demonstrating robust generalization even when depth information is entirely absent in the target domain.

**NTU-60 → HumanML3D (Motion Captioning)** This experiment tests a more challenging scenario: transferring from a recognition-oriented dataset (NTU-60) to a captioning-oriented dataset (HumanML3D), alongside a format shift from Kinect to SMPL. As shown in Table 3,

*Table 4.* Single-model multi-task evaluation. All results use the same NTU-60 (55/5) checkpoint with no task-specific retraining.

| Task | Test Set | Format ($J$) | Superv.? | Metric |
|---|---|---|---|---|
| Open-vocab recog. | NTU-60 (55/5) | Kinect v2 (25) | ✓ | 87.37% Acc |
| Cross-format recog. | NW-UCLA | Kinect v1 (20) | ✗ | 60.38% Acc |
| Cross-format caption. | HumanML3D | SMPL (22) | ✗ | 37.28 BertScore |
| Motion QA | Skeleton-QA | Kinect v2 (25) | ✗ | 68 / 65% Acc |

*Table 5.* Cross-dataset joint training. NTU-60 (25J) + HumanML3D (22J) with shared DrAction and MLLM.

| Training | NTU-60 55/5 | NTU→NW-UCLA | Skeleton-QA (Temp / Causal) |
|---|---|---|---|
| NTU-60 only | 87.37 | 60.38 | 68 / 65 |
| Joint training | **88.52** | **65.04** | **73.2 / 70.9** |

despite never seeing SMPL-format skeletons or captioning supervision during training, SkeletonLLM demonstrates remarkable transfer capability. This success stems from: (1) DrAction's format-agnostic rendering produces consistent visual representations regardless of joint topology; (2) the CR-Distill stage trains the model to generate descriptive causal chains, which naturally transfers to captioning tasks. SkeletonLLM substantially outperforms both InternVL3-8B (+5.35% R@1, +8.76 CIDEr over InternVL3-8B) and tokenization-based methods (MotionGPT, MotionLLM), demonstrating that our visual translation approach creates more transferable representations.

**Single-Model Multi-Task Evaluation** A key question is whether a *single* model can support all the tasks evaluated above. Table 4 confirms this: the same NTU-60 (55/5) checkpoint—with no task-specific retraining—is used across open-vocabulary recognition (Table 1), cross-format recognition (Table 2), cross-format captioning (Table 3), and motion QA (Appendix Table 15). This demonstrates that SkeletonLLM's visual translation approach yields broadly transferable representations from a single training run.

**Cross-Dataset Joint Training** To further validate the unified framework, we jointly train on NTU-60 (Kinect v2, 25J) and HumanML3D (SMPL, 22J) with shared DrAction and shared MLLM. As shown in Table 5, joint training yields consistent positive transfer: NTU-60 recognition improves by 1.15%, cross-format transfer to NW-UCLA by 4.66%, and motion QA by 5.2/5.9%. This confirms that a single model can be jointly trained across heterogeneous datasets, tasks, and skeleton topologies, with DrAction's format-agnostic rendering enabling complementary learning rather than interference.

**DrAction with External MLLMs** Since DrAction outputs standard RGB images, its learned rendering can be directly fed into any closed-source MLLM without modification. Table 6 evaluates this: (1) DrAction's rendering improves GPT-4o over fixed 3D+Velocity rendering (+7.78% on 48/12), confirming that the learned visual representation

*Table 6.* DrAction paired with external closed-source MLLMs on NTU-60 zero-shot recognition. Joint training remains essential.

| Method | 48/12 | 30/30 |
|---|---|---|
| GPT-4o + 3D+Velocity | 35.74 | 19.27 |
| GPT-4o + DrAction | 43.52 | 25.68 |
| GPT-5.4 + DrAction | 53.18 | 33.11 |
| **SkeletonLLM (joint-trained)** | **64.72** | **37.84** |

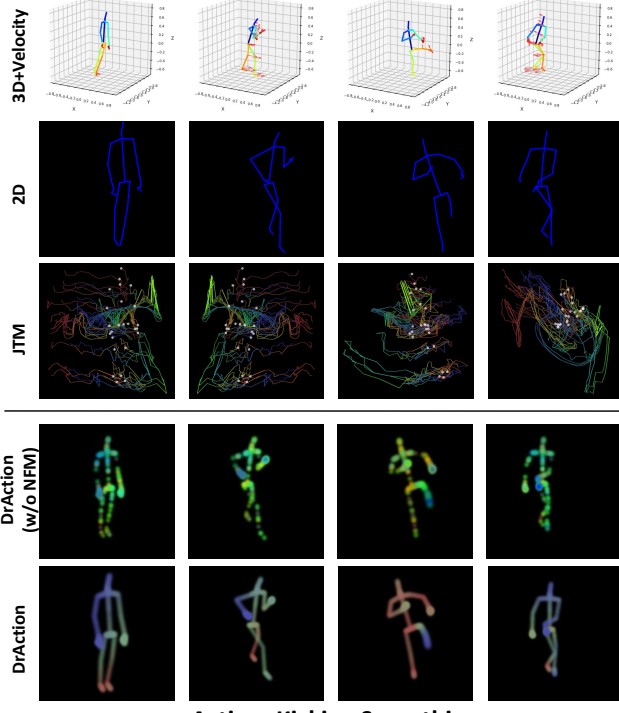

**Action: Kicking Something**

*Figure 4.* Qualitative comparison of rendering methods. Fixed renderers (3D+Velocity, 2D, JTM (Wang et al., 2016b)) produce visualizations that are either generic, information-poor, or perceptually complex. DrAction learns an abstract representation. With the NFM, it dynamically highlights kinematically salient regions (e.g., the kicking leg), producing a more informative visual language for the MLLM.

generalizes beyond the jointly trained backbone; (2) even the powerful GPT-5.4 + DrAction lags behind jointly trained SkeletonLLM by 11.54% on 48/12, demonstrating that joint end-to-end optimization is irreplaceable and achieves improvements beyond what current frontier LLMs can provide with rendering alone.

### 4.4. Ablation Study

We conduct comprehensive ablations to validate the design choices of SkeletonLLM.

**Learnable vs. Non-Learnable Renderers**   Table 7 compares DrAction against fixed, non-learnable renderers. Fixed renderers produce task-agnostic visualizations that may not emphasize kinematically critical regions. JTM's complex multi-view rendering actually hurts performance as the MLLM struggles to interpret its dense visual patterns. Learnable rendering (DrAction w/o NFM) improves over

*Table 7.* Ablation on rendering methods on NTU-60 & NTU-120. Our differentiable DrAction outperforms non-learnable renderers.

| Renderer | NTU-60 (Acc, %) | | | | NTU-120 (Acc, %) | | | |
|---|---|---|---|---|---|---|---|---|
| | 55/5 | 48/12 | 40/20 | 30/30 | 110/10 | 96/24 | 80/40 | 60/60 |
| 3D+Velocity | 81.32 | 58.77 | 37.83 | 31.45 | 70.48 | 60.60 | 37.74 | 28.59 |
| 2D Projection | 81.51 | 58.16 | 38.81 | 31.80 | 70.94 | 61.09 | 38.30 | 27.69 |
| JTM | 55.86 | 26.37 | 23.45 | 16.83 | 45.51 | 34.03 | 15.09 | 8.75 |
| Ours (w/o NFM) | 83.30 | 61.09 | 40.93 | 33.87 | 71.29 | 62.62 | 40.69 | 30.67 |
| **Ours (Full)** | **87.37** | **64.72** | **46.15** | **37.84** | **76.05** | **67.20** | **44.37** | **34.94** |

*Table 8.* Ablation on progressive training strategy on NTU-60 & NTU-120.

| Configuration | NTU-60 (Acc, %) | | | | NTU-120 (Acc, %) | | | |
|---|---|---|---|---|---|---|---|---|
| | 55/5 | 48/12 | 40/20 | 30/30 | 110/10 | 96/24 | 80/40 | 60/60 |
| w/o CR-Distill | 85.27 | 63.37 | 43.06 | 35.90 | 74.68 | 64.97 | 43.96 | 33.79 |
| w/o Disc-FT | 85.80 | 63.66 | 45.58 | 36.29 | 75.25 | 65.90 | 43.05 | 33.40 |
| w/o Both | 84.43 | 62.09 | 40.78 | 35.05 | 72.95 | 64.05 | 41.13 | 32.77 |
| Joint Training | 82.68 | 61.46 | 40.30 | 34.10 | 72.42 | 62.03 | 40.18 | 30.94 |
| **Full Pipeline** | **87.37** | **64.72** | **46.15** | **37.84** | **76.05** | **67.20** | **44.37** | **34.94** |

fixed methods by allowing gradients to shape the visual representation. The Neural Feature Modulator (NFM) provides an additional 3.63% boost on NTU-60 (48/12) by dynamically highlighting motion-salient regions (Figure 4).

**Progressive Training Strategy**   Table 8 validates our four-stage progressive training design.

**CR-Distill is critical for reasoning**: Removing it causes the largest drop on extreme splits (1.94% on 30/30), where causal understanding of motion dynamics matters most. Disc-FT sharpens decision boundaries: removing it particularly affects fine-grained discrimination, as seen from the 1.32% drop on 80/40. Progressive training prevents optimization interference: jointly training all components from scratch leads to gradient instability and suboptimal convergence, confirming the "chicken-and-egg" hypothesis discussed in Section 3.4.

## 5. Conclusion

We introduced SkeletonLLM, a framework that achieves universal skeleton understanding by translating heterogeneous skeleton formats into the MLLM's native visual modality. At its core is DrAction, a format-agnostic, differentiable renderer that converts skeletal kinematics into task-optimized image sequences, enabling end-to-end optimization guided by the MLLM's reasoning gradients. Complemented by a cooperative training strategy that instills causal reasoning and fine-grained discrimination, SkeletonLLM enables a single model to perform recognition, captioning, question answering, and cross-format transfer without architectural modifications. Our results demonstrate that visual translation offers a viable path for extending foundation model reasoning to structured, non-visual data modalities beyond standard images and text.

## Acknowledgements

This work was supported by National Natural Science Foundation of China (No. 62473007), Guangdong Outstanding Youth Fund (No. 2026B1515020015), Shenzhen Innovation in Science and Technology Foundation for The Excellent Youth Scholars (No. RCYX20231211090248064).

## Impact Statement

This paper presents work whose goal is to advance the field of Machine Learning. There are many potential societal consequences of our work, none which we feel must be specifically highlighted here.

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

# Appendix Contents

# A. Ablation Studies

This section presents comprehensive ablation studies to validate the design choices of SkeletonLLM, including the progressive training strategy, rendering methods, and NFM components.

## A.1. Progressive Training Strategy

As introduced in the main paper, we propose a progressive four-stage training strategy to address the joint optimization challenge between the differentiable renderer and the MLLM. This section provides a detailed illustration of the training pipeline, including the mathematical formulation of each stage's objective.

### A.1.1. PROGRESSIVE TRAINING OVERVIEW

Figure 5 illustrates the complete progressive training pipeline. To address the "chicken-and-egg" challenge—where the MLLM requires informative visual inputs to generate meaningful gradients, while the renderer needs such gradients to learn how to produce those visuals—we progressively activate and fine-tune model components in a curriculum manner.

### A.1.2. STAGE-WISE TRAINING OBJECTIVES

Each stage has a distinct objective, task formulation, and set of trainable parameters. We provide the detailed specification below.

### Stage 1: Render Warm-up.

- Objective: Establish a baseline visual representation that the frozen MLLM can interpret.
- Task: Multiple-choice question answering (MQA). Given rendered frames and a list of candidate action labels, the model selects the correct action.
- Trainable Parameters: Only DrAction renderer $\Theta_{\text{render}}$; the vision encoder, projector, and LLM remain frozen.
- Loss: $\mathcal{L}_{\text{MQA}} = -\log p(y^*|\mathbf{V}, \text{prompt})$, where $y^*$ is the ground-truth label token.

### Stage 2: Discriminative Finetuning (Disc-FT).

- Objective: Sharpen decision boundaries between visually similar actions.
- Task: Binary judgment on confusing action pairs. For a rendered clip, the model answers whether it depicts a specific (possibly incorrect) action: "Does this show 'rubbing hands'? Yes/No."
- Trainable Parameters: DrAction $\Theta_{\text{render}}$ and the projector MLP $\Theta_{\text{proj}}$.
- Loss: $\mathcal{L}_{\text{Disc}} = -\log p(y_{\text{binary}}|\mathbf{V}, \text{query})$, where $y_{\text{binary}} \in \{\text{Yes}, \text{No}\}$.

### Stage 3: Causal Reasoning Distillation (CR-Distill).

- Objective: Instill structured, step-by-step reasoning about body-part dynamics.
- Task: Knowledge distillation from a teacher model (GPT-4o). The teacher generates detailed causal rationales describing how the action unfolds temporally, grounded in specific body-part movements. The student learns to reproduce these rationales.
- Trainable Parameters: DrAction $\Theta_{\text{render}}$, projector $\Theta_{\text{proj}}$, and LLM via LoRA adapters $\Theta_{\text{LoRA}}$.
- Loss: $\mathcal{L}_{\text{CR}} = -\sum_t \log p(y_t^{\text{teacher}}|y_{<t}^{\text{teacher}}, \mathbf{V}, \text{prompt})$, the standard auto-regressive cross-entropy over the teacher's full token sequence.

The teacher rationale follows a structured format: (1) action summary, (2) step-by-step causal chain referencing body parts, (3) final label. This supervision teaches the model not just *what* action occurs, but *why* and *how* it unfolds.

### Stage 4: Recognition Refinement.

- Objective: Consolidate learned capabilities for peak classification accuracy.
- Task: Return to MQA, now with a mature renderer and reasoning-aware LLM.
- Trainable Parameters: With DrAction frozen, only $\Theta_{\text{proj}}$ and $\Theta_{\text{LoRA}}$ are updated.
- Loss: $\mathcal{L}_{\text{MQA}}$ as in Stage 1.

### A.1.3. RATIONALE FOR PROGRESSIVE TRAINING

Our ablations confirm that this curriculum is essential for stable optimization. Joint training from scratch leads to gradient instability and suboptimal convergence due to the warm-start problem. The progressive schedule ensures:

1. The renderer first learns to produce coherent visuals before receiving complex gradients from language generation.
2. Discriminative training precedes generative reasoning to establish robust feature boundaries.
3. The final stage consolidates all learned capabilities without disrupting the renderer's learned representations.

### A.1.4. SUMMARY OF COMPONENT CONTRIBUTIONS

Table 9 summarizes the role of each component in addressing the challenges of skeleton-MLLM integration.

### A.1.5. QUANTITATIVE ABLATION ON PROGRESSIVE TRAINING

We conduct a comprehensive ablation study to validate the design of our four-stage progressive training curriculum. Specifically, we evaluate several variants against the full model:

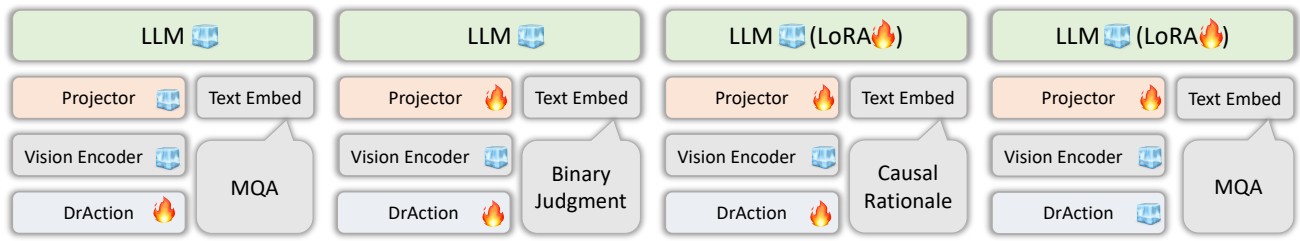

**(a) Stage1: Render Warm-up**  **(b) Stage2: Discriminative Finetuning**  **(c) Stage3: Causal Reasoning Distillation** **(d) Stage4: Recognize Refinement**

*Figure 5.* Our Progressive Training Pipeline. To address the joint optimization challenge, we progressively activate and fine-tune model components. The training curriculum begins with (a) warming up the renderer to generate intelligible visuals and concludes with (d) refining recognition, both utilizing a multiple-choice question & answer (MQA) task. In between, the strategy incorporates (b) learning discriminative features via a binary judgment task and (c) instilling causal reasoning through knowledge distillation from a teacher model.

*Table 9.* Summary of SkeletonLLM components and their contributions.

| Component | Addresses | Key Mechanism |
|---|---|---|
| DrAction | Modality gap, format heterogeneity | Differentiable rendering with LBS |
| NFM | Information preservation | Pose-conditioned appearance |
| Cooperative Training | Optimization stability, reasoning depth | Progressive curriculum + distillation |

- Variant A: Removes Stage 3 (CR-Distill), training without causal reasoning distillation.

- Variant B: Removes Stage 2 (Disc-FT), training without discriminative finetuning.

- Variant C: Removes both Stage 2 and Stage 3, relying only on render warm-up and recognition refinement.

- Variant D: Trains all components jointly from the beginning without a progressive schedule.

The quantitative results are presented in Table 8 (main paper). The ablation clearly demonstrates the effectiveness of our progressive training strategy across all evaluation splits on both NTU-60 and NTU-120 datasets. The four variants correspond to: Variant A (w/o CR-Distill), Variant B (w/o Disc-FT), Variant C (w/o Both), and Variant D (Joint Training).

**Analysis of Results.**  The results lead to several important observations:

(1) Both CR-Distill and Disc-FT are critical. Removing either CR-Distill (Variant A) or Disc-FT (Variant B) leads to a significant performance drop across all splits. For instance, on the challenging 30/30 split of NTU-60, removing CR-Distill causes a 1.94% accuracy drop (from 37.84% to 35.90%), while removing Disc-FT results in a 1.55% drop (to 36.29%). This confirms that both stages contribute essential and complementary capabilities to the model.

(2) CR-Distill and Disc-FT serve complementary roles. CR-Distill is vital for instilling a deep, causal understanding of complex temporal actions by teaching the model to generate step-by-step reasoning chains that describe body-part dynamics. Disc-FT, on the other hand, is essential for sharpening decision boundaries between visually similar categories by training the model to distinguish confusable action pairs (e.g., "clapping" vs. "rubbing hands").

(3) The synergistic effect of combining both stages. Variant C, which omits both CR-Distill and Disc-FT, suffers a more substantial degradation than either Variant A or B alone. For example, on the 80/40 split of NTU-120, Variant C achieves only 41.13%, compared to 43.96% (Variant A) and 43.05% (Variant B). This highlights the synergistic and complementary nature of these two training phases.

(4) Progressive training is essential for stable optimization. Variant D, which trains all components jointly from the beginning without a progressive schedule, consistently underperforms compared to the full model across all splits. This confirms our hypothesis that simultaneous end-to-end training is unstable due to the "chicken-and-egg" problem. The progressive strategy is crucial for mitigating optimization interference, allowing the renderer to first learn a stable visual representation before the LLM is fine-tuned for complex reasoning tasks. The performance gap is particularly pronounced on more challenging splits with fewer seen classes (e.g., NTU-60 30/30 and NTU-120 60/60), where stable optimization becomes even more critical.

### A.2. Ablation on Rendering Methods

Table 7 (main paper) compares DrAction against fixed, non-learnable renderers. Fixed renderers produce task-agnostic

*Table 10.* Ablation on NFM components and temporal modeling strategies on NTU-60 & NTU-120. We compare different temporal fusion methods (GRU, LSTM, RNN) within the NFM. The full NFM with GRU achieves the best performance across nearly all splits.

| Configuration | NTU-60 (Acc, %) | | | | NTU-120 (Acc, %) | | | |
|---|---|---|---|---|---|---|---|---|
| | 55/5 | 48/12 | 40/20 | 30/30 | 110/10 | 96/24 | 80/40 | 60/60 |
| w/o NFM | 83.30 | 61.09 | 40.93 | 33.87 | 71.29 | 62.62 | 40.69 | 30.67 |
| w/o Temporal | 86.35 | 63.56 | 45.62 | 36.94 | 74.80 | 65.98 | 43.01 | 33.45 |
| w/ RNN | 86.33 | 64.45 | 45.99 | 37.18 | 75.97 | 66.89 | 43.17 | 34.36 |
| w/ LSTM | 86.65 | 64.95 | 45.41 | 37.62 | 75.72 | 67.43 | 43.54 | 34.27 |
| w/ GRU (Full) | 87.37 | 64.72 | 46.15 | 37.84 | 76.05 | 67.20 | 44.37 | 34.94 |

visualizations that may not emphasize kinematically critical regions. JTM's complex multi-view rendering actually hurts performance as the MLLM struggles to interpret its dense visual patterns. Learnable rendering (DrAction w/o NFM) improves over fixed methods by allowing gradients to shape the visual representation. The Neural Feature Modulator (NFM) provides an additional boost by dynamically highlighting motion-salient regions.

Figure 6 provides qualitative comparisons, contrasting our learnable DrAction renderer against fixed rendering pipelines on representative actions. The figure, which organizes actions by column and rendering schemes by row, shows that static methods like 3D+Velocity, 2D projection, and JTM often miss or clutter fine-grained motion cues. In contrast, DrAction, particularly when enhanced by the Neural Feature Modulator (NFM), highlights motion-critical joints using adaptive color and opacity.

### A.3. Ablation on NFM Components and Temporal Modeling

To validate the contribution of each NFM component, we conduct ablation experiments on NTU-60 and NTU-120. As shown in Table 10, removing the NFM entirely ("w/o NFM") causes a substantial performance drop across all splits, confirming that pose-conditioned appearance modulation is essential for encoding motion-salient cues.

We also compare different temporal modeling strategies within the NFM, including GRU, LSTM (Hochreiter & Schmidhuber, 1997), and vanilla RNN (Elman, 1990). The GRU component captures temporal dependencies within the feature sequence, enabling the renderer to emphasize phase-specific motion cues (e.g., the acceleration phase of a kick versus the deceleration phase). Among the temporal modeling alternatives, GRU achieves the best performance across most splits, outperforming LSTM by 0.22–0.83% and vanilla RNN by 0.08–1.20% on splits where GRU leads. LSTM shows slight advantages on two splits (48/12 on NTU-60 and 96/24 on NTU-120), but introduces additional computational overhead due to its gating mechanism. Vanilla

RNN, lacking gating mechanisms, shows competitive results on certain splits but suffers from gradient vanishing issues during training and underperforms on splits requiring longer temporal reasoning. The lightweight GRU strikes the optimal balance between temporal modeling capacity and gradient stability for our per-primitive feature modulation task.

## B. Additional Experiments

This section provides additional experimental results, including evaluation on the PKU-MMD dataset, parameter sensitivity analysis on rendered frame count and input resolution, and detailed error analysis with causal reasoning comparisons.

### B.1. Results on PKU-MMD

To further assess the generalization of our approach, we evaluate on the PKU-MMD dataset (Liu et al., 2017), a large-scale multi-modal benchmark with 51 action classes. As shown in Table 11, for a fair comparison we follow the experimental protocol of Chen et al. (2024a), adopting its 46/5 and 39/12 seen/unseen class splits under both the cross-subject (Xsub) and cross-view (Xview) evaluation settings. The results show that SkeletonLLM outperforms all prior methods across both splits and evaluation settings, demonstrating robustness to subject variation and viewpoint changes. Notably, SkeletonLLM also outperforms LLM-based motion understanding methods (MotionGPT, MotionLLM), further validating the effectiveness of our visual translation paradigm over tokenization-based approaches.

### B.2. Ablation on Rendered Frame Count

We investigate the impact of temporal coverage by varying the rendered frame count $N_{\text{frames}}$ from 4 to 16. As illustrated in Figure 7, recognition accuracy consistently improves as more frames are included, since longer sequences capture richer motion dynamics and reduce temporal ambiguity. The performance gain saturates around 12 frames, indicating that

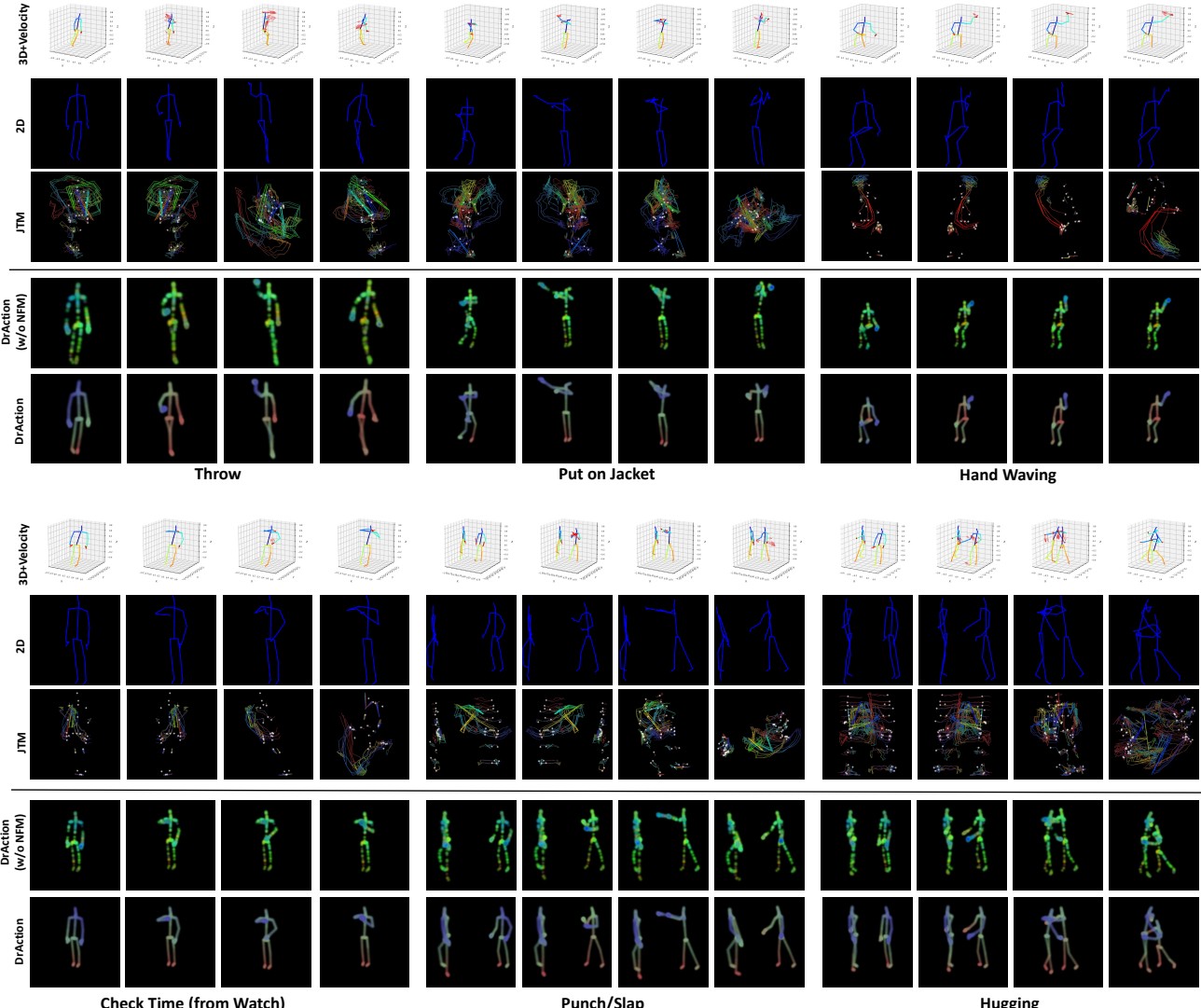

*Figure 6.* Qualitative comparison of different rendering methods. Each column shows a representative action instance. From top to bottom, we compare three fixed renderers (3D+Velocity, 2D projection, and JTM (Wang et al., 2016b)) with our learnable DrAction variants (w/o NFM and full DrAction). Static methods often miss or clutter fine-grained motion cues. In contrast, DrAction, particularly when enhanced by the Neural Feature Modulator (NFM), highlights motion-critical joints using adaptive color and opacity. These visualizations substantiate our claim that a task-optimized, differentiable renderer yields a more effective visual language for MLLMs than static, hand-designed representations. Additional rendered videos are available in the code repository: https://github.com/wangzy01/SkeletonLLM.

*Table 11.* Top-1 accuracy (%) on the PKU-MMD dataset under cross-subject (Xsub) and cross-view (Xview) settings. We follow the splits from Chen et al. (2024a). The best results are highlighted in red. $^\dagger$Results for Qwen2.5-VL-7B and InternVL3-8B were obtained by rendering skeletons with fixed visualization and finetuning on MQA for 6 epochs.

| Methods | Type | Venue | Xsub (%) | | Xview (%) | |
|---|---|---|---|---|---|---|
| | | | 46/5 | 39/12 | 46/5 | 39/12 |
| ReViSE (Hubert Tsai et al., 2017) | Align | ICCV'17 | 54.2 | 19.3 | 54.1 | 12.7 |
| JPoSE (Wray et al., 2019) | Align | ICCV'19 | 57.4 | 27.0 | 53.1 | 22.8 |
| CADA-VAE (Schonfeld et al., 2019) | Align | CVPRW'19 | 73.9 | 33.7 | 74.5 | 29.5 |
| SynSE (Gupta et al., 2021) | Align | ICIP'21 | 69.5 | 36.5 | 71.7 | 25.4 |
| SMIE (Zhou et al., 2023) | Align | ACM MM'23 | 72.9 | 44.2 | 71.6 | 40.7 |
| STAR (Chen et al., 2024b) | Align | ACM MM'24 | 76.3 | 50.2 | 75.4 | 50.5 |
| Neuron (Chen et al., 2024a) | Align | CVPR'25 | 89.2 | 61.4 | 88.2 | 62.2 |
| MotionGPT (Jiang et al., 2023) | LLM | NeurIPS'23 | 34.7 | 16.0 | 32.1 | 10.6 |
| MotionLLM (Chen et al., 2025) | LLM | TPAMI'25 | 46.4 | 27.8 | 49.4 | 20.9 |
| Qwen2.5-VL-7B$^\dagger$ (Bai et al., 2025) | MLLM | - | 80.9 | 57.3 | 83.8 | 60.8 |
| InternVL3-8B$^\dagger$ (Zhu et al., 2025b) | MLLM | - | 83.4 | 58.2 | 85.0 | 60.6 |
| SkeletonLLM (Ours) | MLLM | - | 90.1 | 63.9 | 89.5 | 64.2 |

most key action phases essential for distinguishing complex actions are sufficiently covered. Consequently, we adopt $N_{\text{frames}} = 12$ as the optimal trade-off between recognition performance and computational efficiency.

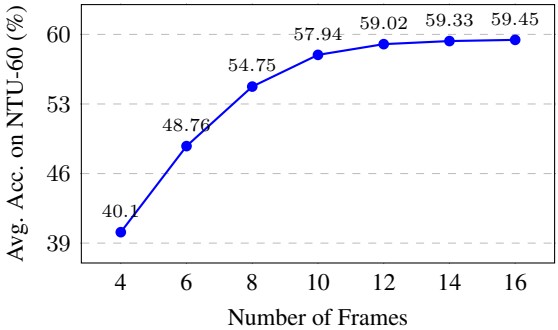

*Figure 7.* Impact of rendered frame count. Accuracy on NTU-60 increases sharply from 4 to 10 frames and saturates at 12 frames (59.02%). Increasing to 16 frames yields minimal gains (59.45%), making 12 frames the optimal trade-off.

**B.3. Ablation on Input Resolution**

We evaluate the impact of rendering resolution ($H \times W$) on recognition performance. As shown in Table 12, increasing the resolution from $224 \times 224$ to $448 \times 448$ yields a substantial accuracy boost (from 54.24% to 59.02%). This significant gain is largely attributed to $448 \times 448$ being the native input resolution of the InternVL3 backbone, which allows the model to extract visual features optimally without resizing artifacts. Although scaling further to $672 \times 672$ provides a marginal improvement (59.98%) due to finer rasterization details, it incurs disproportionately higher memory and computational costs. We therefore standardize on $448 \times 448$ to achieve the best balance between accuracy and

efficiency for all subsequent experiments.

*Table 12.* Impact of input resolution. Average Top-1 accuracy on NTU-60 increases with resolution, with $448 \times 448$ offering the best trade-off.

| Resolution | $224 \times 224$ | $448 \times 448$ | $672 \times 672$ |
|---|---|---|---|
| Top-1 (%) | 54.24 | 59.02 | 59.98 |

**B.4. Error Distribution and Causal Reasoning Analysis**

We provide a qualitative comparison to investigate the source of SkeletonLLM's performance gains on the NTU-60 (48/12 split). Figure 8 visualizes the confusion matrices of the baseline InternVL3 (using fixed rendering and MQA finetuning) versus our SkeletonLLM, while Figure 9 contrasts the step-by-step reasoning processes of SkeletonLLM w/o Disc-FT & CR-Distill and the full SkeletonLLM on a complex action instance.

As shown in Figure 8 (Left), the baseline InternVL3 exhibits significant confusion among visually similar actions. Specifically, within the highlighted yellow regions, distinctive actions like "put on shoe" (class 16) are frequently misclassified as "pick up" (class 6) or "falling down" (class 43), owing to their shared bending postures. Similarly, subtle physiological motions like "sneezing/cough" (Figure 8, class 41) and "nausea/vomiting" (class 48) are heavily entangled. In contrast, SkeletonLLM (Right) markedly suppresses these off-diagonal errors. The accuracy for "put on shoe" surges from 50% to 82%, and the separation between "sneezing" and "nausea" is substantially clearer (highlighted in red boxes). This improvement validates the effectiveness of our Disc-FT. By explicitly training the model to answer binary "Yes/No" questions on these confusable pairs, Disc-

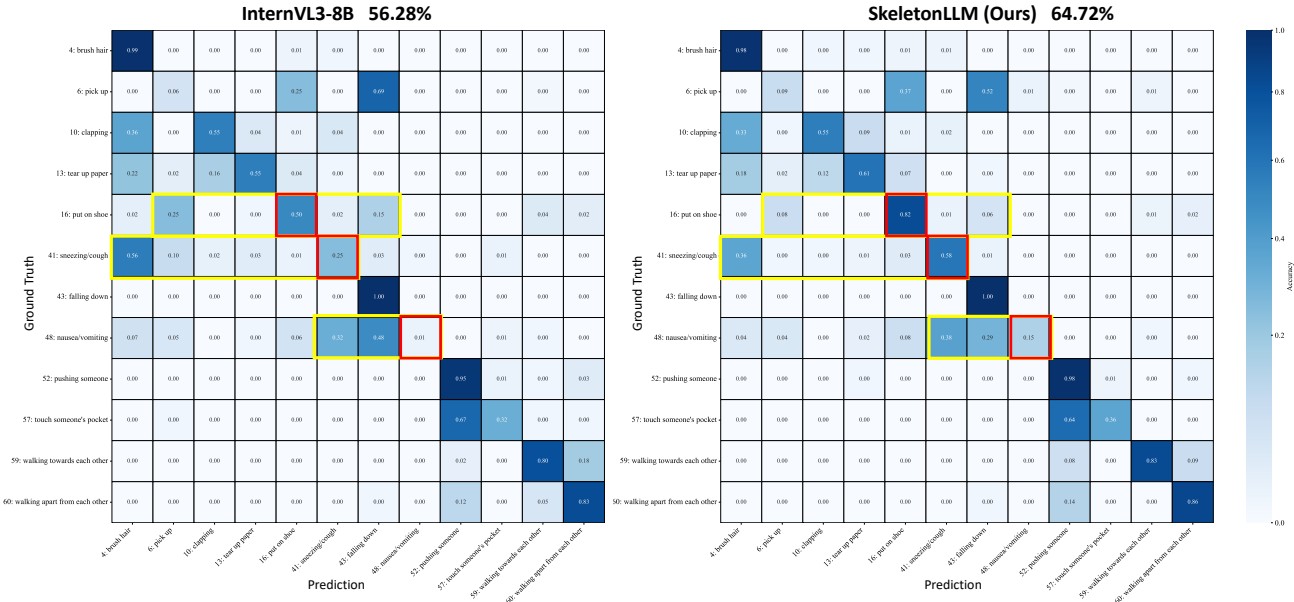

*Figure 8.* Confusion matrix comparison on the NTU-60 (48/12 split). Left: InternVL3 baseline. Right: SkeletonLLM (Ours). Our method significantly reduces confusion between visually similar actions (highlighted in yellow) and improves accuracy on fine-grained classes (red boxes), demonstrating the effectiveness of Discriminative Finetuning.

FT enforces sharper decision boundaries, preventing the model from relying on superficial pose similarities.

Figure 9 further illustrates how the models arrive at their decisions. In the case of "Put on shoe", SkeletonLLM w/o Disc-FT & CR-Distill correctly identifies the "bending" posture but fails to discern the fine-grained interaction, hallucinating a "Pick up" action. Conversely, the full Skeleton-LLM generates a coherent causal chain (Wei et al., 2022). Although both models employ DrAction, the full model, reinforced by CR-Distill, accurately interprets the visual details, noting that the "hands move towards the feet" and the leg is raised to "bring the foot closer to the hands" (highlighted in red). Furthermore, driven by this causal reasoning, the model demonstrates robust logic: it explicitly rules out multi-person interactions and distinguishes the hand-to-foot motion from object retrieval (highlighted in purple). This confirms that our cooperative training strategy does not merely fit labels but equips the MLLM with the ability to ground its reasoning in specific, causal kinematic evidence.

### B.5. Computational Cost Analysis

To better reflect practical deployment scenarios, we conduct a comprehensive efficiency evaluation of SkeletonLLM on a single consumer-grade NVIDIA RTX 4090 GPU (24GB). All measurements use batch size = 1, with DrAction rendering in FP32 full precision and the MLLM in BF16 mixed precision. We note that the current implementation is an engineering development version without inference-specific optimizations.

**DrAction Rendering Efficiency.** DrAction is built on 3D Gaussian Splatting and benefits from its efficient differentiable rasterization. Table 13 reports rendering latency under different configurations.

*Table 13.* DrAction rendering efficiency at different resolutions.

| Configuration | Per-frame (ms) | 12 frames (ms) | Memory (GB) |
|---|---|---|---|
| $448 \times 448$, FP32 | 1.7 | 20.4 | 0.5 |
| $224 \times 224$, FP32 | 1.6 | 19.3 | 0.5 |

DrAction incurs minimal rendering overhead: rendering 12 frames at $448 \times 448$ takes only ∼20ms. Notably, the per-frame latency is nearly identical across resolutions (1.6ms vs. 1.7ms) because 3DGS rendering complexity is dominated by the number of Gaussian primitives rather than output resolution. In our implementation, each skeleton uses a fixed number of primitives ($\sim J + 10 \times |\mathcal{E}|$, where $J$ is joint count and $|\mathcal{E}|$ is the number of bone edges), resulting in stable rendering time regardless of resolution.

**End-to-End Inference Performance.** Table 14 breaks down the complete inference pipeline (skeleton input → DrAction rendering → MLLM forward → text output). The end-to-end latency of ∼192ms corresponds to a throughput of ∼5.2 samples/sec, meeting real-time interaction requirements (<200ms response latency is generally considered the threshold for fluid interaction). The inference bottleneck lies in LLM autoregressive generation (∼65%), consistent with most MLLM systems where each token requires a full decoder forward pass. The Vision Encoder accounts for

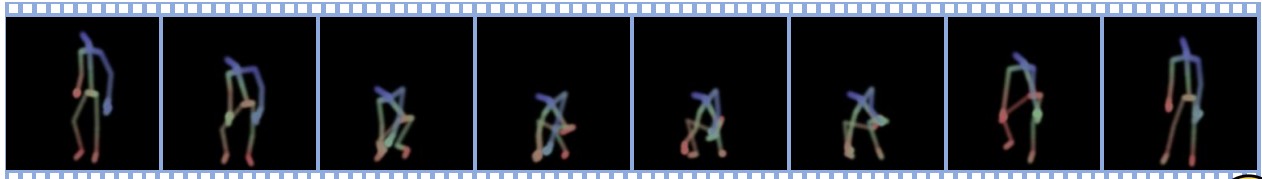

*Figure 9.* Comparison of reasoning processes on the NTU-60 (48/12 split). We visualize the chain-of-thought generated by SkeletonLLM w/o Disc-FT & CR-Distill (top) and the full SkeletonLLM (bottom) for a "Put on shoe" sequence. While the ablated variant hallucinates a "Pick up" action based on a coarse bending posture, the full SkeletonLLM accurately identifies fine-grained hand-foot interactions (red text) and employs causal logic (purple text) to rule out distractors.

*Table 14.* End-to-end inference latency breakdown.

| Stage | Latency (ms) | Proportion |
|---|---|---|
| DrAction rendering (12 frames) | 20.4 | 10.6% |
| Vision Encoder | 46.2 | 24.1% |
| LLM generation (avg. 16 tokens) | 125.0 | 65.3% |
| End-to-end total | 191.6 | 100% |

~24%, processing 12 frames of $448 \times 448$ images (~3072 visual tokens). DrAction rendering contributes only ~11% of total compute, indicating that the cost of "translating" skeletons to the visual modality is acceptable and does not become a deployment bottleneck.

**Memory Footprint Analysis.** Table 15 details the memory consumption of each component.

*Table 15.* GPU memory breakdown during inference.

| Component | Memory (GB) | Note |
|---|---|---|
| InternVL3-8B (BF16) | 13.5 | Model weights |
| LoRA Adapters | 0.4 | Finetuning params |
| DrAction (Gaussians + NFM) | 0.5 | Renderer params |
| Intermediate activations (12 frames) | 1.1 | KV Cache + activations |
| Peak inference total | 15.5 | — |

SkeletonLLM consumes ~15.5GB (65%) on RTX 4090, leaving ample headroom for larger batch sizes or longer sequences. DrAction occupies only 0.5GB thanks to 3DGS's compact representation. Importantly, DrAction is a lightweight, modular translation module whose functionality is independent of the backend MLLM's scale. This

modular design enables flexible combinations: pairing with 1B/2B lightweight MLLMs for low-cost edge deployment, or interfacing with 70B+ models to fully leverage reasoning capacity, providing broad adaptability across different compute platforms.

**Optimization Potential.** The reported performance is based on an unoptimized engineering version. In production deployments, mature techniques such as INT8/INT4 weight quantization, KV Cache compression, and speculative decoding can further improve efficiency. With such optimizations, end-to-end latency could potentially drop below 100ms and memory usage to 8–10GB, enabling deployment on a wider range of consumer GPUs.

### B.6. Feature Space Analysis

We visualize the feature space using t-SNE in Figure 10. The features are extracted from the hidden states of the MLLM's final Transformer decoder block (before the final normalization and linear head). The visualization corresponds to the Xsub 48/12 open-vocabulary split on NTU-60.

In the plot, the lighter background points represent samples from the 48 seen classes used during training, while the bold, darker points represent the 12 unseen classes encountered only during testing. As shown, even though the unseen actions were never part of the training set, their samples spontaneously organize into compact, well-separated clusters rather than being scattered randomly. This indicates that SkeletonLLM generalizes well, mapping novel motion

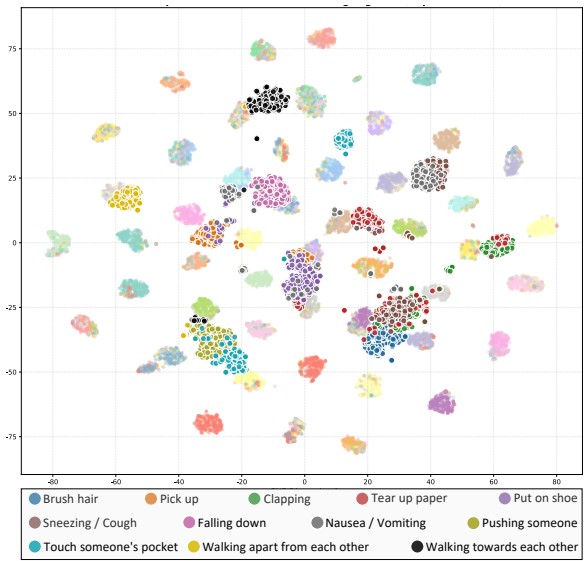

*Figure 10.* t-SNE visualization of feature representations on NTU-60 (48/12 split). The visualization is derived from the 48/12 open-vocabulary split (Xsub benchmark). Points in faded, lighter colors represent the 48 seen classes, while the darker, highlighted clusters correspond to the 12 unseen classes. Even though the unseen actions were never part of the training set, their samples spontaneously organize into compact, well-separated clusters.

patterns to distinct semantic regions. Furthermore, visually or semantically similar actions, such as "Pushing someone" and "Touch someone's pocket", maintain clear separation with distinct boundaries. This suggests that the learned feature space preserves discriminative structure, aiding in the distinction of fine-grained differences among unseen categories, which is essential for robust open-vocabulary recognition.

## C. Cross-Format Generalization

This section provides additional details on the cross-format transfer experiments presented in the main paper, along with a deeper exposition of DrAction's format-agnostic design principles.

### C.1. Skeleton Format Differences

*Table 16.* Comparison of skeleton formats used in cross-format experiments.

| Dataset | Joints | Acquisition | Topology |
| --- | --- | --- | --- |
| NTU-60/NTU-120 | 25 | Kinect v2 | Kinect skeleton |
| NTU-60 (2D) | 17 | 2D Pose Est. | COCO keypoints |
| NW-UCLA | 20 | Kinect v1 | Kinect skeleton |
| HumanML3D | 22 | MoCap | SMPL model |
| PKU-MMD | 25 | Kinect v2 | Kinect skeleton |

Table 16 summarizes the skeleton formats used in our experiments. The key differences include:

- Joint count: Varies from 17 (NTU-60 2D) to 25 (NTU,

PKU-MMD), with NW-UCLA at 20 and HumanML3D at 22.
- Joint definitions: Kinect and SMPL define joints differently. For example, SMPL includes pelvis as a separate joint while Kinect uses spine base. 2D poses from HRNet follow the COCO keypoint format.
- Coordinate systems: Different acquisition systems use different coordinate conventions. Notably, NTU-60 (2D) lacks depth information entirely.

### C.2. DrAction's Format-Agnostic Design

A critical property of DrAction is its ability to render any skeleton format into a visually consistent representation. This format agnosticism is achieved through three design principles:

**Topology-Independent Gaussian Initialization.** Unlike methods that assume a fixed skeleton structure, DrAction dynamically instantiates Gaussian primitives based on the input skeleton's adjacency structure. The total number of Gaussians adapts automatically:

$$K = J + |\mathcal{E}| \cdot N_{\text{samples}} \tag{9}$$

where $J$ is the number of joints, $|\mathcal{E}|$ is the number of bone edges (defined by the skeleton's connectivity), and $N_{\text{samples}}$ is the number of Gaussians sampled per bone (we use $N_{\text{samples}} = 10$). Joint Gaussians are centered at joint positions, while bone Gaussians are uniformly interpolated along each edge. This ensures that any skeleton—regardless of its joint count or topology—receives appropriate visual coverage.

**Adaptive LBS Weight Computation.** Linear Blend Skinning weights $\mathbf{w}_k \in \Delta^{J-1}$ are computed based on the input skeleton's connectivity rather than a fixed template:

- Joint Gaussians: Receive concentrated weights via one-hot encoding in logit space, i.e., $\text{logit}_{k,j} = +10$ for the associated joint $j$ and $-10$ otherwise.
- Bone Gaussians: For a Gaussian at interpolation factor $\alpha$ between joints $a$ and $b$, the logits are set as $\text{logit}_{k,a} = \log(1-\alpha)+10$ and $\text{logit}_{k,b} = \log(\alpha)+10$, with all other entries at $-10$.

After softmax normalization, these weights ensure that each Gaussian is primarily influenced by its parent joints, preserving kinematic validity across different skeleton formats.

**Invariant Visual Rendering.** Regardless of the underlying joint count or topology, DrAction produces image sequences depicting human figures in a consistent visual style. This is achieved by:

1. Rendering all skeletons through the same differentiable rasterization pipeline with identical camera parameters.

2. Using the Neural Feature Modulator (NFM) to produce task-optimized appearances that emphasize motion-salient regions rather than skeleton-specific details.

3. Blending learned colors with depth-based visualization to maintain spatial coherence across formats.

This creates a *visual lingua franca* that the MLLM can interpret uniformly, enabling seamless cross-format transfer: a model trained on Kinect skeletons (25 joints) can directly process MoCap data (22 joints) or 2D pose estimations (17 joints) without any architectural changes or retraining.

**NFM Behavior Across Formats.** Figure 11 further visualizes how the Neural Feature Modulator behaves under topology changes. The same learned renderer weights are applied without target-format finetuning, while only the input graph and joint count change. The resulting heatmaps remain concentrated around anatomically and functionally similar regions across formats, indicating that NFM's appearance residuals are driven by local kinematic cues aggregated through sparse topology-based LBS weights. This provides qualitative evidence complementary to the cross-format recognition and captioning results in the main paper.

### C.3. Baseline Adaptation for Cross-Format Experiments

For fair comparison, we adapted baseline methods as follows:

MotionGPT: This method expects a fixed 263-dimensional motion representation based on the HumanML3D format. For cross-format experiments:

- We convert source skeletons to the 263-dim representation using joint position, velocity, and rotation features.
- Missing joints are interpolated from neighboring joints when possible.

SKI-LVLM/TDSM: Following Wang et al. (2024), we use zero-padding to unify skeleton formats:

- Skeletons are padded to the maximum joint count (25).
- Missing joints are filled with zeros.
- Models are retrained from scratch on the padded format.

## D. Motion Question Answering

Beyond classification and captioning, we investigate whether SkeletonLLM can perform complex reasoning about motion through natural language question answering. This section presents the Motion QA experiments: we first introduce the Skeleton-QA benchmark and describe its construction, then present the evaluation protocol, and finally report quantitative results followed by qualitative analysis.

### D.1. Skeleton-QA Benchmark

We construct a Motion QA test set based on 200 carefully selected samples from NTU-120, covering four types of reasoning:

- Temporal reasoning: Understanding sequential structure of actions (e.g., "What is the first step of this action?")
- Causal reasoning: Inferring intentions and consequences (e.g., "Why might this person be performing this action?")
- Fine-grained understanding: Identifying specific body part movements (e.g., "Which body parts are primarily involved?")
- Contrastive judgment: Distinguishing similar actions (e.g., "Is this 'clapping' or 'rubbing hands'? Explain why.")

Each question has a human-annotated reference answer. We evaluate both MQA (accuracy) and open-ended QA (ROUGE-L, BertScore).

### D.2. Benchmark Construction Details

The Skeleton-QA benchmark was constructed through the following process:

Sample Selection: We selected 200 samples from and NTU-120, ensuring diversity in:

- Action types (single-person, two-person interactions)
- Motion complexity (simple gestures to compound actions)
- Temporal length (short clips to long sequences)

Question Generation: For each sample, we generated questions across four reasoning types:

- Temporal (1-2 questions): Focus on sequential structure and phase transitions.
- Causal (1-2 questions): Focus on intentions, goals, and consequences.
- Fine-grained (1-2 questions): Focus on specific body part movements.
- Contrastive (1 question): Present similar action pairs for discrimination.

Table 18 provides example question templates for each reasoning type.

### D.3. Evaluation Protocol

MQA Evaluation: For multiple-choice questions, we provide 4 options (1 correct, 3 distractors). Distractors are selected based on:

- Semantic similarity (e.g., "clapping" vs. "rubbing hands")
- Shared body parts (e.g., both involving arm movements)
- Visual similarity in rendered form

Open-ended Evaluation: For free-form answers, we compute:

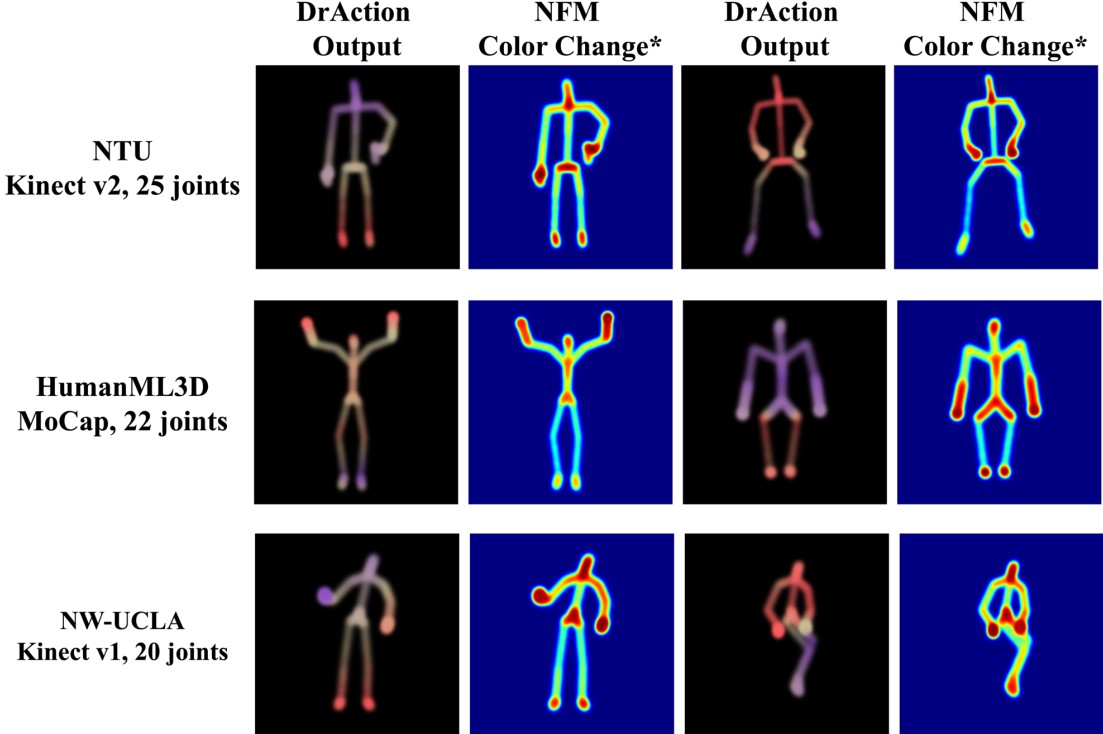

|  | **DrAction Output** | **NFM Color Change*** | **DrAction Output** | **NFM Color Change*** |

**NTU Kinect v2, 25 joints**

**HumanML3D MoCap, 22 joints**

**NW-UCLA Kinect v1, 20 joints**

**\*Not classifier saliency or attention; it shows the magnitude of NFM-induced RGB changes during rendering.**

*Figure 11.* NFM color modulation transfers across skeleton formats. We use the same NTU-60-trained DrAction renderer to process examples from three heterogeneous skeleton formats: NTU-60/Kinect v2 (25 joints), HumanML3D/SMPL (22 joints), and NW-UCLA/Kinect v1 (20 joints). Each pair shows the rendered skeleton and the corresponding heatmap of the NFM-induced RGB modification magnitude, where warmer colors indicate stronger appearance updates. Although the joint sets and graph topologies differ, NFM consistently emphasizes semantically corresponding motion-functional regions such as the arms, head-neck area, pelvis, and feet. This suggests that NFM learns local body-part saliency patterns rather than memorizing format-specific joint indices, supporting DrAction's cross-format generalization.

*Table 17.* Motion QA performance on the Skeleton-QA benchmark. Temporal, Causal, Fine-grained, and Contrastive columns report MQA accuracy (%).

| Method | Temporal | Causal | Fine-grained | Contrastive | ROUGE-L | BertScore |
|---|---|---|---|---|---|---|
| MotionGPT | 31 | 26 | 33 | 40 | 22 | 34 |
| MotionLLM | 37 | 33 | 39 | 43 | 27 | 39 |
| SKI-LVLM | 29 | 25 | 32 | 38 | 21 | 35 |
| InternVL3-8B | 53 | 48 | 55 | 59 | 39 | 49 |
| Ours (w/o CR) | 55 | 52 | 59 | 62 | 43 | 51 |
| Ours (Full) | 68 | 65 | 73 | 76 | 52 | 57 |

*Table 18.* Example question templates for each reasoning type in Skeleton-QA. Templates shown are simplified illustrations; actual prompts include detailed instructions.

| Type | Example Templates |
|---|---|
| MQA | "Analyze the action sequence. Select the best match from options below. Describe your reasoning, then output the final result. A. [action1] B. [action2] ..." |
| Temporal | "What is the first/last step of this action?" "In what order do the body parts move?" |
| Causal | "Why might this person be performing this action?" "What could be the goal of this movement?" |
| Fine-grained | "Which body parts are primarily involved?" "How do the left and right hands move differently?" |
| Contrastive | "Is this 'action A' or 'action B'? Explain why." "What distinguishes this from [similar action]?" |

- ROUGE-L: Measures longest common subsequence overlap with reference.
- BertScore: Measures semantic similarity using BERT embeddings.

### D.4. Quantitative Results

As shown in Table 17, SkeletonLLM substantially outperforms all baselines across all reasoning types. Based on the rounded scores, the full model improves over the best baseline (InternVL3-8B) by 15 points on temporal reasoning and 17 points on contrastive judgment. The gap between SkeletonLLM (Full) and the ablated version without CR-Distill is 13 points on both temporal and causal reasoning, demonstrating that CR-Distill is critical for instilling structured reasoning capabilities.

### D.5. Qualitative Analysis

Figure 12 illustrates the reasoning processes of different models on a "headache" sequence. Given a multiple-choice question asking to identify the action with reasoning, MotionGPT captures only a coarse pattern ("raises their arm and touches the upper body"), failing to distinguish fine-grained motion details and incorrectly selecting "neck pain." InternVL3 with fixed rendering produces a more detailed description, noting the arm movement to the head region with the "hand approximately at forehead or temple height," but critically misinterprets the gesture as a "salute"—it emphasizes that the upper body remains "relatively upright and stable" and describes the "hand held at brow level" as matching a "formal gesture pattern," missing the subtle pressing and head-drooping cues. SkeletonLLM without CR-Distill exhibits similar behavior: despite using DrAction's learnable rendering, it still misclassifies the action as "salute," describing a "formal gesture pattern" with an "upright torso." This reveals that without causal reasoning supervision, even improved visual representations are insufficient for fine-grained discrimination. In contrast, the full SkeletonLLM correctly identifies "headache" by gen-erating a coherent causal chain: it recognizes that the hand is in a "pressing posture rather than a flat salute position," notes the "head is slightly drooped with subtle swaying motion," and systematically rules out alternatives—"neck pain typically involves hands touching the back or side of the neck"; "salute is a standardized gesture with the hand raised to the brow in a flat, non-contact position." This comparison demonstrates that CR-Distill is essential for instilling structured causal reasoning, enabling the model to ground its decisions in specific kinematic evidence rather than superficial pose patterns.

## E. Implementation Details

This section provides detailed implementation information for reproducibility.

### E.1. Training Setup

Our SkeletonLLM is built upon InternVL3-8B (Zhu et al., 2025b) and trained on 2 NVIDIA H20 GPUs. For skeleton rendering, each joint and bone segment is modeled as a set of 3D Gaussians (Kerbl et al., 2023), where segments are constructed by uniformly sampling 10 intermediate points along the line connecting each joint pair. To balance temporal coverage and GPU memory usage, we sample 12 frames per sequence using a uniform segment-based strategy (Wang et al., 2016a) and render them at a $448 \times 448$ resolution. The camera intrinsics are derived from a 60° field of view in both horizontal and vertical directions. To ensure stable training, we freeze the parameters of the language model and visual backbone, exclusively training a vision-to-language MLP connector and LoRA adapters (Hu et al., 2022). These adapters, configured with a rank of 32, a scaling factor $\alpha$ of 64, and a dropout rate of 0.05, are injected into the query/key/value/output projections of the self-attention modules and the up/down/gate projections of the MLP blocks. We employ the AdamW (Loshchilov & Hutter, 2017) optimizer with a batch size of 2, an initial learning rate of $2 \times 10^{-5}$, a weight decay of 0.05, and a per-step cosine annealing schedule (Loshchilov & Hutter, 2016) with a 0.03 warm-up ratio. The four training stages are trained for 1, 1, 1, and 3 epochs, respectively. For evaluation, we conduct a single test run where the predicted label is compared against the ground-truth after standard text normalization (lowercasing and whitespace trimming).

### E.2. DrAction Implementation

This section provides a detailed exposition of our differentiable renderer, DrAction, elaborating on the mathematical formulations, network architectures, and initialization strategies.

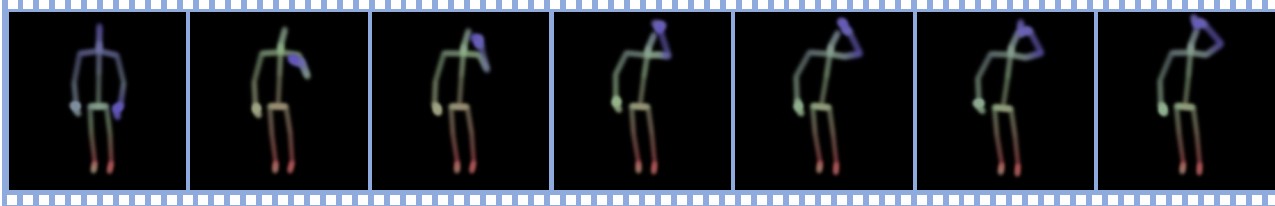

**Question:**
Carefully analyze the behavioral characteristics of the person in this action sequence. Select the most matching action category from the options below. You need to first describe your reasoning process, explain your rationale, and then output the final result.
A. neck pain
B. headache
C. back pain
D. salute

**MotionGPT:**
The person raises their arm and touches the upper body, with the action involving hand movement. Choice A.

**InternVL3 (fixed render) :**
The figure is observed raising an arm with the hand moving to the region near the head. The arm is in a bent position, with the hand approximately at forehead or temple height. Looking at the overall posture, the upper body remains relatively upright and stable throughout the sequence. The hand appears to be positioned at a fixed angle relative to the head, maintaining a consistent distance. This structured arm positioning, combined with the upright torso and the hand held at brow level, closely matches the formal gesture pattern commonly associated with a salute. Label: D. salute.

**SkeletonLLM w/o CR-Distill:**
The figure is observed raising an arm with the hand moving to the region near the head. The arm is in a bent position, with the hand approximately at forehead or temple height. The upper body remains relatively upright and stable throughout the sequence. The hand appears to be positioned at a fixed angle relative to the head, maintaining a consistent distance. This structured arm positioning, combined with the upright torso and the hand held at brow level, closely matches the formal gesture pattern commonly associated with a salute. Label: D. Salute.

**SkeletonLLM:**
The skeleton video shows a single figure performing a distinct action. The figure raises one hand toward the forehead and temple region, with the hand in a pressing posture rather than a flat salute position. The head is slightly drooped with subtle swaying motion, which is characteristic of headache-relief behavior. Evaluating the given options: neck pain typically involves hands touching the back or side of the neck, with the head held stiffly rather than swaying; chest pain would show contact in the chest cavity region; salute is a standardized gesture with the hand raised to the brow in a flat, non-contact position, accompanied by an upright body posture, and the hand does not press against the head. The pressing motion and drooped head suggest headache. Label: B. headache.

*Figure 12.* Qualitative comparison of reasoning processes. Given a "headache" sequence rendered by DrAction, we compare outputs from different models on a multiple-choice QA task. MotionGPT captures only coarse patterns and selects "neck pain." InternVL3 (fixed render) and SkeletonLLM w/o CR-Distill both misinterpret the gesture as a "salute," focusing on superficial posture cues ("upright and stable," "hand at brow level") while missing subtle pressing and head-drooping dynamics. Only the full SkeletonLLM correctly identifies "headache" by recognizing the pressing posture at the temples and head drooping, while systematically ruling out alternatives with causal reasoning.

### E.2.1. CANONICAL REPRESENTATION AND INITIALIZATION

The foundation of DrAction is a canonical, action-agnostic representation of the human form as a set of $K$ 3D Gaussian primitives. This representation is initialized once and subsequently deformed by skeletal motion.

**Canonical Gaussian Placement and Features.** The total number of Gaussians is $K = J + K_{\text{bone}}$, where $J$ is the number of joints. The first $J$ Gaussians are centered at the canonical joint positions, $\boldsymbol{\mu}_j^c = \mathbf{j}_j^c$ for $j \in \{1, \dots, J\}$. The canonical joint configuration $\{\mathbf{j}_j^c\}_{j=1}^J$ is initialized from the joint positions of the first available frame (e.g., the first frame of a given sequence) and remains fixed thereafter. The remaining $K_{\text{bone}}$ Gaussians are uniformly sampled along the skeletal bones, defined by anatomically connected joint pairs. For a bone connecting joints $a$ and $b$, we sample $N$ points, such that the $i$-th point is $\boldsymbol{\mu}_k^c = (1 - \alpha_i)\mathbf{j}_a^c + \alpha_i\mathbf{j}_b^c$, where $\alpha_i = i/(N + 1)$ for $i \in \{1, \dots, N\}$. The canonical orientation for all Gaussians is initialized to the identity quaternion, $\mathbf{q}_k^c = [1, 0, 0, 0]^\top$. The learnable appearance feature vector $\mathbf{f}_k \in \mathbb{R}^d$ for each Gaussian is initialized from a narrow Gaussian distribution, $\mathbf{f}_k \sim \mathcal{N}(\mathbf{0}, 0.01^2\mathbf{I})$.

**Adaptive Canonical Scaling.** The canonical scale $\mathbf{s}_k^c$ is initialized adaptively based on local bone structure to ensure appropriate coverage. For a joint Gaussian $j$, its scale is proportional to the median length of bones connected to it. For a bone Gaussian $k$, its scale is proportional to the length of the bone it lies on. This is formulated as:

$$s_j = \text{clip}\left(s_{\text{base}}^{\text{joint}}\left(\frac{\text{median}(\{L_b | j \in b\})}{L_{\max}}\right)^\gamma, s_{\min}^{\text{joint}}, s_{\max}^{\text{joint}}\right) \tag{10}$$

where $L_b$ is the length of bone $b$, $L_{\max}$ is the maximum bone length in the skeleton, and $s_{\text{base/min/max}}^{\text{joint}}$ are hyperparameters. A similar rule applies to bone Gaussians. The final scale vector is $\mathbf{s}_k^c = [s_k, s_k, s_k]^\top$.

**Linear Blend Skinning (LBS) Weights.** The blend weights $\mathbf{w}_k \in \mathbb{R}^J$ that associate each Gaussian $k$ with the skeleton's joints are pre-computed and fixed. For a joint Gaussian $j$, its weight vector is effectively a one-hot encoding, with a large positive logit for joint $j$ and large negative logits for all other joints. For a bone Gaussian $k$ sampled between joints $a$ and $b$ with interpolation factor $\alpha$, its weights are non-zero only for $a$ and $b$, with logits proportional to $\log(1 - \alpha)$ and $\log(\alpha)$ respectively, ensuring that it is primarily influenced by its parent joints. These fixed weights provide a stable kinematic prior.

### E.2.2. KINEMATIC DEFORMATION AND RENDERING

**LBS and Projection to SO(3).** Our kinematic deformation model utilizes Linear Blend Skinning (LBS) with pose-dependent rigid transforms derived from the skeleton's orientation data. For each joint $i$, its current orientation quaternion $\mathbf{q}_i$ is converted into a rotation matrix $\mathbf{R}_i = \text{quat2mat}(\mathbf{q}_i)$. These per-joint rotations are then blended using the LBS weights $\mathbf{w}_k$: $\tilde{\mathbf{R}}_k = \sum_{i=1}^J w_{k,i}\mathbf{R}_i$. Since the blended matrix $\tilde{\mathbf{R}}_k$ is not guaranteed to be in the special orthogonal group SO(3), it must be projected to the nearest valid rotation. This is robustly achieved via Singular Value Decomposition (Golub & Van Loan, 2013) (SVD). Let the SVD of the blended matrix be $\tilde{\mathbf{R}}_k = \mathbf{U}\boldsymbol{\Sigma}\mathbf{V}^\top$. The closest valid rotation is then given by:

$$\mathbf{R}_k^{\text{blend}} = \mathbf{U}\text{diag}(1, 1, \det(\mathbf{U}\mathbf{V}^\top))\mathbf{V}^\top. \tag{11}$$

This projection corrects for potential reflections and ensures numerical stability. The final posed rotation for a Gaussian is $\mathbf{R}_k = \mathbf{R}_k^{\text{blend}}\mathbf{R}_k^c$, where $\mathbf{R}_k^c$ is the rotation from its canonical quaternion. The posed covariance matrix $\boldsymbol{\Sigma}_k$ is then computed as in Eq. (5) of the main paper. It is worth noting that this full kinematic model is employed for datasets providing joint orientation, such as NTU-60 and NTU-120. For datasets like PKU-MMD that lack such data, our model gracefully falls back to a translation-only LBS, where each $\mathbf{R}_i$ is treated as an identity matrix.

**Camera Projection and Differentiable Rasterization.** Given camera intrinsics $\mathbf{K}$ and a world-to-camera transform $\mathbf{W}$, the 3D mean $\boldsymbol{\mu}_k$ is projected to 2D coordinates $\boldsymbol{\mu}_{k,\text{2D}}$. The 3D covariance $\boldsymbol{\Sigma}_k$ is projected to a 2D covariance $\boldsymbol{\Sigma}_{k,\text{2D}}$ via the Jacobian $\mathbf{J}$ of the perspective projection: $\boldsymbol{\Sigma}_{k,\text{2D}} = \mathbf{J}\boldsymbol{\Sigma}_k\mathbf{J}^\top$. For each pixel $\mathbf{x}$, the final color $\mathbf{I}(\mathbf{x})$ is computed via front-to-back alpha compositing:

$$\mathbf{I}(\mathbf{x}) = \sum_{k=1}^{K'} \mathbf{C}_k \alpha_k' \prod_{j=1}^{k-1} (1 - \alpha_j'),$$

$$\text{where} \quad \alpha_k'(\mathbf{x}) = 1 - \exp(-\alpha_k \cdot w_k(\mathbf{x})),$$

$$\text{and} \quad w_k(\mathbf{x}) = \exp\left(-\frac{1}{2}(\mathbf{x} - \boldsymbol{\mu}_{k,\text{2D}})^\top \boldsymbol{\Sigma}_{k,\text{2D}}^{-1}(\mathbf{x} - \boldsymbol{\mu}_{k,\text{2D}})\right). \tag{12}$$

Here, Gaussians are pre-sorted by depth, $w_k(\mathbf{x})$ is the 2D Gaussian influence at the pixel, and $\alpha_k$ is the final modulated opacity for Gaussian $k$.

**Camera Configuration and Viewpoint Selection.** In all experiments, DrAction uses a fixed pinhole camera whose parameters are fully determined by the image resolution and a symmetric field of view, and are not learned during training. Let the rendered frame resolution be $H \times W$ and the horizontal and vertical fields of view be $\phi_x$ and $\phi_y$,

respectively. The camera intrinsics are defined as

$$f_x = \frac{W}{2\tan(\phi_x/2)}, \quad f_y = \frac{H}{2\tan(\phi_y/2)},$$

$$c_x = \frac{W}{2}, \quad c_y = \frac{H}{2},$$

$$\mathbf{K} = \begin{bmatrix} f_x & 0 & c_x \\ 0 & f_y & c_y \\ 0 & 0 & 1 \end{bmatrix}. \tag{13}$$

In our implementation, we set $\phi_x = \phi_y = 60°$ and $H = W = 448$, so that $\mathbf{K}$ is uniquely determined by these hyperparameters and remains fixed across sequences.

The world-to-camera transform $\mathbf{W} \in \mathrm{SE}(3)$ is chosen as the identity, i.e.,

$$\mathbf{W} = \begin{bmatrix} \mathbf{I}_3 & \mathbf{0} \\ \mathbf{0}^\top & 1 \end{bmatrix}, \tag{14}$$

so that the 3D joint positions and Gaussian centers are expressed directly in the camera coordinate system. For each Gaussian center $\boldsymbol{\mu}_k = (X_k, Y_k, Z_k)^\top$, the corresponding homogeneous camera coordinate is

$$\tilde{\boldsymbol{\mu}}_k^c = \mathbf{W} \begin{bmatrix} \boldsymbol{\mu}_k \\ 1 \end{bmatrix} = \begin{bmatrix} X_k^c \\ Y_k^c \\ Z_k^c \\ 1 \end{bmatrix}, \tag{15}$$

and the pixel location $(u_k, v_k)$ is obtained via

$$\begin{bmatrix} u_k \\ v_k \\ 1 \end{bmatrix} = \mathbf{K} \begin{bmatrix} X_k^c/Z_k^c \\ Y_k^c/Z_k^c \\ 1 \end{bmatrix},$$

$$u_k = f_x \frac{X_k^c}{Z_k^c} + c_x, \quad v_k = -f_y \frac{Y_k^c}{Z_k^c} + c_y, \tag{16}$$

where the negative sign in the vertical direction follows the standard image-coordinate convention with $v$ increasing downward.

Before projection, we apply a simple depth normalization to keep the skeleton comfortably in front of the camera and to avoid numerical issues near $Z_k^c \approx 0$. Let $\{\mathbf{j}_i^t\}_{t,i}$ denote the joints of the entire sequence after LBS. We compute the sequence-level median depth

$$\tilde{z} = \mathrm{median}\left(\{(\mathbf{j}_i^t)_z \mid t = 1, \ldots, T, i = 1, \ldots, J\}\right), \tag{17}$$

and if $|\tilde{z}| < \tau$ (we use $\tau = 0.5$ in practice), we translate all joints and associated Gaussians along the optical axis by a constant offset $\Delta z$:

$$\mathbf{j}_i^{t\,'} = \mathbf{j}_i^t + \Delta z\,\mathbf{e}_z, \quad \Delta z = \begin{cases} 1, & |\tilde{z}| < \tau, \\ 0, & \text{otherwise}, \end{cases} \tag{18}$$

where $\mathbf{e}_z = (0,0,1)^\top$. This deterministic depth-shift is applied consistently across all frames to preserve temporal continuity. Together with the fixed $(\mathbf{K}, \mathbf{W})$, it defines a canonical, front-view camera for all sequences.

### E.2.3. NEURAL FEATURE MODULATOR (NFM) ARCHITECTURE

**Input Feature Vector.** The NFM conditions appearance on local kinematics, enabling the renderer to highlight motion-salient regions adaptively. For each Gaussian $k$, we compute its aggregated position $\mathbf{p}_k^t$ and velocity $\mathbf{v}_k^t$ via the same LBS weights used for geometric deformation:

$$\mathbf{p}_k^t = \sum_{i=1}^{J} w_{k,i} \mathbf{j}_i^t, \qquad \mathbf{v}_k^t = \sum_{i=1}^{J} w_{k,i} \dot{\mathbf{j}}_i^t. \tag{19}$$

A lightweight appearance head $\mathrm{MLP}_{\mathrm{app}} : \mathbb{R}^d \to \mathbb{R}^4$ maps the learnable canonical feature $\mathbf{f}_k$ to a base RGBA value: $\mathrm{RGBA}_k^{\mathrm{base}} = \mathrm{MLP}_{\mathrm{app}}(\mathbf{f}_k)$. The NFM input is the concatenation $\mathbf{x}_k^t = [\mathbf{p}_k^t, \mathbf{v}_k^t, \mathrm{RGBA}_k^{\mathrm{base}}] \in \mathbb{R}^{10}$.

**Network Architecture and Output.** A single-layer GRU (hidden size 10) models temporal dependencies across frames, producing $\mathbf{h}_k^t = \mathrm{GRU}(\mathbf{x}_k^t, \mathbf{h}_k^{t-1})$. A two-layer MLP $(10 \to 64 \to 5)$ with ReLU activation then predicts:

$$[\Delta \mathrm{RGB}_k, \Delta \alpha_k, g_k] = \mathrm{MLP}_{\mathrm{NFM}}(\mathbf{h}_k^t), \tag{20}$$

where $\Delta \mathrm{RGB}_k \in \mathbb{R}^3$ and $\Delta \alpha_k \in \mathbb{R}$ are appearance residuals, and $g_k \in \mathbb{R}$ is a saliency logit. The modulated color and opacity are computed as:

$$\mathbf{C}_k^{\mathrm{learned}} = \sigma(\mathrm{RGB}_k^{\mathrm{base}} + \Delta \mathrm{RGB}_k), \tag{21}$$

$$\alpha_k^{\mathrm{learned}} = \sigma(\alpha_k^{\mathrm{base}} + \Delta \alpha_k) \cdot \sigma(g_k), \tag{22}$$

where $\sigma(\cdot)$ denotes the sigmoid function. The saliency gate $\sigma(g_k)$ allows the network to suppress visually uninformative Gaussians (e.g., stationary joints) while amplifying those undergoing significant motion.

**Depth-Color Blending.** The final rendered color blends the NFM-modulated appearance with a depth-based pseudo-color visualization:

$$\mathbf{C}_k = (1 - \lambda)\,\mathbf{C}_k^{\mathrm{learned}} + \lambda\,\mathbf{C}_k^{\mathrm{depth}}, \tag{23}$$

where $\lambda = \sigma(\theta_{\mathrm{mix}})$ is a learnable mixing weight and $\mathbf{C}_k^{\mathrm{depth}}$ maps the Gaussian's depth to a triangular RGB colormap (red $\to$ green $\to$ blue from near to far). This blending preserves spatial structure while allowing learned colors to encode semantic motion cues.

### E.3. Details of MQA, Disc-FT and CR-Distill

This section provides additional details on the three training tasks used in SkeletonLLM—the Multiple-Choice Question & Answer (MQA) task, Discriminative Finetuning (Disc-FT), and Causal Reasoning Distillation (CR-Distill). We include the corresponding natural-language prompt templates, and the complete prompt texts are shown in Figure 13.

## Prompt for Multiple-Choice Question & Answer

**SYSTEM PROMPT**

You are an expert in video categorization and human action recognition.

**USER INPUT**

Frame 1: <IMG_TOKENS>
Frame 2: <IMG_TOKENS>

…

Frame 12: <IMG_TOKENS>

The clip you see is rendered from 3D skeleton data into abstract pseudo-images. Each frame encodes joint positions, bone connections, depth, and motion cues, and may look different from natural RGB videos. Treat these frames as a faithful visualization of human body kinematics.
You need to identify the action in the skeleton video and select one from the action labels below as the classification result. You are required to output your thought process, and finally, provide the specific classification label on a separate line in the format 'Label: <action>'.

- drink water
- eat food
- brush teeth
- brush hair
- drop something
- pick up
- throw
- sit down
- stand up
- clapping
- reading
- writing

…

## Prompt for Discriminative Finetuning

**SYSTEM PROMPT**

You are an expert in video categorization and human action recognition.

**USER INPUT**

Frame 1: <IMG_TOKENS>
Frame 2: <IMG_TOKENS>

…

Frame 12: <IMG_TOKENS>

The clip you see is rendered from 3D skeleton data into abstract pseudo-images. Each frame encodes joint positions, bone connections, depth, and motion cues, and may look different from natural RGB videos. Treat these frames as a faithful visualization of human body kinematics.
Is the action in this video clip '{action}'? Answer with ONLY the word \"YES\" or \"NO\".

## Teacher Prompt for Causal Reasoning Distillation

**SYSTEM PROMPT**

You are an expert in video categorization and human action recognition.

**USER INPUT**

Frame 1: <IMG_TOKENS>
Frame 2: <IMG_TOKENS>

…

Frame 12: <IMG_TOKENS>

This clip is from the NTU RGB+D dataset for human action recognition. It is rendered from 3D skeleton data into abstract motion-aware images.
Ground-truth action label: <action_name>
Please:
1) Briefly summarize what the person is doing in this skeleton video.
2) Provide a step-by-step causal chain that explains the action over time, explicitly referring to body parts and their motion (e.g., which joints move first, how limbs follow, how the pose evolves).
3) Conclude with the final classification label.
End your response with exactly one line:
Label: <action_name>

## Prompt for Causal Reasoning Distillation

**SYSTEM PROMPT**

You are an expert in video categorization and human action recognition.

**USER INPUT**

Frame 1: <IMG_TOKENS>
Frame 2: <IMG_TOKENS>

…

Frame 12: <IMG_TOKENS>

The clip you see is rendered from 3D skeleton data into abstract pseudo-images. Each frame encodes joint positions, bone connections, depth, and motion cues, and may look different from natural RGB videos. Treat these frames as a faithful visualization of human body kinematics.
Please describe what the person does in the skeleton video, infer the likely activity, and end with exactly one classification label on a separate line formatted as 'Label: <action>'.

*Figure 13.* Prompt templates for MQA, Disc-FT, and CR-Distill. (Top-left) Prompt template for the MQA task. (Bottom-left) Prompt template for Disc-FT. (Top-right) Teacher prompt for CR-Distill. (Bottom-right) Student prompt for CR-Distill.

### E.3.1. MULTIPLE-CHOICE QUESTION & ANSWER

We formulate open-vocabulary skeleton-based action recognition as a multiple-choice question answering task over a set of candidate action labels. For each clip, DrAction renders a sequence of $N_{frames}$ pseudo-images that are fed to the MLLM together with a text query listing the candidate classes and instructing the model to output a final decision in the format `Label: <action>`. During training, the candidate list contains only seen (training) classes; at test time, it is constructed from the corresponding evaluation split. The corresponding prompt template is illustrated in Figure 13 (top-left).

We optimize the model by minimizing the negative log-likelihood of the ground-truth answer sequence, focusing on the tokens forming the final label line. At inference, we use greedy autoregressive decoding, selecting at each step the token with the highest predicted probability until termination. We then extract the action string following the `Label:` prefix via regex and apply standard text normalization (lowercasing, stripping leading/trailing whitespace) before comparing with the ground-truth label.

### E.3.2. DISCRIMINATIVE FINETUNING

Discriminative Finetuning (Disc-FT) is the second stage in our progressive training strategy. Its goal is to sharpen the decision boundaries between visually similar actions by teaching SkeletonLLM to answer targeted binary questions. Instead of selecting a label from a long candidate list, the model is asked whether a rendered clip depicts a specific action and must respond with a single "YES" or "NO" token. The exact prompt template is shown in Figure 13 (bottom-left).

To focus learning on hard cases, Disc-FT operates on pairs of semantically similar actions. For each dataset and for every class name $y$, we query a strong MLLM (Hurst et al., 2024) with all other class names and ask it to rank which actions are most similar to $y$ in terms of body parts involved and motion patterns. We then retain the top-5 most similar actions as candidate negatives for $y$. The mined neighbors for NTU-60 and NTU-120 are visualized in Figures 14 and 15. Several consistent patterns emerge: actions sharing similar body-part involvement form natural clusters (e.g., hand-centric actions such as "clapping," "rub two hands," "hand waving," and "cheer"), and many confusing pairs differ only in subtle temporal or spatial cues (e.g., "put on shoe" vs. "take off shoe," "bow" vs. "hopping"). These mined neighborhoods directly validate Disc-FT's design: the hardest recognition errors stem from semantically proximate categories that require fine-grained visual distinction.

Disc-FT must also respect the open-vocabulary evaluation protocol. For a given split (e.g., the 48/12 split on NTU-60),

only classes in the seen set are allowed to appear in training prompts. Thus, for each anchor class $y$ in the seen set, we intersect its MLLM-mined neighbor list with the seen classes of the current split and discard any neighbors that belong to the unseen set. Given a training clip $(\mathbf{V}, y)$ whose label $y$ is in the seen set, we construct Disc-FT samples based on this filtered neighbor set. For positive ("YES") samples, we instantiate the template with the true label $y$, and the correct target token is "YES". For negative ("NO") samples, we randomly select a label $z$ from the filtered candidate set of $y$ (with $z \neq y$) and ask the same question with the action replaced by $z$; the correct target token is "NO". We balance the training set such that the number of "YES" and "NO" examples is approximately 1:1.

### E.3.3. CAUSAL REASONING DISTILLATION

CR-Distill aims to inject explicit, temporally grounded reasoning into SkeletonLLM. Instead of supervising the model only with class labels, we use a stronger MLLM as a teacher to provide step-wise causal chains that describe how the action unfolds over time.

We adopt GPT-4o (Hurst et al., 2024) as the teacher model. For each training clip, we first render the skeleton sequence into pseudo-image frames using the DrAction renderer after Stage 1 (Render Warm-up) and Stage 2 (Disc-FT) have been trained. These frames are fed to the teacher together with the prompt shown in Figure 13 (top-right). The prompt states that the clip is rendered from 3D skeleton data, provides the ground-truth action label, and asks the teacher to (i) briefly summarize the action, (ii) produce a step-by-step causal chain that explicitly refers to body parts and their motion, and (iii) end with a final line `Label: <action_name>`. Representative examples of such teacher-generated causal chains are illustrated in Figure 16.

Given the same rendered frames, SkeletonLLM is queried with the student prompt in Figure 13 (bottom-right), which explains that the input frames are abstract pseudo-images encoding joint positions, bone connections, depth, and motion cues, and asks the model to describe the action and output a final label line formatted as `Label: <action>`. The student is not given the ground-truth label in the prompt; instead, it is trained to reproduce the full teacher response, including both the causal rationale and the terminal `Label:` line. We minimize an auto-regressive cross-entropy loss over the teacher token sequence, masking out the prompt tokens. During this stage, we update the parameters of DrAction, the vision-to-language projector, and the LoRA adapters of the language model, encouraging the whole pipeline to encode motion cues in a way that supports detailed, body-part-specific reasoning.

| action | similar_1 | similar_2 | similar_3 | similar_4 | similar_5 |
|---|---|---|---|---|---|
| drink water | eat food | brush teeth | sneezing/cough | wipe face | phone call |
| eat food | drink water | brush teeth | sneezing/cough | wipe face | nausea/vomiting |
| brush teeth | drink water | eat food | wipe face | sneezing/cough | nausea/vomiting |
| brush hair | put on hat/cap | take off hat/cap | put on glasses | take off glasses | brush teeth |
| drop something | pick up | throw | tear up paper | reach into pocket | giving object |
| pick up | drop something | throw | reach into pocket | giving object | touch someone's pocket |
| throw | pick up | drop something | giving object | point at something/directions | hand waving |
| sit down | stand up | falling down | staggering | jump up | hopping |
| stand up | sit down | falling down | staggering | jump up | hopping |
| clapping | rub two hands | pray with hands together | hand waving | cheer (raise arms) | cross hands in front |
| reading | writing | type on keyboard | play with phone/tablet | tear up paper | put on glasses |
| writing | type on keyboard | reading | play with phone/tablet | tear up paper | put on glasses |
| tear up paper | reading | writing | drop something | throw | pick up |
| put on jacket | take off jacket | put on hat/cap | take off hat/cap | put on shoe | take off shoe |
| take off jacket | put on jacket | put on hat/cap | take off hat/cap | put on shoe | take off shoe |
| put on shoe | take off shoe | sit down | stand up | bow (bend forward) | hopping |
| take off shoe | put on shoe | sit down | stand up | bow (bend forward) | hopping |
| put on glasses | take off glasses | put on hat/cap | take off hat/cap | brush hair | wipe face |
| take off glasses | put on glasses | put on hat/cap | take off hat/cap | brush hair | wipe face |
| put on hat/cap | take off hat/cap | brush hair | put on glasses | take off glasses | salute |
| take off hat/cap | put on hat/cap | brush hair | put on glasses | take off glasses | salute |
| cheer (raise arms) | hand waving | clapping | jump up | salute | point at something/directions |
| hand waving | cheer (raise arms) | clapping | salute | point at something/directions | point at someone |
| kicking the air/something | kicking someone | jump up | hopping | falling down | staggering |
| reach into pocket | touch someone's pocket | giving object | take a selfie | play with phone/tablet | check time (from watch) |
| hopping | jump up | kicking the air/something | falling down | staggering | stand up |
| jump up | hopping | kicking the air/something | falling down | staggering | stand up |
| phone call | play with phone/tablet | take a selfie | check time (from watch) | reach into pocket | reading |
| play with phone/tablet | phone call | type on keyboard | take a selfie | check time (from watch) | reach into pocket |
| type on keyboard | writing | reading | play with phone/tablet | check time (from watch) | phone call |
| point at something/directions | point at someone | hand waving | salute | cheer (raise arms) | giving object |
| take a selfie | play with phone/tablet | phone call | check time (from watch) | reach into pocket | type on keyboard |
| check time (from watch) | phone call | play with phone/tablet | take a selfie | reach into pocket | type on keyboard |
| rub two hands | clapping | pray with hands together | cross hands in front | hand waving | cheer (raise arms) |
| bow (bend forward) | put on shoe | take off shoe | sit down | stand up | falling down |
| shake head | headache | neck pain | wipe face | sneezing/cough | take off hat/cap |
| wipe face | brush teeth | sneezing/cough | drink water | eat food | take off glasses |
| salute | hand waving | cheer (raise arms) | clapping | point at something/directions | point at someone |
| pray with hands together | rub two hands | cross hands in front | clapping | bow (bend forward) | salute |
| cross hands in front | pray with hands together | rub two hands | clapping | salute | hand waving |
| sneezing/cough | wipe face | drink water | eat food | brush teeth | nausea/vomiting |
| staggering | falling down | stand up | sit down | jump up | hopping |
| falling down | staggering | sit down | stand up | jump up | hopping |
| headache | neck pain | chest pain | back pain | nausea/vomiting | shake head |
| chest pain | back pain | neck pain | headache | nausea/vomiting | fan self |
| back pain | neck pain | chest pain | headache | nausea/vomiting | bow (bend forward) |
| neck pain | headache | chest pain | back pain | shake head | wipe face |
| nausea/vomiting | fan self | sneezing/cough | wipe face | headache | chest pain |
| fan self | nausea/vomiting | sneezing/cough | wipe face | headache | chest pain |
| punch/slap someone | pushing someone | kicking someone | pat someone on the back | hug each other | shaking hands |
| kicking someone | kicking the air/something | punch/slap someone | pushing someone | jump up | hopping |
| pushing someone | punch/slap someone | pat someone on the back | hug each other | shaking hands | giving object |
| pat someone on the back | hug each other | shaking hands | pushing someone | punch/slap someone | giving object |
| point at someone | point at something/directions | hand waving | salute | cheer (raise arms) | giving object |
| hug each other | shaking hands | pat someone on the back | pushing someone | punch/slap someone | walking towards each other |
| giving object | touch someone's pocket | reach into pocket | point at someone | point at something/directions | shaking hands |
| touch someone's pocket | reach into pocket | giving object | pick up | drop something | point at someone |
| shaking hands | hug each other | pat someone on the back | pushing someone | point at someone | hand waving |
| walking towards each other | walking apart from each other | hug each other | shaking hands | pushing someone | pat someone on the back |
| walking apart from each other | walking towards each other | hug each other | shaking hands | pushing someone | pat someone on the back |

*Figure 14.* Top-5 MLLM-mined semantically similar actions for each class on NTU-60, used as hard-negative candidates in Disc-FT. Actions sharing similar body-part involvement form natural clusters (e.g., "clapping," "rub two hands," "pray with hands together," "high-five"); confusing pairs often differ only in subtle temporal or spatial cues (e.g., "put on shoe" vs. "take off shoe," "sit down" vs. "stand up").

| action | similar_1 | similar_2 | similar_3 | similar_4 | similar_5 |
|---|---|---|---|---|---|
| drink water | eat food | cheers and drink | open a bottle | wipe face | brush teeth |
| eat food | drink water | cheers and drink | brush teeth | wipe face | sniff/smell |
| brush teeth | wipe face | apply cream on face | blow nose | sneezing/cough | yawn |
| brush hair | touch hair | put on hat/cap | take off hat/cap | put on hat/cap | take off glasses |
| drop something | pick up | toss a coin | throw | throw ball (shoot a basket) | throw up hat/cap |
| pick up | drop something | squat down | put object into bag | take object out of bag | move heavy objects |
| throw | throw ball (shoot a basket) | toss a coin | throw up hat/cap | bounce ball | swing paddle |
| sit down | stand up | squat down | bow (bend forward) | jump up | hopping |
| stand up | sit down | squat down | jump up | hopping | run on the spot |
| clapping | rub two hands | pray with hands together | high-five | hand waving | cheer (raise arms) |
| reading | writing | play with phone/tablet | take a photo | tear up paper | check time (from watch) |
| writing | reading | type on keyboard | play with phone/tablet | staple papers with a small stapler | fold paper |
| tear up paper | cutting paper | ball up paper | fold paper | put on hat/cap | writing |
| put on jacket | take off jacket | put on bag | take off bag | take off hat/cap | take off hat/cap |
| take off jacket | put on jacket | take off bag | put on bag | take off hat/cap | put on hat/cap |
| put on shoe | take off shoe | squat down | sit down | stand up | touch toes repeatedly |
| take off shoe | put on shoe | squat down | sit down | stand up | touch toes repeatedly |
| put on glasses | take off glasses | put on hat/cap | take off hat/cap | put on headphones | take off headphones |
| take off glasses | put on glasses | take off hat/cap | put on hat/cap | take off headphone | put on headphones |
| put on hat/cap | take off hat/cap | brush hair | put on glasses | put on glasses | put on headphone |
| take off hat/cap | put on hat/cap | brush hair | take off glasses | put on glasses | take off headphone |
| cheer (raise arms) | hand waving | clapping | high-five | give thumbs up | make victory sign |
| hand waving | cheer (raise arms) | clapping | high-five | point at something/directions | give thumbs up |
| kicking the air/something | kicking someone | side kick | butt kicks | run on the spot | jump up |
| reach into pocket | touch someone's pocket | put object into bag | take object out of bag | open a box | grab something from someone |
| hopping | jump up | run on the spot | butt kicks | side kick | kicking the air/something |
| jump up | hopping | run on the spot | side kick | kicking the air/something | butt kicks |
| phone call | play with phone/tablet | take a selfie | check time (from watch) | reading | writing |
| play with phone/tablet | phone call | take a selfie | take a photo | check time (from watch) | type on keyboard |
| type on keyboard | writing | reading | play with phone/tablet | counting money | play with a rubik's cube |
| point at something/directions | hand waving | point at someone | give thumbs up | make victory sign | make victory sign |
| take a selfie | take a photo | phone call | play with phone/tablet | take a photo | reading |
| check time (from watch) | phone call | play with phone/tablet | take a selfie | check time (from watch) | high-five |
| rub two hands | clapping | pray with hands together | cross hands in front | shaking hands | salute |
| bow (bend forward) | squat down | sit down | stand up | pray with hands together | sneezing/cough |
| shake head | brush hair | touch hair | wipe face | yawn | blow nose |
| wipe face | brush teeth | apply cream on face | brush hair | touch hair | give thumbs up |
| salute | hand waving | rub two hands | point at something/directions | point at someone | high-five |
| pray with hands together | clapping | rub two hands | cross hands in front | bow (bend forward) | capitulate |
| cross hands in front | pray with hands together | rub two hands | clapping | cross arms | drink water |
| sneezing/cough | blow nose | yawn | nausea/vomiting | wipe face | nausea/vomiting |
| staggering | falling down | run on the spot | walking towards each other | walking apart from each other | drop something |
| falling down | staggering | squat down | sit down | stand up | wipe face |
| headache | neck pain | back pain | chest pain | nausea/vomiting | staggering |
| chest pain | back pain | neck pain | headache | nausea/vomiting | stretch oneself |
| back pain | neck pain | chest pain | headache | nausea/vomiting | shake head |
| neck pain | headache | back pain | chest pain | nausea/vomiting | back pain |
| nausea/vomiting | sneezing/cough | blow nose | headache | chest pain | hand waving |
| fan self | wipe face | yawn | sniff/smell | shake head | throw |
| punch/slap someone | kicking someone | pushing someone | hit someone with an object | swing a knife at someone | side kick |
| kicking someone | punch/slap someone | pushing someone | step on someone's foot | kicking the air/something | grab something from someone |
| pushing someone | punch/slap someone | kicking someone | knock into someone | pat someone on the back | high-five |
| pat someone on the back | hug each other | shaking hands | support someone | pushing someone | give thumbs down |
| point at someone | point at something/directions | hand waving | salute | give thumbs up | giving object |
| hug each other | shaking hands | high-five | pat someone on the back | support someone | shaking hands |
| giving object | exchange objects | take object out of bag | put object into bag | grab something from someone | put object into bag |
| touch someone's pocket | reach into pocket | grab something from someone | giving object | take object out of bag | pray with hands together |
| shaking hands | high-five | hug each other | pat someone on the back | rub two hands | run on the spot |
| walking towards each other | walking apart from each other | hug each other | shaking hands | high-five | follow someone |
| walking apart from each other | walking towards each other | run on the spot | stand up | sit down | put on hat/cap |
| put on headphones | take off headphone | phone call | play with phone/tablet | check time (from watch) | take off hat/cap |
| take off headphone | put on headphones | phone call | play with phone/tablet | check time (from watch) | juggle table tennis ball |
| throw ball (shoot a basket) | throw | toss a coin | bounce ball | swing paddle | arm swings |
| bounce ball | throw ball (shoot a basket) | bounce ball | juggle table tennis ball | swing paddle | arm swings |
| swing paddle | throw ball (shoot a basket) | bounce ball | throw | juggle table tennis ball | toss a coin |
| juggle table tennis ball | bounce ball | swing paddle | throw ball (shoot a basket) | throw | writing |
| hush | whisper to someone | phone call | play with phone/tablet | reading | put on hat/cap |
| touch hair | brush hair | shake head | wipe face | apply cream on face | point at someone |
| give thumbs up | give thumbs down | make ok sign | make victory sign | point at something/directions | point at someone |
| give thumbs down | give thumbs up | make ok sign | make victory sign | point at something/directions | point at someone |
| make ok sign | make victory sign | give thumbs up | give thumbs down | point at something/directions | hand waving |
| make victory sign | make ok sign | give thumbs up | give thumbs down | point at something/directions | point at someone |
| staple papers with a small stapler | cutting paper | tear up paper | fold paper | ball up paper | writing |
| counting money | tear up paper | fold paper | ball up paper | staple papers with a small stapler | type on keyboard |
| cutting nails | cutting paper | tear up paper | apply cream on hand | apply cream on face | fold paper |
| cutting paper | tear up paper | fold paper | ball up paper | staple papers with a small stapler | writing |
| snap fingers | rub two hands | clapping | give thumbs up | give thumbs down | make ok sign |
| open a bottle | drink water | eat food | cheers and drink | open a box | reach into pocket |
| sniff/smell | sneezing/cough | blow nose | yawn | fan self | drink water |
| squat down | sit down | stand up | bow (bend forward) | put on shoe | take off shoe |
| toss a coin | throw | throw ball (shoot a basket) | juggle table tennis ball | bounce ball | swing paddle |
| fold paper | tear up paper | ball up paper | cutting paper | staple papers with a small stapler | writing |
| ball up paper | tear up paper | fold paper | cutting paper | staple papers with a small stapler | writing |
| play with a rubik's cube | counting money | cutting nails | fold paper | ball up paper | tear up paper |
| apply cream on face | wipe face | brush teeth | brush hair | touch hair | apply cream on hand |
| apply cream on hand | apply cream on face | cutting nails | rub two hands | wipe face | touch hair |
| put on bag | take off bag | put on jacket | take off jacket | put object into bag | take object out of bag |
| take off bag | put on bag | take off jacket | put on jacket | put object into bag | take object out of bag |
| put object into bag | take object out of bag | put on bag | take off bag | reach into pocket | open a box |
| take object out of bag | put object into bag | put on bag | take object out of bag | reach into pocket | open a box |
| open a box | reach into pocket | put object into bag | pick up | drop something | pick up |
| move heavy objects | carry objects together | drop something | give thumbs up | put on bag | take off bag |
| shake fist | cheer (raise arms) | hand waving | throw ball (shoot a basket) | give thumbs down | make ok sign |
| throw up hat/cap | throw | toss a coin | cross arms | bounce ball | swing paddle |
| capitulate | bow (bend forward) | cross hands in front | bow (bend forward) | pray with hands together | hug each other |
| cross arms | cross hands in front | pray with hands together | throw ball (shoot a basket) | capitulate | swing paddle |
| spin arms in circles | arm swings | throw | throw ball (shoot a basket) | bounce ball | swing paddle |
| arm swings | spin arms in circles | throw | kicking the air/something | bounce ball | side kick |
| run on the spot | hopping | jump up | run on the spot | butt kicks | jump up |
| butt kicks | kicking the air/something | side kick | stand up | hopping | side kick |
| touch toes repeatedly | squat down | sit down | butt kicks | bow (bend forward) | stretch oneself |
| side kick | kicking someone | kicking the air/something | sniff/smell | run on the spot | hopping |
| yawn | sneezing/cough | blow nose | sit down | fan self | wipe face |
| stretch oneself | touch toes repeatedly | yawn | sniff/smell | stand up | bow (bend forward) |
| blow nose | sneezing/cough | yawn | sniff/smell | wipe face | fan self |
| hit someone with an object | punch/slap someone | kicking someone | pushing someone | swing a knife at someone | throw |
| swing a knife at someone | hit someone with an object | punch/slap someone | kicking someone | pushing someone | throw |
| knock into someone | pushing someone | punch/slap someone | kicking someone | hit someone with an object | swing a knife at someone |
| grab something from someone | giving object | touch someone's pocket | reach into pocket | take object out of bag | put object into bag |
| point a gun at someone | punch/slap someone | kicking someone | pushing someone | hit someone with an object | swing a knife at someone |
| step on someone's foot | kicking someone | kicking the air/something | side kick | butt kicks | run on the spot |
| high-five | shaking hands | hug each other | pat someone on the back | clapping | rub two hands |
| cheers and drink | drink water | eat food | pick up | clapping | hug each other |
| carry objects together | move heavy objects | drop something | phone call | put object into bag | take object out of bag |
| take a photo | take a selfie | play with phone/tablet | phone call | check time (from watch) | reading |
| follow someone | walking towards each other | walking apart from each other | run on the spot | support someone | giving object |
| whisper to someone | hush | phone call | play with phone/tablet | pat someone on the back | hug each other |
| exchange objects | giving object | touch someone's pocket | take object out of bag | giving object | pat someone on the back |
| support someone | hug each other | shaking hands | pat someone on the back | giving object | exchange objects |
| rock-paper-scissors | make ok sign | make victory sign | give thumbs up | give thumbs down | snap fingers |

*Figure 15.* Top-5 MLLM-mined semantically similar actions for each class on NTU-120, used as hard-negative candidates in Disc-FT.

## Examples of Teacher Causal Chains for Causal Reasoning Distillation

This clip illustrates a person putting on an upper garment such as a jacket. The sequence begins with the person extending their arms backwards and downwards, angling the elbows to guide the hands into imaginary sleeves behind the back. As the motion progresses, the arms rotate upwards and forwards over the shoulders, accompanied by a shrugging motion of the shoulders to settle the garment on the body. Finally, the hands converge at the center of the torso near the chest or waist, performing a zipping or buttoning gesture, which completes the dressing process.

Label: Put on jacket

────────────────────────────────────────────────────────

This clip depicts a sudden loss of balance and a fall. Initially in a standing or walking posture, the skeleton's center of mass shifts uncontrollably, causing the vertical coordinates of the head and spine to drop rapidly. The arms may flail outwards or upwards instinctively to break the fall or regain balance. The motion is characterized by a lack of control and high velocity descent, ending with the skeleton oriented horizontally near the ground level, clearly distinguishing it from a controlled action like lying down.

Label: Falling down

────────────────────────────────────────────────────────

The sequence demonstrates a person experiencing distress in the chest area. The subject abruptly or deliberately raises a hand, typically pressing the palm firmly against the center or left side of the chest. Simultaneously, the torso hunches forward or curls slightly, reflecting a somatic reaction to internal pain. The hand remains clutched to the chest area while the body posture stiffens, and the head may bow down, creating a clear causal chain representing cardiac distress or chest pain.
Label: Chest pain

────────────────────────────────────────────────────────

The skeleton video shows a person lowering their body into a squatting position. Initially standing upright, the person bends their knees deeply while simultaneously lowering their hips towards the ground. The torso leans slightly forward to maintain balance, and the arms may extend forward or rest on the knees for stability. The descent is smooth and controlled, reaching a low point where the thighs are parallel to the ground or lower, clearly distinguishing this controlled exercise or posture change from a fall or sitting on a chair.

Label: Squat down

────────────────────────────────────────────────────────

This clip depicts a person folding their arms across their chest in a classic closed posture. The sequence begins with the arms relaxed at the sides. The person then bends their elbows and brings both forearms simultaneously across the front of the torso. The hands travel towards the opposite elbows or biceps, effectively locking the arms in a horizontal position across the chest. The final pose is held statically, signifying a defensive, waiting, or determined stance.

Label: Cross arms

────────────────────────────────────────────────────────

The sequence shows a person grooming or adjusting their hair. One or both hands are raised to the head region. Unlike a headache gesture which involves a static press, the hands here perform smooth, brushing or sweeping motions along the sides, top, or back of the head. The wrist joints trace the curvature of the skull, mimicking the action of smoothing down hair or tucking it behind the ears, indicating a grooming behavior.

Label: Touch hair

────────────────────────────────────────────────────────

The sequence captures the distinctive action of putting on a backpack. The person reaches one or both arms backwards and downwards, extending the elbows behind the torso to mimic sliding arms through straps. Following this, there is often a shrugging motion of the shoulders and a slight forward lean of the torso to settle the weight. The hands may then move to the chest or shoulder area to adjust the straps, confirming the interaction with a back-mounted object.

Label: Put on bag

*Figure 16.* Examples of teacher-generated causal reasoning chains used in CR-Distill.

## F. Design Rationale: Why 3DGS + LBS

We employ 3D Gaussian Splatting driven by Linear Blend Skinning to address the specific constraints of skeletal data: the need for structural preservation, kinematic validity, format agnosticism, and differentiable optimization.

3DGS offers a discrete set of primitives that naturally map to the nodes and edges of a skeleton graph. Unlike implicit neural fields (e.g., NeRF) which represent geometry continuously, 3DGS allows us to explicitly attach visual elements to specific body parts. This discrete correspondence is vital for part-aware reasoning, as it ensures that the learned visual features are directly grounded in the underlying skeletal topology. Crucially, 3DGS primitives can be dynamically instantiated based on the input skeleton's joint count and connectivity, enabling format-agnostic rendering across heterogeneous skeleton topologies (e.g., Kinect, MoCap, 2D poses).

LBS acts as a robust kinematic regularizer. By binding the Gaussian primitives to joint transforms, LBS ensures that the rendered motion remains physically plausible and temporally consistent. Without this constraint, optimizing primitives freely could lead to artifacts where visual elements detach from the skeletal structure, breaking the semantic link between motion and appearance. LBS also provides a principled mechanism for handling arbitrary skeleton formats: the blending weights are computed based on the input skeleton's adjacency structure rather than a fixed template.

This combination enables a design that is both structured and expressive. While LBS enforces geometric consistency, the learnable parameters of 3DGS (e.g., opacity, color) provide the degrees of freedom needed to encode semantic information. Through end-to-end gradients, the model can learn to highlight salient motion cues (such as a moving limb) or encode velocity into color, creating a visual language optimized for understanding rather than mere reconstruction. The format-agnostic nature of this design is central to achieving universal skeleton understanding across diverse data sources.

## G. Future Work

SkeletonLLM takes a first step toward enabling MLLMs to operate on sparsely structured, non-visual signals by translating skeleton sequences into a task-optimized visual language. Natural extensions include applying DrAction-style differentiable rendering to other structured modalities (e.g., LiDAR point clouds, object trajectories), and exploiting the differentiability for motion generation—optimizing skeleton sequences to match textual descriptions for text-to-motion synthesis.

Another avenue concerns long-horizon, compositional ac-

tion understanding. Our current evaluation focuses on short clips depicting atomic actions, yet real-world scenarios often involve minute-scale sequences with multiple sub-actions and temporal dependencies, requiring hierarchical temporal modeling and efficient memory mechanisms.

Finally, improving visual token efficiency remains an open challenge. As shown in Table 12, reducing resolution from $448 \times 448$ to $224 \times 224$ leads to $\sim$5% performance degradation. More fundamentally, rendered skeleton images are inherently sparse: the human figure occupies only a small fraction of pixels, while the majority of the frame consists of uninformative black background. This sparsity suggests that current dense visual encoding is suboptimal. Potential solutions include sparse attention mechanisms that focus computation on skeleton-occupied regions, or hybrid representations that combine compact rendered patches with numerical joint encodings to reduce redundant pixel processing.

