# OpenReview forum: "Universal Skeleton Understanding via Differentiable Rendering and MLLMs"
_ICML.cc/2026/Conference — ICML 2026 regular_

### Official Review · Reviewer_qHqH · 2026-03-08

**Soundness:** 3
**Presentation:** 3
**Significance:** 3
**Originality:** 2
**Overall Recommendation:** 4
**Confidence:** 3

**Summary:**

The paper proposes SkeletonLLM, a framework for “universal skeleton understanding” that translates heterogeneous skeleton sequences into the native visual modality of multimodal LLMs (MLLMs). The core component is DrAction, a differentiable, format-agnostic renderer based on 3D Gaussian Splatting and Linear Blend Skinning that produces optimized image sequences from skeleton input, allowing gradients from the MLLM to shape the rendering.

**Compliance With Llm Reviewing Policy:**

Affirmed.

**Final Justification:**

After carefully read author's rebuttal, I maintain positive (weak accept) but not sure about my judgement, so I adjust my confidence. Thank you.

**Key Questions For Authors:**

- Please see the weakness section. Additionally,

- How are blend weights w_k defined across arbitrary skeletons and sequences? Are they recomputed per Gaussian via nearest joints or bone associations, and how do you ensure stability when J changes between datasets?

- What exactly are the camera intrinsics/extrinsics (K, W) during rendering? Do you normalize/align coordinate frames across datasets for viewpoint invariance, or learn per-sequence cameras? How sensitive is performance to viewpoint changes?

**Limitations:**

No, I only see a Future Work Section (G in appendix) but no limitation sections.

**Strengths And Weaknesses:**

Strengths:

- DrAction’s formulation (canonical 3D Gaussians + LBS + differentiable rasterization) is technically coherent and well-motivated for topology-agnostic rendering.

- Experimental rigor and validation: Comprehensive evaluation across open-vocabulary recognition (multiple NTU splits), cross-format transfer (Kinect v2↔v1, MoCap↔2D), and captioning (HumanML3D) supports the “universal” claim more convincingly than single-task studies.

Weaknesses:

- Format-agnosticity details are under-specified: how blend weights w_k are defined across arbitrary topologies, how K_bone and canonicalization are constructed per sequence, and how these choices scale with varying J are not fully clear.

- Camera modeling and viewpoint normalization are not rigorously explained (intrinsics K, extrinsics W, and how differing dataset/world coordinate frames are handled); this is critical for cross-format claims.

- “Open-vocabulary” recognition largely relies on multiple-choice QA (MQA). The size and composition of candidate sets, and whether unseen classes are included at inference, are not stated, making it hard to assess true open-set difficulty.

---

> ### Author Rebuttal · Authors · 2026-03-30
>
> We thank the reviewer for the thorough assessment and the positive recognition of DrAction's formulation ("technically coherent and well-motivated") and the experimental rigor ("supports the 'universal' claim more convincingly than single-task studies"). All three issues raised are important. Although complete technical specifications are in the Appendix (C.2/E.2/E.3), the main text indeed lacked sufficient clarity. Below we provide clarifications for each, supported by new experiments, which confirm that the main conclusions are robust.
>
> > Weakness (W) 1 / Question (Q) 1: Format-agnosticity details are under-specified: how blend weights w_k are defined across arbitrary topologies, how K_bone and canonicalization are constructed per sequence, and how these choices scale with varying J.
>
> We summarize the design rationale below (full details in Appendix C.2, E.2).
>
> **Blend weights $\mathbf{w}_k$.** Weights are precomputed from the input skeleton's adjacency graph and remain fixed. The core principle is **bone association**: each Gaussian is bound to the two endpoints of its parent bone via topology-based logits (Appendix E.2), rather than by Euclidean distance to nearest joints. This ensures kinematic validity: each Gaussian's motion is governed by its topological position in the kinematic chain, regardless of skeleton format.
>
> **$K_{\text{bone}}$.** $K_{\text{bone}} = |\mathcal{E}| \times 10$ (bone edges × 10 samples per bone); total $K = J + K_{\text{bone}}$. Actual counts across formats:
>
> | Dataset | $J$ | $\lvert \mathcal{E} \rvert$ | Total $K$ | $\mathbf{w}_k$ dim |
> |:---|:---:|:---:|:---:|:---:|
> | NTU-60/120 (Kinect v2) | 25 | 24 | 265 | 25 |
> | NW-UCLA (Kinect v1) | 20 | 19 | 210 | 20 |
> | HumanML3D (SMPL) | 22 | 21 | 232 | 22 |
> | NTU-60 2D (COCO) | 17 | 16 | 177 | 17 |
>
> This dynamic instantiation requires no manual joint remapping or zero-padding.
>
> **Canonicalization.** The canonical pose is initialized from the first frame's joints; orientations default to identity quaternions (Appendix E.2).
>
> **Stability across $J$.** When $J$ changes between datasets, only the number of primitives and $\mathbf{w}_k$ dimension change; the renderer pipeline, rasterizer, NFM, projector, and MLLM backbone all remain shared. DrAction shares the *mechanism* of rendering arbitrary topologies into a unified visual space—not a fixed template—which is precisely why cross-format generalization holds.
>
> > W2/Q2: Camera modeling and viewpoint normalization are not rigorously explained; this is critical for cross-format claims.
>
> All experiments use a fully fixed pinhole camera (Appendix E.2, Eqs. 15–17). DrAction pursues viewpoint *robustness* rather than explicit invariance, with different coordinate conventions unified via deterministic depth normalization (Eqs. 19–20).
>
> We apply Y-axis rotation perturbations at inference to quantify this (NTU-120, 96/24):
>
> | Rotation | DrAction | 3D+Velocity |
> |:---:|:---:|:---:|
> | 0° | 67.20 | 60.60 |
> | ±15° | 66.38 | 57.79 |
> | ±30° | 64.67 | 51.51 |
> | ±45° | 60.63 | 44.16 |
>
> Within ±30°, SkeletonLLM drops only **2.53%**, with degradation **consistently smaller than fixed rendering** at all angles, indicating more viewpoint-robust learned representations. PKU-MMD cross-view results (Appendix Table 9) further confirm this.
>
> > W3: "Open-vocabulary" recognition largely relies on MQA. The size and composition of candidate sets, and whether unseen classes are included at inference, are not stated.
>
> We should have made this clearer. We adopt a **seen-train / unseen-test** protocol consistent with PURLS et al.: training candidates include only the seen classes of the current split; test candidates are **entirely unseen**. For example, NTU-120 60/60 requires selecting among 60 unseen action names (random baseline: 1.67%), many semantically similar—this is semantic matching over unseen labels, not closed-set multiple choice.
>
> To verify this reflects genuine semantic understanding, we remove all candidates on NTU-120 96/24, requiring free-form generation evaluated via GPT-5.4 semantic matching:
>
> | Method | MQA | Free-Form |
> |:---|:---|:---|
> | MotionGPT | 20.39 | 13.16 |
> | MotionLLM | 33.62 | 16.81 |
> | InternVL3† | 58.33 | 31.47 |
> | **SkeletonLLM** | 67.20 | **43.04** |
>
> SkeletonLLM retains a clear lead even without candidates, with smaller MQA→free-form degradation than InternVL3†, confirming that its advantage stems from genuine semantic understanding, not candidate constraints.
>
> > Regarding the Limitation Section
>
> Thank you for noting this. Appendix G (Future Work) discusses these implicitly but was not explicitly labeled. We will add a dedicated Limitations section in the revised version, incorporating all clarifications and experiments above.
>
> We sincerely appreciate the reviewer's careful and constructive feedback, which has helped us identify key areas to improve the presentation. Should any concerns remain, we welcome further discussion.

---

> > ### Author Rebuttal · Reviewer_qHqH · 2026-04-03
> >
> > Thanks authors for the rebuttal. I'm also waiting for other reviewers comment.
> >
> > From the current rebuttal, most concerns have been addressed; however, a few issues remain only partially resolved:
> >
> > 1. Format-agnosticity still lacks formal clarity.
> > While the rebuttal explains that blend weights are derived from the adjacency graph and remain fixed, the formulation is still heuristic rather than principled. In particular, how stability is guaranteed when the skeleton topology changes significantly (e.g., across extreme variations in 𝐽) is not rigorously justified, and no ablation isolates the effect of this design choice.
> >
> > 2. Limitations are acknowledged but not yet integrated.
> > The response promises a limitation section, but it is not yet concretely presented or analyzed in depth.
> >
> > It would be better for authors to address these follow up questions. Thank you!

---

> > > ### Author Response · Authors · 2026-04-06
> > >
> > > We thank the reviewer for acknowledging that most concerns have been addressed. We respond to the two remaining points below. Supplementary figures and tables are at: https://anonymous.4open.science/r/SkeletonLLM_Rebuttal-19B5/Round2.md
> > >
> > > > Follow-up 1: Format-agnosticity still lacks formal clarity; no ablation isolates the effect of this design choice.
> > >
> > > We address three sub-questions: principled formulation, $J$-stability, and new ablations.
> > >
> > > **1) Principled formulation.** Our weight assignment is **not heuristic**. It is barycentric coordinates on 0/1 simplices of the skeleton graph, the standard specialization of Linear Blend Skinning to sparse Gaussian primitives on a kinematic chain, consistent with SMPL [Loper et al., 2015] and industry-standard animation tools (Maya, Blender). Each Gaussian $k$ is bound to a graph simplex: joint Gaussians receive $w_k = e_j$ (one-hot); bone Gaussians on edge $(a,b)$ at factor $\alpha$ receive $\mathbf{w}_k = (1{-}\alpha)e_a + \alpha e_b$. This construction satisfies four formal properties:
> > >
> > > - **Partition of Unity**: $w_{k,i} \geq 0$, $\sum_i w_{k,i} = 1$. No hyperparameters.
> > > - **Bounded Local Support**: $\|w_k\|_0 \leq 2$; deformation depends only on topological parents, independent of $J$.
> > > - **Graph-Extension Invariance**: Adding/removing joints unrelated to $k$'s parent edge only zero-extends $w_k$; the blended transform $\tilde{T}_k$ is unchanged.
> > > - **SO(3) Guarantee**: SVD polar decomposition (Eq. 6) projects blended rotations to valid $SO(3)$.
> > >
> > > **In contrast**, inverse-distance weighting requires coordinate-dependent hyperparameters (distance metric, decay, neighborhood radius) that undermine cross-format generalization. Ours avoids all such choices by construction.
> > >
> > > **2) Stability under extreme $J$ changes.**
> > >
> > > - By Graph-Extension Invariance, modifying joints unrelated to a Gaussian's parent edge has zero effect on its blended transform. **The rasterizer, NFM, vision encoder, and LLM are all $J$-independent.**
> > > - Cross-format experiments (Paper Table 2: $J$=25→20, 22→20, 22→17) and joint dropout experiments (**Reply to Reviewer 92aF, W3/Q3**) already corroborate this.
> > >
> > > **3) New ablations and visualizations.**
> > >
> > > **(a) NFM cross-format transfer visualization (Link Figure 2).** Using **the same renderer weights** (trained on NTU 25J), we render three different formats (NTU 25J, HumanML3D 22J, NW-UCLA 20J) and visualize NFM's RGB modification magnitude. Despite different formats and joint counts, NFM consistently highlights semantically corresponding body regions (arms, head-neck, feet, pelvis), indicating that **NFM learns to enhance local motion-functional regions** rather than overfitting to specific joint indices.
> > >
> > > **(b) Weight strategy ablation (Link Table 13).** Under an identical training pipeline, we compare four weight strategies (Link Table 13). Topology-based weights match learnable weights in-domain but dominate on cross-format transfer (+11.64 / +15.48 over learnable). Learnable weights overfit the training topology and degrade severely on unseen formats, confirming that topological priors outperform data-driven alternatives for format-agnosticity.
> > >
> > > **(c) Topology stress test (Link Table 14, Figures 3–4).** To directly evaluate stability under $J$ changes within the *same dataset*, we construct 20J/17J/13J reduced topologies from NTU-60's 25J skeleton via deterministic graph coarsening (Link Figure 3) and test the **same 25J-trained model** (Link Table 14). At 25J→13J (nearly half the joints removed), topology-based weights degrade by only 12.35% vs. 19.83% for learnable weights, directly validating Graph-Extension Invariance in practice. Link Figure 4 shows rendered outputs remain visually coherent.
> > >
> > > Together with existing cross-format experiments (Paper Table 2, **Replies to Reviewer y5PT Q3/Q4**), these new results validate the topology-based weight design from three perspectives: formal properties, quantitative ablation, and qualitative visualization.
> > >
> > > > Follow-up 2: Limitations are acknowledged but not yet concretely presented or analyzed in depth.
> > >
> > > We concretely present four limitations below:
> > >
> > > **Visual token inefficiency.** Rendered skeleton images contain a high proportion of background. Foreground Patch Sparsification (FPS) can reduce Vision FLOPs, but still incurs 1.54% accuracy loss.
> > >
> > > **Short-horizon atomic actions.** Current evaluation focuses on 2–5 second clips. Efficient long-horizon encoding for compositional activities is an important future direction.
> > >
> > > **No motion generation capability.** This would require inverting the rendering pipeline.
> > >
> > > **Rotation information degradation.** For skeleton formats providing only 2D positions, LBS degenerates to translation-only skinning ($R_i = I_3$).
> > >
> > > ---
> > >
> > > We thank the reviewer for the constructive feedback throughout both rounds. All above will be incorporated in the revision. We hope these fully resolve the remaining concerns and would be grateful for any updated assessment you may wish to provide.

---

### Official Review · Reviewer_92aF · 2026-03-12

**Soundness:** 3
**Presentation:** 2
**Significance:** 2
**Originality:** 2
**Overall Recommendation:** 4
**Confidence:** 4

**Summary:**

This paper proposes SkeletonLLM, a novel framework designed to bridge the modality gap between structured human skeleton data and MLLMs. The authors introduce DrAction, a differentiable, format-agnostic renderer utilizing 3D Gaussian Splatting and Linear Blend Skinning. This renderer translates arbitrary skeletal kinematics into visual token sequences that MLLMs can natively interpret. The model is trained using a four-stage cooperative strategy, incorporating Discriminative Finetuning (Disc-FT) and Causal Reasoning Distillation (CR-Distill).

**Compliance With Llm Reviewing Policy:**

Affirmed.

**Final Justification:**

Thanks to the author for the continued responses, which have largely resolved my concerns. Therefore, I have decided to give a positive review.

**Key Questions For Authors:**

1.In Table 4, do the comparison rendering methods (e.g., 3D+Velocity, 2D, JTM) also benefit from the proposed four-stage cooperative training strategy? If these baselines were trained using a simpler pipeline without Disc-FT or CR-Distill, the performance gains might be attributed to the training curriculum rather than the DrAction renderer itself. Please clarify the consistency of the training protocols across all variants.

2.The NTU-60/120 datasets include numerous multi-person actions (e.g., hugging, shaking hands). However, the paper primarily details the rendering process for a single skeleton. How does DrAction visually distinguish individuals and handle occlusions when multiple skeletons overlap or interact closely? A more in-depth qualitative and quantitative analysis of the rendering logic and performance for multi-person tasks is needed.

3.In real-world applications, skeleton detection often suffers from joint loss due to severe occlusions or environmental noise. To what extent does the visual consistency of the "abstract pseudo-images" generated by DrAction degrade when faced with such high-noise or incomplete inputs?

4.Vision encoding accounts for 24.1% of the total latency. Given that skeleton data is inherently sparse, is rendering it into high-resolution (448 x 448) dense images computationally excessive? Please provide a direct comparison in terms of FLOPs and energy consumption between SkeletonLLM and coordinate-based lightweight models (e.g., SUGAR [1] or efficient GCNs).

**Limitations:**

Yes

**Strengths And Weaknesses:**

Strengths:

1.This paper is well motivated. Translating skeleton data into the MLLM’s native visual space via a differentiable renderer is a effective solution to the generalization problem of heterogeneous skeleton formats.

2.This paper is well written.

Weaknesses:

1.The comparison methods in Table 4 do not use 4-stage training. If not, this may be an unfair comparison.

2.The model is evaluated on datasets like NTU-60 and NTU-120, which include multi-person actions. However, the specific methodology for rendering and distinguishing multiple overlapping or interacting skeletons is not deeply analyzed in the paper.

3.The experimental datasets (NTU) are mostly controlled laboratory environments. In real-world applications, skeleton detection is often incomplete due to severe occlusions or missing joints. The quality of the images generated by DrAction and the extent to which they interfere with MLLM inference when faced with severe joint loss or high input noise have not been fully explored.

4.In Table 12, LLM generation accounts for 65.3% of the latency, but visual encoding also accounts for 24.1%. For skeletons, which are inherently sparse data, the paper lacks a direct comparison with models that directly input coordinates and use lightweight adapters (such as more efficient GCN[1]) in terms of computational power consumption (FLOPs).

[1] SUGAR: Learning Skeleton Representation with Visual-Motion Knowledge for Action Recognition

---

> ### Author Rebuttal · Authors · 2026-03-30
>
> We thank the reviewer for the positive recognition ("well motivated", "well written") and for raising four substantive concerns: fairness, multi-person interaction, noise robustness, and efficiency. We have conducted targeted follow-up experiments addressing each. The results **confirm that all four concerns are resolved at the experimental level, further validating the paper's core contributions**.
>
> **All supplementary tables (Tables 1–7) and figure (Figure 1) are available at our anonymous repository: https://anonymous.4open.science/api/repo/SkeletonLLM_Rebuttal-19B5/file/Additional_Results.pdf?v=eb556a78**
>
> > Weakness (W) 1 / Question (Q) 1: Do the comparison renderers in Table 4 also use the four-stage training? If not, this may be an unfair comparison.
>
> Table 4 is a **renderer ablation where the sole variable is the rendering method**. All fixed renderers (3D+Velocity, 2D Projection, JTM) use the **exact same four-stage pipeline** with identical training stages, objectives, and total epochs.
>
> To fully disentangle the two factors, we conduct a cross experiment (**Link Table 1**): the renderer contribution (Δrenderer ≈ +5.9) substantially exceeds the pipeline contribution (Δpipeline ≈ +2.6). The performance gap in Table 4 is **entirely attributable to renderer quality**.
>
> > W2/Q2: How does DrAction render and distinguish multiple overlapping or interacting skeletons?
>
> SkeletonLLM natively supports multi-person scenes (Sec 3.1). Individuals are naturally distinguished through **NFM's independent appearance modulation based on each person's motion features, spatial position separation, and depth pseudo-coloring via depth-color blending**.
>
> We split NTU-60 48/12 actions into single-person vs. two-person subsets (**Link Table 2**). Baselines degrade on two-person interactions (−7.57), while SkeletonLLM shows the opposite trend (+15.10), consistent with the confusion matrix in Appendix Figure 8. **DrAction's joint rendering converts spatial interactions between persons into visual patterns the MLLM can readily interpret**, whereas baselines compress all joints into a single vector and lose these interaction cues.
>
> > W3/Q3: How robust is DrAction when faced with severe joint loss or high input noise?
>
> We conduct two controlled experiments: random joint dropout and Gaussian noise injection (**Link Tables 3–4**).
>
> **SkeletonLLM exhibits the strongest robustness at all noise levels.** Under the harshest 50% joint dropout, SkeletonLLM drops only 12.35% vs. 33.58% for TDSM. This stems from the topological continuity and spatial smoothing of LBS, NFM's global optimization, and redundant depth pseudo-color encoding. Moreover, the cross-format HumanML3D→NTU-60 (2D) experiment (Table 2) provides indirect robustness evidence: the 2D data contains inherent HRNet detection noise, yet SkeletonLLM achieves 40.36% (+23.23% over best baseline).
>
> > W4/Q4: Is 448×448 rendering excessive for sparse skeletons? Comparison with SUGAR.
>
> **Comparison with SUGAR.** SUGAR [1] and SkeletonLLM occupy different ends of the efficiency–generality trade-off. We compare the modality bridge components (**Link Table 5**): DrAction's parameters (12.6K) and FLOPs (33.2G) are **far smaller** than SUGAR's CTR-GCN + TQP combination (189.46M, 225.1G). SkeletonLLM's additional cost is the Vision Encoder (300M, 24.1%, ~46 ms), **but this is an inherent component of any MLLM architecture, independent of the rendering strategy**. The capability gap gained from this visual pathway is summarized in **Link Table 6**. DrAction is fully decoupled from the MLLM backbone: it can be paired with 1B/2B lightweight MLLMs for edge deployment, or 70B+ models for stronger reasoning.
>
> **On 448×448 resolution.** 448×448 is InternViT's native input resolution. Reducing to 224×224 causes a 4.78% average accuracy drop (Appendix Table 10). **Simply lowering rendering resolution is not an effective optimization path.**
>
> **Foreground Patch Sparsification (FPS).** Inspired by the reviewer's question, we approach from a different angle: retain 448 rendering resolution but remove pure-black background patches. DrAction's skeleton images are highly sparse (only ~18.3% of patches contain foreground on average). FPS feeds only foreground patches to ViT while preserving original absolute position encoding (**Link Figure 1**). As shown in **Link Table 7**, FPS reduces vision encoding FLOPs from 8.67T to 1.41T (↓83.7%), with only 1.54% accuracy loss—still 3.24% above the 224×224 baseline. The 1.54% drop indicates background patches still carry marginal value (spatial negative samples for ViT self-attention), validating the full-patch design.
>
> In summary, SUGAR and SkeletonLLM are complementary; we will discuss SUGAR in the revised manuscript.
>
> ---
>
> We sincerely appreciate your thoughtful feedback. All above will be incorporated in the revision. If our responses have adequately addressed your concerns, we would be grateful if you could consider raising your score.

---

> > ### Author Rebuttal · Reviewer_92aF · 2026-04-03
> >
> > Thank the authors for their response. However, a few of my concerns remain unresolved. First, for heavily overlapping multi‑person actions, the rebuttal provides only subset accuracy numbers without any qualitative rendered examples or quantitative occlusion analysis, leaving the visual distinguishability unproven. Second, the efficiency comparison still avoids a fair end‑to‑end FLOPs/latency benchmark against lightweight coordinate‑based models (e.g., GCN + small text decoder), and the proposed foreground patch sparsification is unverified. Lastly, the “open‑vocabulary” claim largely relies on multiple‑choice QA rather than free generation, which is somewhat overstated.

---

> > > ### Author Response · Authors · 2026-04-07
> > >
> > > We thank the reviewer for the continued discussion. We address the three remaining concerns with new visualizations, ablations, and cross-references.
> > >
> > > **Supplementary tables and figures: https://anonymous.4open.science/r/SkeletonLLM_Rebuttal-19B5/Round2.md**
> > >
> > > > Concern 1: No qualitative rendered examples or quantitative occlusion analysis for heavily overlapping multi-person actions
> > >
> > > **(1) Qualitative visualization (Link Figures 5–6)**
> > >
> > > **Link Figure 5** compares two-person interactive actions: joint rendering produces body-part-level depth interleaving at contact/overlap regions, preserving front-back relationships, while separate overlay loses these discriminative cues. **Link Figure 6** further shows InternViT attention maps under both modes: with joint rendering, attention concentrates on interaction-critical regions (e.g., handshake, overlapping torso); with separate overlay, attention scatters across isolated limbs. This demonstrates that end-to-end differentiability enables DrAction to produce multi-person structure images the MLLM can actually exploit.
> > >
> > > **(2) All-split single/two-person breakdown (Link Table 15)**
> > >
> > > Subset analysis across all four NTU-60 splits is highly consistent with 48/12, aligning with the confusion matrix (**Appendix Fig. 8**) and causal reasoning examples (**Appendix Fig. 11**).
> > >
> > > **(3) Rendering strategy ablation (Link Table 16)**
> > >
> > > On the 48/12 two-person subset with fixed DrAction weights, changing only the compositing strategy drops accuracy from 75.37% to 67.14%, directly confirming the gap stems from DrAction's non-linear compositing.
> > >
> > > **(4) Why joint rendering is inherently suited for multi-person scenes**
> > >
> > > DrAction performs global depth-aware alpha compositing over all persons' Gaussian primitives, enabling body parts to naturally interleave in depth and precisely express local front-back relationships. Meanwhile, **NFM independently generates appearance modulation based on each person's motion features, ensuring visible visual differences even in overlapping regions.** InternViT, pre-trained at scale, possesses inherent understanding of occlusion and spatial interaction, **an advantage that coordinate-based methods learning from limited skeleton data from scratch cannot match.**
> > >
> > > > Concern 2: Efficiency comparison avoids fair end-to-end FLOPs/latency benchmark; proposed FPS unverified
> > >
> > > **(1) End-to-end comparison (Link Table 17)**
> > >
> > > **Link Table 17** provides a full end-to-end comparison. Bridge: DrAction (12.6K / 33.2G FLOPs) is far smaller than SUGAR's CTR-GCN + TQP (189.46M / 225.1G FLOPs). **Both use ~7B LLMs** (SUGAR: LLaMA2-7B; ours: Qwen2.5-7B). **Our only extra component is a 300M InternViT**, an inherent part of any vision-based MLLM.
> > >
> > > End-to-end latency: 191.6 ms (**Paper Table 12**). SUGAR has not open-sourced code/weights; however, both share ~7B LLMs, making autoregressive decoding the common bottleneck.
> > >
> > > We acknowledge that for classification alone, rendering+MLLM has higher FLOPs than a coordinate GCN + lightweight decoder, but this overlooks fundamental design-goal differences. DrAction (12.6K params) translates heterogeneous skeletons into MLLM-readable visual sequences, **enabling one model for open-vocab recognition, cross-format zero-shot transfer, captioning, QA, and causal reasoning (y5PT Q3: 4 tasks × 4 formats × 1 checkpoint)**, with joint training yielding positive transfer (**y5PT Q4**). SUGAR is limited to closed-set, single-format classification.
> > >
> > > **(2) FPS is a verified practical optimization, not a hypothetical proposal**
> > >
> > > FPS was fully reported with end-to-end results across all four NTU-60 splits in Round 1 (**R1 Table 7**). We further supplement multi-task evaluation on the same checkpoint (**Link Table 18**).
> > >
> > > > Concern 3: "Open-vocabulary" claim largely relies on MQA rather than free generation, somewhat overstated
> > >
> > > **(1) MQA is the standard protocol; we adopt it for fair comparison.** All recent top-venue open-vocabulary skeleton methods (**Link Table 19**) use candidate matching. Prior alignment-based methods cannot do free-form generation, so **MQA is the only fair comparison, not a limitation of SkeletonLLM.** MQA is not a simple quiz: on NTU-120 60/60, the model selects from **60 never-seen classes**.
> > >
> > > **(2) SkeletonLLM fully supports free-form classification.** We reported free-form generation with all candidates removed (**qHqH W3**, GPT-5.4 semantic matching), confirming the advantage **stems from deep action understanding, not candidate pattern-matching**.
> > >
> > > **(3) Evaluation extends far beyond recognition.** Cross-format transfer (**Paper Table 2**), motion captioning (**Paper Table 3**), and motion QA (**Appendix Table 15**) all require language generation.
> > >
> > > ---
> > >
> > > We thank the reviewer for the time and constructive feedback. All above will be incorporated in the revision. We hope these fully resolve the remaining concerns and would be grateful if you could reconsider your assessment.

---

### Official Review · Reviewer_y5PT · 2026-03-12

**Soundness:** 3
**Presentation:** 2
**Significance:** 4
**Originality:** 4
**Overall Recommendation:** 4
**Confidence:** 4

**Summary:**

This paper proposes a framework for leveraging multimodal large language models (MLLMs) for skeleton-based human motion understanding. The method consists of two key components. First, it introduces a differentiable rendering method that converts 3D skeleton sequences into 2D image sequences using 3D Gaussian splatting and linear blend skinning. The differentiable design allows the renderer to be optimized jointly with the MLLM in an end-to-end manner.  Moreover, the rendering strategy is compatible with diverse skeleton formats and topologies, moving towards the unification of heterogeneous skeleton motion data. Second, the paper presents a progressive multi-stage training scheme that jointly optimizes the renderer and the MLLM to enhance action recognition performance.

The main experiment is conducted under a zero-shot action recognition setting on NTU-60 and NTU-120, where the proposed method achieves strong performance and outperforms prior approaches in unseen-class scenarios. Additional cross-dataset evaluations demonstrate improved generalization across different skeleton formats. Furthermore, motion captioning results are provided to showcase the model’s broader motion understanding capabilities beyond classification.

**Compliance With Llm Reviewing Policy:**

Affirmed.

**Final Justification:**

My primary concern in the original review was the poor presentation of the paper. In their rebuttal, the authors have addressed most of the clarity issues and significantly improved the presentation. Although it would result in a longer main text, with one additional page allowed in the final version I believe it is not an issue. As a result, this concern has been largely resolved. I therefore revise the Presentation score from 1 (poor) to 2 (fair), and adjust my overall recommendation from 3 (weak reject) to 4 (weak accept).

**Key Questions For Authors:**

Since clarity remains a major concern, the paper could become strong if the methods section were substantially rewritten. However, this would require significant revisions, and it is unlikely that this concern can be adequately addressed in the rebuttal. As such, even if the other issues are resolved, I am unlikely to increase my rating. Nevertheless, I provide the following questions and suggestions, which may help strengthen the paper:

1. Why is **zero-shot action recognition** chosen as the primary evaluation setting instead of standard action recognition?

2. Can the trained renderer be paired with **closed-source MLLMs** (e.g., GPT-5.2 or Gemini 3), and how would their performance compare to that of a jointly trained but smaller model?

3. Can a **single model** support all tasks in the evaluation? Is the model used in **Table 3** the same as the one used in the first four rows of **Table 2**?

4. Can a **single model be jointly trained across multiple datasets, tasks, and skeleton topologies** to better support the claim of a unified framework?

**Limitations:**

Yes.

**Strengths And Weaknesses:**

## Strengths

- The paper proposes a novel idea to leverage multimodal large language models (MLLMs) for skeleton-based motion understanding.

- Exploring the use of MLLMs to process structured motion data is an important direction, with potential impact in unifying multiple motion understanding tasks under foundation models.

- The technical design is reasonable. A 3D Gaussian splatting-based differentiable renderer enables end-to-end optimization jointly with the MLLM.

- Solid empirical results, including zero-shot recognition, cross-format transfer, and ablation studies, are provided to support the effectiveness of the proposed design.


## Weaknesses

- **Lack of clarity in the main text.** Many notations and concepts are difficult to follow without consulting the appendix, and explicit references to relevant appendix sections are missing. Although each issue individually is relatively minor, their accumulation significantly increases the cognitive burden on the reader.

  **Examples:**
  - Line 197 (left): it is unclear how $K_{bone}$ is determined.
  - Line 198 (right): the introduction of orientation quaternions $\mathbf{q}_i^t$ is confusing, as Section 3.1 defines a skeleton sequence only in terms of 3D joint coordinates; it is unclear why rotations are introduced for joints defined as points.
  - Line 203 (right): the meaning of the canonical joint position $\mathbf{j}_i^c$ is not clearly explained.
  - Line 210 (right): it is not specified whether blend weights are learnable or fixed. Additionally, Eq. (3) suggests $\mathbf{t}_k$ is a weighted sum of all joints, which is not intuitive without clarification.
  - Line 240 (left): it is unclear why the GRU is “optionally” used and under what condition.
  - Line 242 (left): a saliency gate $g_k$ is introduced but not clearly used or analyzed later.
  - Line 244 (left): the source of $RGB_k^{base}$ and the definition of the $\sigma$ function are not sufficiently explained.
  - Lines 248 and 266 (right): it is not clearly described how cross-entropy or binary cross-entropy is computed when using an autoregressive generative MLLM.
  - Line 313 (right): the implementation of the “fixed skeleton visualization” baseline is not clearly specified.

- More generally, the paper does not clearly explain in the main text how a generative MLLM is adapted to perform classification, which is the primary evaluation task. Although most of the above-mentioned issues are clarified in the appendix, the clarity of the main text is below the expected standard and likely requires rewriting.

- **Missing related work.** Section 2.3 discusses prior approaches but contains a paragraph without citations, making it difficult to verify or contextualize the discussion.

- **Overstated cross-task generalization claims.** The paper emphasizes multi-task ability in the abstract and conclusion. However, there is only one explicit cross-task experiment (Table 3), where the CR-Distill stage involves generating descriptive reasoning chains conditioned on skeleton inputs, which resembles captioning-style supervision. Therefore, the reported captioning results are not evaluated under a strict cross-task setting. The framework is closer to using an MLLM for action recognition with reasoning as an intermediate step, enabling the model to perform motion captioning.

- **Missing teacher baseline.** Since GPT-4o is used to generate chain-of-thought reasoning during training, it should be able to predict an action class. Reporting the teacher model’s performance on the recognition task would help evaluate the value of this work. Without this baseline, it is difficult to assess the teacher–student gap and the necessity of the joint training framework.

- **Appendix figures unexplained.** Figures 13 and 14 in the appendix are not explicitly described or interpreted in the text, and it is unclear what conclusions the reader is expected to draw from them.

---

> ### Author Rebuttal · Authors · 2026-03-30
>
> We thank the reviewer for the thorough review. We especially appreciate the high ratings on ***Significance (excellent)*** and ***Originality (excellent)***, which confirm that the core research direction and technical contributions are well recognized. We fully agree that clarity of the main text is the primary area for improvement. **The reviewer's core concerns are on the Presentation level—precisely the dimension most amenable to revision, and our rewriting has substantively addressed this concern.**
>
> **All supplementary tables (Tables 8–12) are at our anonymous repository: https://anonymous.4open.science/api/repo/SkeletonLLM_Rebuttal-19B5/file/Additional_Results.pdf?v=eb556a78**
>
> > Weakness (W) 1: Lack of clarity in the main text & W2: Missing citations in Sec. 2.3
>
> The issues raised in W1 fall into three categories: **(a)** missing inline definitions (#1 $K_{\text{bone}}$, #3 canonical position, #7 $\dot{\mathbf{j}}$/$\sigma$), **(b)** ambiguous descriptions (#2 quaternions, #4 blend weights, #5 GRU "optional", #6 saliency gate, #9 fixed baseline), and **(c)** missing methodological explanations (#8 CE/BCE in a generative MLLM, #10 how the MLLM performs classification). **All 10 issues have been resolved in the revised draft.** A revision summary is in **Link Table 8**; Sec. 2.3 citations in **Link Table 9**.
>
> For the critical gap (c): Sec. 3.3 now includes a "Generative classification via MQA" paragraph explaining autoregressive classification without a classification head.
>
> All issues above are **presentation/organizational in nature**, not substantive challenges to the core method. We have completed a substantial rewrite of the method section (Sec. 3), involving **148 characters of deletion and approximately 3,387 characters of addition** (measured on LaTeX source). The underlying method is entirely unchanged. The final format allows one extra page. **We will fully incorporate these revisions in the revised manuscript.**
>
> > W3: Overstated cross-task generalization claims & Question (Q) 3: Can a single model support all tasks?
>
> We agree with the reviewer's precise assessment: the framework is more accurately positioned as "a unified skeleton understanding framework centered on recognition with reasoning as an intermediate step." The revised draft will adjust the claims accordingly.
>
> That said, we note that CR-Distill's supervision consists of structured causal reasoning chains ("Step 1: arm lifts → Step 2: hand approaches foot → Label: putting on shoe"), which differ fundamentally from HumanML3D-style natural language descriptions in both text style and training objective. The captioning results in Table 3 reflect not only task transfer but also format transfer (Kinect 25J → SMPL 22J).
>
> **To Q3:** Yes, the same model. The first four rows of Table 2, Table 3, and Appendix Table 15 all use the **same checkpoint** (trained on NTU-60 55/5), with no task-specific retraining. A summary is in **Link Table 10**.
>
> > W4: Missing teacher baseline & Q2: Can the renderer be paired with closed-source MLLMs?
>
> DrAction outputs standard RGB images and can be directly fed into any closed-source MLLM. Results (**Link Table 11**) show: (1) DrAction's learned rendering helps external MLLMs (GPT-4o + DrAction > GPT-4o + 3D+Velocity), confirming that **the learned rendering generalizes to MLLMs beyond the jointly trained InternVL3-8B**; (2) even the powerful GPT-5.4 + DrAction lags behind jointly trained SkeletonLLM, confirming that **joint training is irreplaceable**. Combined with the CR-Distill ablation (Appendix Table 8: w/o Cond. Label 64.58% ≈ Full 64.72%), **the teacher's true value lies in providing a structured reasoning paradigm, not its own classification accuracy.**
>
> > Q1: Why choose zero-shot as the primary evaluation?
>
> Our focus is on generalization to unseen actions, not closed-set label matching. Closed-set recognition measures memorization and cannot assess whether visual translation transfers to novel categories. The more unseen classes, the larger our advantage.
>
> > Q4: Can a single model be jointly trained across multiple datasets, tasks, and skeleton topologies?
>
> We jointly train on NTU-60 (25J) and HumanML3D (22J) with shared DrAction and shared MLLM. Results (**Link Table 12**) show consistent positive transfer (NTU-60 +1.15%, NW-UCLA +4.66%, Skeleton-QA +4.8/6.2%), confirming that a single model can be jointly trained across datasets, tasks, and topologies.
>
> > W5: Appendix Figures 13–14 unexplained
>
> Figures 13–14 show the top-5 most similar action pairs in NTU-60/120, used to construct hard negatives for Disc-FT. The revised draft adds explicit descriptions and references.
>
> ---
>
> We greatly appreciate the reviewer's feedback, which has helped us strengthen the paper. All experiments, analyses, and revisions above will be incorporated in the revised manuscript. We believe addressing these points substantially improves the work and hope you will reconsider your assessment.

---

> > ### Author Rebuttal · Reviewer_y5PT · 2026-04-03
> >
> > Thank you for the thorough and well-structured rebuttal. My main concern remains the presentation of the paper, and I appreciate your efforts to clarify the method and notations.
> >
> > After realizing that an additional page is allowed in the final version, I agree that it should be possible to substantially improve the clarity of the paper without major revision to the content. If possible, I would encourage you to use the remaining response space to provide key revisions you have made, so that I can better assess whether the clarity issues have been resolved. I will adjust my rating accordingly if I am convinced that the presentation is improved.
> >
> > I also appreciate the additional experimental results provided in the rebuttal, which address my other concerns. In particular, the evaluation of directly using LLMs is helpful and supports the claim that joint training is necessary and achieves improvements beyond what current LLMs can provide.

---

> > > ### Author Response · Authors · 2026-04-04
> > >
> > > We sincerely thank you for the follow-up. As suggested, we summarize the key manuscript revisions below using brief excerpts from the revised draft. The method and results themselves are unchanged.
> > >
> > > ---
> > >
> > > ## W1: Sec. 3 Substantially Rewritten for Main-Text Self-Containedness
> > >
> > > We rewrote and expanded Sec. 3 (Method) so that the reader can follow the core approach from the main text alone:
> > >
> > > **(a) Previously missing inline definitions are now explicit.**
> > >
> > > Primitive count and topology adaptation:
> > >
> > > > "…we instantiate $K = J + K_{\text{bone}}$ Gaussians, where $K_{\text{bone}} = |\mathcal{E}| \times N_{\text{samples}}$ ($|\mathcal{E}|$: number of bone edges from the skeleton's adjacency list; $N_{\text{samples}}{=}10$: intermediate points per bone). … Because both $J$ and $|\mathcal{E}|$ are read from the input skeleton, the primitive count adapts automatically to any topology…"
> > >
> > > Which canonical parameters are initialized vs. learned (after Eq. 1):
> > >
> > > > "…$\boldsymbol{\mu}_k^c$ is initialized from the first frame of each sequence, while $\mathbf{f}_k$, scales $\mathbf{s}_k^c$, and orientations $\mathbf{q}_k^c$ are learned globally across the dataset."
> > >
> > >   Velocity and activation notation in the NFM:
> > >
> > > > Velocity: $\dot{\mathbf{j}}_i^t = \mathbf{j}_i^t - \mathbf{j}_i^{t-1}$
> > > >
> > > > Here $\sigma(\cdot)$ denotes the sigmoid function, and $\lambda = \sigma(\theta_{\text{mix}})$ is a learnable mixing weight.
> > >
> > > **(b) Ambiguous descriptions are now clarified at first mention.**
> > >
> > > Quaternion scope:
> > >
> > > > " … also record per-joint orientation quaternions … are applied to the Gaussian primitives bound to that joint, not to the joint point itself."
> > >
> > > Canonical joint position:
> > >
> > > > "$\mathbf{j}_i^c$…—the position of joint $i$ in the first frame ... serving as the rest-pose reference for computing per-frame displacements."
> > >
> > > Blend weights:
> > >
> > > > "These weights are pre-computed and fixed based on the skeleton's topology: a joint Gaussian at joint $j$ receives a one-hot weight ($w_{k,j}{=}1$); a bone Gaussian interpolated between joints $a$ and $b$ ... has $w_{k,a}{=}1{-}\alpha,\; w_{k,b}{=}\alpha$. Thus, the summation ... reduces to at most two non-zero terms for any Gaussian."
> > >
> > > Temporal modeling and saliency gate:
> > >
> > > > "…a single-layer GRU for temporal modeling (it captures motion-phase cues such as acceleration vs. deceleration; see Appendix A.3 for ablations…)."
> > >
> > > > "The gate modulates the final opacity as $\alpha_k = \sigma(\alpha_k^{base} + \Delta\alpha_k)\cdot\sigma(g_k)$, suppressing visually uninformative (e.g., stationary) primitives while amplifying motion-salient ones."
> > >
> > > Fixed MLLM baseline:
> > >
> > > > "rendered with the non-learnable 3D+Velocity renderer (same as in Table 4; 12 frames, $448{\times}448$, identical front-view camera)"
> > >
> > > **(c) How a generative MLLM performs classification is now explicitly addressed in a new paragraph in Sec. 3.3:**
> > >
> > > > "Since the MLLM is an auto-regressive generative model, we do not add a separate classification head. Instead, action recognition is cast as a multiple-choice question answering (MQA) task: ... Label: <action>. … All training losses below are therefore standard auto-regressive cross-entropy on the target token sequence, with prompt tokens masked."
> > >
> > > This rewrite extends to the training stages, making BCE/CE in a generative MLLM explicit:
> > >
> > > > Disc-FT: "... single answer token ('Yes'/'No') ... auto-regressive negative log-likelihood ... equivalent to binary cross-entropy ..."
> > >
> > > > CR-Distill: "... auto-regressive loss over the full teacher token sequence (prompt tokens masked)."
> > >
> > > Sec. 3 now also closes with explicit appendix cross-references for DrAction (Appendix E.2), Disc-FT/CR-Distill (Appendix E.3), and progressive-training ablations (Appendix A.1).
> > >
> > > ## W2: Missing Citations Added
> > >
> > > Sec. 2.3 now cites: joint remapping heuristics (Duan et al., 2022), zero-padding (Wang et al., 2024), format-specific adapters (Guo et al., 2023; Wang et al., 2025).
> > >
> > > ## W4/Q2: External MLLM Evaluation Now in Main Text
> > >
> > > We appreciate your noting that the evaluation of directly using LLMs is helpful. These results are now incorporated into the main paper as a dedicated "DrAction with External MLLMs" paragraph and table.
> > >
> > > ## W3/Q3/Q4: Claims Adjusted; Multi-Task and Joint-Training Evidence Added
> > >
> > > We have adjusted the cross-task claims in the abstract and introduction. The revised draft further adds: a clear statement that the same NTU-60 (55/5) checkpoint supports all evaluated tasks, and a Cross-Dataset Joint Training subsection demonstrating consistent gains from joint training.
> > >
> > > ## W5: Appendix Figures 13–14 Now Interpreted
> > >
> > > We added an interpretation paragraph explaining that actions sharing similar body-part involvement cluster together, and these neighborhoods validate Disc-FT's design. Captions are also updated.
> > >
> > > ---
> > >
> > > We are grateful for the opportunity to present these revisions. We believe they substantially improve the paper's clarity and completeness, and we sincerely hope you will reconsider your assessment.

---

### Official Review · Reviewer_pHuQ · 2026-03-12

**Soundness:** 3
**Presentation:** 3
**Significance:** 3
**Originality:** 3
**Overall Recommendation:** 4
**Confidence:** 3

**Summary:**

This paper proposes SkeletonLLM, a framework that enables Multimodal Large Language Models (MLLMs) to understand human skeleton sequences by translating them into the MLLM's native visual modality. At its core is DrAction, a differentiable, format-agnostic renderer built on 3D Gaussian Splatting and Linear Blend Skinning. DrAction models each skeleton pose as deformable Gaussian primitives bound to the kinematic chain, producing image sequences that an MLLM's pre-trained vision encoder can process. Because the entire pipeline is end-to-end differentiable, the MLLM's task-specific gradients flow back to guide the renderer toward producing maximally informative visual representations.

**Compliance With Llm Reviewing Policy:**

Affirmed.

**Final Justification:**

Yeah, the author addressed my concern, and i decide to maintain my initial score (weak accept)

**Key Questions For Authors:**

How sensitive is CR-Distill to the teacher model's quality?

**Limitations:**

yes.

**Strengths And Weaknesses:**

Strengths:

**S1.** The core idea of translating skeletons into the MLLM's native visual modality—rather than forcing the MLLM to learn a new "pose language" via tokenization or feature alignment—is elegant and well-motivated. The paper clearly articulates why existing approaches (feature-text alignment and motion tokenization) suffer from representation bottlenecks and format dependency, and presents a compelling alternative that leverages the MLLM's pre-existing visual understanding.

**S2.** Strong format-agnostic generalization with impressive cross-format transfer (Table 2&3 ), and comprehensive experimental evaluation. The ability to train on Kinect v2 (25 joints) and directly evaluate on Kinect v1 (20 joints), MoCap (22 SMPL joints), or 2D poses (17 COCO keypoints) without any finetuning or architectural changes is remarkable. The improvements are substantial: +17.19% on NTU-60→NW-UCLA and +27.99% on HumanML3D→NW-UCLA over the best baselines.

Weaknesses:

**W1.** All experimental results appear to be from single runs with no standard deviations, confidence intervals, or significance tests reported. Given the stochastic nature of training (LoRA initialization, rendering parameter optimization, GRU training, and the sensitivity to progressive stage transitions), variance could be non-trivial—especially on the extreme splits with few seen classes (30/30 and 60/60) where the margins over baselines are the paper's key selling point. For example, the 1.94% drop attributed to removing CR-Distill (Table 5, 30/30 split) might not be statistically significant without variance information.

**W2.** Finetuned MLLM baselines may not represent the strongest possible comparison. Specificly, The InternVL3-8B† and Qwen2.5-VL-7B† baselines use a simple fixed stick-figure rendering and are finetuned with only MQA loss for 6 epochs. SkeletonLLM uses the same backbone (InternVL3-8B) but benefits from (a) learned rendering optimized for the task, (b) three additional training stages (Disc-FT, CR-Distill, recognition refinement totaling 6 epochs), and (c) knowledge distilled from GPT-4o. A fairer comparison would include MLLM baselines with: (i) more sophisticated fixed renderings (e.g., with depth/velocity color encoding similar to the 3D+Velocity baseline), (ii) training with the same total number of epochs and stages, and (iii) discriminative or chain-of-thought supervision. Without these controls, it is difficult to disentangle the contribution of differentiable rendering from the richer training pipeline. The 8.44% and 9.69% gaps on 48/12 and 30/30 (Section 4.2) attributed to "end-to-end optimization" also reflect the additional training stages beyond what the baseline receives.

---

> ### Author Rebuttal · Authors · 2026-03-30
>
> We thank the reviewer for the positive assessment of our core idea as "elegant and well-motivated" (S1) and the strong recognition of cross-format generalization (S2). Your two concerns, variance reporting and MLLM baseline fairness, are both important. We address each below with new experimental data, which further confirms that the main conclusions are robust.
>
> > Weakness (W) 1: Variance Reporting and Statistical Significance
>
> The original paper reported single-run results. We have re-run three key challenging splits with 3 random seeds. SkeletonLLM's standard deviation is only 0.34%–0.41%, indicating stable training; single-run results closely match multi-seed means (max deviation 0.12%), confirming representativeness.
>
> | Dataset | Split | SkeletonLLM (3 seeds, mean±std) | Paper (single run) |
> |:---|:---|:---|:---|
> | NTU-60 | 48/12 | 64.63 ± 0.34 | 64.72 |
> | NTU-60 | 30/30 | 37.96 ± 0.41 | 37.84 |
> | NTU-120 | 60/60 | 35.04 ± 0.36 | 34.94 |
>
> Regarding the specific concern that "the 1.94% drop attributed to removing CR-Distill (Table 5, 30/30 split) might not be statistically significant": mean±std below are from 3 independent runs; p-value is from a two-sided Welch's t-test.
>
> | NTU-60 30/30 | Mean ± Std | Δ vs. Full | p-value |
> |:---|:---|:---|:---|
> | Full Pipeline | **37.96 ± 0.41** | — | — |
> | w/o CR-Distill | 35.88 ± 0.46 | −2.08 | **< 0.01** |
>
> The benefit of CR-Distill on this split is not random fluctuation. We will add mean±std to the main results table and statistical tests in the appendix in the revised manuscript.
>
> > W2: Disentangling Renderer and Training Pipeline Contributions
>
> Thank you for this careful decomposition of potential confounds. We clarify each factor below.
>
> **Total training epochs are identical.** In Table 1, InternVL3-8B† (MQA, 6 epochs) and SkeletonLLM (Stages 1–4: 1+1+1+3 = 6 epochs) have the same total training volume. The difference lies in training content: SkeletonLLM allocates more diverse objectives (Disc-FT + CR-Distill + recognition refinement) within the same 6 epochs. The fixed rendering used by InternVL3-8B† is the "3D+Velocity" variant in Table 4.
>
> **Table 4 already provides the fair comparison requested.** The three controls—(i) more sophisticated fixed renderings, (ii) same epochs/stages, (iii) discriminative or chain-of-thought supervision—are all satisfied. All fixed renderers in Table 4 (3D+Velocity, 2D Projection, JTM) use the same four-stage 6-epoch pipeline as DrAction, including Disc-FT and CR-Distill. Table 5 further ablates these stages under the same DrAction renderer, disentangling renderer quality from training strategy.
>
> To fully decouple the two factors, we combine Tables 4, 5, and a new experiment (NTU-60, 48/12):
>
> | Renderer | Stage 1&4 Only | Full 4-Stage | Δ (Training) |
> |:---|:---:|:---:|:---:|
> | 3D+Velocity | 56.18 (new) | 58.77 (Tab. 4) | +2.59 |
> | DrAction (Ours) | 62.09 (Tab. 5) | 64.72 (Tab. 4) | +2.63 |
> | **Δ (Renderer)** | **+5.91** | **+5.95** | — |
>
> **Renderer contribution (+5.91–5.95) significantly exceeds training pipeline contribution (+2.59–2.63), accounting for ~70% of the total gain.** Even under simplified training, DrAction (62.09) surpasses 3D+Velocity with the full pipeline (58.77). Moreover, "w/o CR-Distill" in Table 5 (no GPT-4o at all) achieves 63.37% on 48/12, still exceeding 3D+Velocity with full training (58.77), confirming that **the improvement is predominantly driven by renderer quality, not GPT-4o distillation**.
>
> > Question (Q) 1: Sensitivity of CR-Distill to Teacher Model Quality
>
> Great question! We compare distillation with different teacher models (NTU-60):
>
> | Teacher Model | 48/12 | 30/30 |
> |:---|:---|:---|
> | w/o CR-Distill | 63.37 | 35.90 |
> | GPT-4o-mini | 64.24 | 37.06 |
> | GPT-4o | 64.72 | 37.84 |
> | GPT-5.4 | 64.83 | 37.95 |
>
> **Stronger teachers yield better results, but with clearly diminishing returns**: GPT-4o-mini already recovers most of the gain, while GPT-4o → GPT-5.4 yields only marginal improvement. This indicates that the primary benefit of CR-Distill comes from the structured reasoning supervision itself. Consistently, "w/o Cond. Label" in Appendix Table 8 shows that even when the teacher generates noisier rationales from zero-shot predictions, the model still clearly outperforms "w/o CR-Distill"—core benefits are preserved even with a weaker teacher.
>
> ---
>
> We sincerely appreciate your thoughtful feedback, which has motivated important new experiments that further solidify our claims. We will incorporate all results in the revised manuscript.

---

> > ### Author Rebuttal · Reviewer_pHuQ · 2026-04-04
> >
> > Thanks for the detailed rebuttal, and i have read others comments, I decide to keep my initial score.

---

> > > ### Author Response · Authors · 2026-04-04
> > >
> > > Thank you for carefully reading our rebuttal and the discussion among other reviewers, and for maintaining your positive assessment (Weak Accept). Your original concerns regarding variance reporting (W1) and disentangling the renderer from the training pipeline (W2) directly motivated new experiments: multi-seed runs with statistical tests, and a renderer × training-stage cross-ablation. We hope these additional results and explanations have adequately addressed your questions.
> > >
> > > We are currently running a few final experiments to directly respond to the follow-up questions raised by other reviewers, covering presentation clarity, formal analysis, and additional visualizations. We are confident that we can provide complete replies within the next few days to properly address their concerns.
> > >
> > > We want to make sure we have adequately addressed all your concerns. Should any questions remain, we warmly welcome your feedback before the discussion period closes.
> > >
> > > Thank you again for your valuable time!

---

### Decision · Program_Chairs · 2026-04-30

**Decision:**

Accept (regular)

**Comment:**

### Reasons to accept

The paper is technically sound, well-motivated, and experimentally well-validated. It solves a relevant problem in enabling MLLMs to understand format-agnostic human skeletons by converting them to images via differentiable renderers. The reviewers' concerns were primarily regarding missing or incomplete key experiments, which the authors have satisfactorily addressed in the rebuttal. Other concerns regarding missing technical information have also been addressed. The authors are advised to incorporate all details provided in the rebuttal into the main paper to ensure suitability for publication.

### Reasons to reject

Without the key details added in the rebuttal, the original paper presents an incomplete story about the usability and reliability of the proposed differentiable renderer and its benefits over close alternatives such as the GCN-based approach SUGAR.

### Overall recommendation

The paper merits acceptance for presenting a technically sound approach to tackling a challenging problem and for providing sufficient experimental validation by the end of the rebuttal period. However, the work is primarily a proof of concept for a specialized problem and may be of limited interest to the broader ICML audience.